# What AlphaFold tells us about cohesin's retention on and release from chromosomes

Kim A Nasmyth[1]*, Byung-Gil Lee[2†], Maurici Brunet Roig[1], Jan Löwe[2]

[1]Department of Biochemistry, University of Oxford, Oxford, United Kingdom; [2]MRC Laboratory of Molecular Biology, Cambridge, United Kingdom

*For correspondence:
ashley.nasmyth@bioch.ox.ac.uk

Present address: †Department of Biochemistry, Lee Gil Ya Cancer and Diabetes Institute, Gachon University College of Medicine, Incheon, Republic of Korea

Competing interest: The authors declare that no competing interests exist.

**Abstract** Cohesin is a trimeric complex containing a pair of SMC proteins (Smc1 and Smc3) whose ATPase domains at the end of long coiled coils (CC) are interconnected by Scc1. During interphase, it organizes chromosomal DNA topology by extruding loops in a manner dependent on Scc1's association with two large hook-shaped proteins called SA (yeast: Scc3) and Nipbl (Scc2). The latter's replacement by Pds5 recruits Wapl, which induces release from chromatin via a process requiring dissociation of Scc1's N-terminal domain (NTD) from Smc3. If blocked by Esco (Eco)-mediated Smc3 acetylation, cohesin containing Pds5 merely maintains pre-existing loops, but a third fate occurs during DNA replication, when Pds5-containing cohesin associates with Sororin and forms structures that hold sister DNAs together. How Wapl induces and Sororin blocks release has hitherto remained mysterious. In the 20 years since their discovery, not a single testable hypothesis has been proposed as to their role. Here, AlphaFold 2 (AF) three-dimensional protein structure predictions lead us to propose formation of a quarternary complex between Wapl, SA, Pds5, and Scc1's NTD, in which the latter is juxtaposed with (and subsequently sequestered by) a highly conserved cleft within Wapl's C-terminal domain. AF also reveals how Scc1's dissociation from Smc3 arises from a distortion of Smc3's CC induced by engagement of SMC ATPase domains, how Esco acetyl transferases are recruited to Smc3 by Pds5, and how Sororin prevents release by binding to the Smc3/Scc1 interface. Our hypotheses explain the phenotypes of numerous existing mutations and are highly testable.

## eLife assessment

This **important** study makes use of AlphaFold2 to predict the models of tens of cohesin subcomplexes from different species. The models, which are in most cases consistent with published cohesin variants with compromised in vitro and in vivo cohesin activity, provide **convincing** evidence that leads to testable hypotheses of cohesin dynamics and regulation. More broadly, this study serves as an example of how to use AlphaFold2 to build models of protein complexes that involve the docking of flexible regions to globular domains.

## Introduction

Cohesin is a multi-subunit complex responsible for determining much of the topology of chromosomal DNA during interphase as well as sister chromatid cohesion (*Yatskevich et al., 2019*). At its heart are a pair of rod-shaped Structural Maintenance of Chromosomes (SMC) proteins, Smc1 and Smc3, that have a dimerization domain called the 'hinge' at one end and an ABC-like ATPase domain at the other, separated by 50-nm-long intra-molecular antiparallel coiled coils. These are interrupted in two places, at an 'elbow,' 16 nm from the hinge, around which the coiled coils fold onto each other (*Bürmann*

*et al., 2019*), and a disruption of the coiled coil called the 'joint' (*Diebold-Durand et al., 2017*), about 6.5 nm from the ATPase domains (*Figure 1*).

In the presence of ATP, the Smc1 and Smc3 ATPase domains associate with each other, thereby sandwiching a pair of ATP molecules between them (*Shi et al., 2020*). Crucially, they are also interconnected by a kleisin subunit (*Schleiffer et al., 2003*), called Scc1, whose N- and C-terminal domains bind to the 'base' and 'neck' of the Smc1 and Smc3 ATPases, respectively, creating a huge tripartite SMC-kleisin ring, even in the absence of ATP (*Chapard et al., 2019*; *Haering et al., 2002*; *Figure 1*). Cohesin's various activities depend on the association with the largely unstructured sections of the kleisin polypeptide connecting its N- and C-terminal domains, of three large hook-shaped HEAT repeat containing regulatory subunits called HAWKs (HEAT repeat proteins Associated With Kleisins). Only two of these are thought to be bound at any one time, Scc3 (human: SA) with either Pds5 or Scc2 (human: Nipbl). Complexes containing Scc2/Nipbl are active as DNA-dependent ATPases, while those containing Pds5 are not (*Petela et al., 2018*).

In addition to holding sister chromatids together in post-replicative cells, cohesin is capable of extruding long loops of DNA (*Davidson et al., 2019*; *Kim et al., 2019*). This loop extruding (LE) activity occurs throughout interphase in mitotic cells and during prophase in meiosis when it is essential for the formation of the synaptonemal complex (*Klein et al., 1999*). Importantly, LE and sister chromatid cohesion are conferred by distinct populations of cohesin, as LE requires the continuous presence of Scc2/Nipbl (*Mitter et al., 2020*), whereas cohesion does not (*Ciosk et al., 2000*; *Srinivasan et al., 2019*). LE facilitates VDJ recombination (*Zhang et al., 2022*), as well as the interaction of enhancers with distant promoters and is regulated throughout the genome by the site-specific DNA binding protein CTCF (*Fudenberg et al., 2016*; *Li et al., 2020*).

Cohesin's initial association with chromosomes as well as active LE are mediated by Smc-kleisin rings occupied by Scc2/Nipbl and Scc3/SA (*Figure 2*). LE is halted at asymmetric CTCF sites, but only when they face in one direction (*Fudenberg et al., 2016*). This arrest depends on modification by yeast Eco1 (*Bastié et al., 2022*) or human Esco1 and 2 (*van Ruiten et al., 2022*; *Wutz et al., 2020*) acetyl-transferases of a pair of lysine residues (yeast Smc3K112 and K113; human Smc3K105 and K106) on the ATPase domain of Smc3 (*Unal et al., 2008*), an event that depends on replacement of Scc2/Nipbl by Pds5 (*Chan et al., 2013*; *Figure 2*). Unlike Scc2/Nipbl, which is required for LE and for building but not maintaining cohesion, SA/Scc3 is essential for all of cohesin's activities, namely loading onto chromatin and loop extrusion, as well as the establishment and maintenance of cohesion (*Davidson et al., 2019*; *Roig et al., 2014*; *Tóth et al., 1999*). Unrestricted LE between convergent CTCF sites, but not past them, ensures that any sequence between convergent sites is more likely to be associated with other sequences within this interval than with those beyond these sites, giving rise to the phenomenon of topologically associated domains (TADs) in HiC genome contact maps (*Rao et al., 2017*). The inhibition of cohesin's LE by CTCF is presumed to depend on Nipbl's replacement by Pds5, as well as acetylation of Smc3 by Esco1 and is accompanied by cohesin's association with CTCF sites (*Parelho et al., 2008*; *Wendt et al., 2008*).

Post-replicative cells have three major types of cohesin complexes: those holding sisters together (sister chromatid cohesion), those actively extruding loops, and those whose LE has been arrested by CTCF and which hold together convergent CTCF sites at TAD boundaries. Crucially, all three are susceptible to the action of a fourth regulatory subunit called Wapl (*Chan et al., 2012*; *Gandhi et al., 2006*; *Kueng et al., 2006*), which induces cohesin's release from chromatin at a certain frequency, albeit only when Nipbl/Scc2 has been replaced by Pds5 and when Smc3(K105) and K106 are unmodified (*Figure 2*). The increased processivity of cohesin's LE upon Wapl's artificial depletion induces formation of chromatid-like structures during interphase (*Tedeschi et al., 2013*), a process that occurs naturally during meiosis (*Klein et al., 1999*). Moreover, Wapl's downregulation during B cell development is essential for recombination between DJ junctions and distal V genes (*Hill et al., 2020*), which requires highly processive LE, mediated by cohesin to bring them together in the correct orientation so that they can be joined by the RAG recombinase.

Because its maintenance does not require Scc2/Nipbl, sister chromatid cohesion is thought to be a passive structure that does not require continual ATP hydrolysis. Once generated during replication, cohesion must also be shielded from Wapl. This protection depends on Smc3 acetylation, and in animal and plant cells on cohesin's association during S phase with a protein called Sororin (*Nishiyama et al., 2010*; *Rankin et al., 2005*). When cells enter M phase, Sororin's phosphorylation by Cdk1

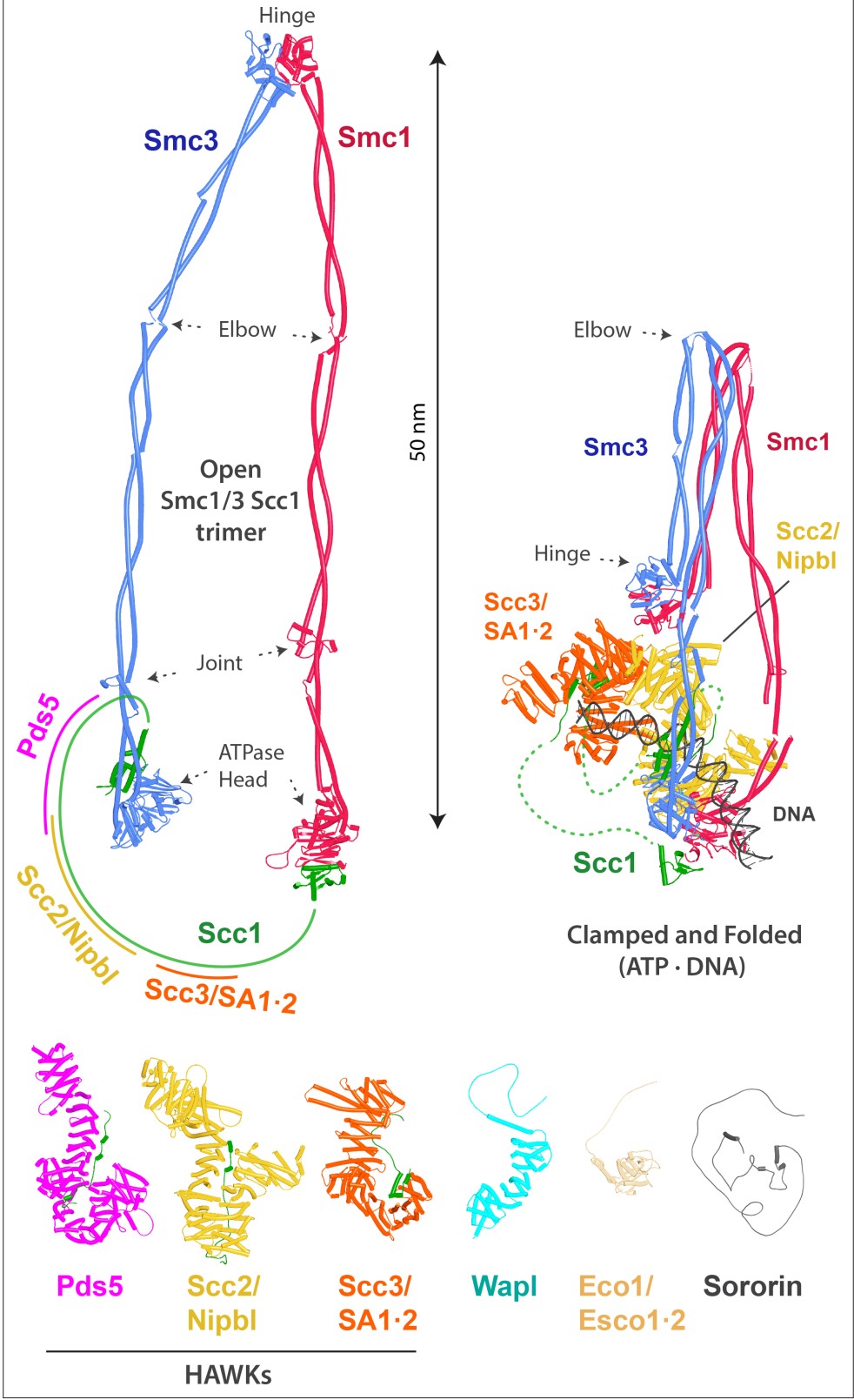

**Figure 1.** Overview of cohesin, its subunits, and regulatory proteins. Top left: cohesin's trimeric Smc1/Smc3/Scc1 ring. The approximate HAWK binding sites on the kleisin Scc1 are indicated. Top right: cohesin in the DNA-clamped and folded state (hybrid model combining PDBs 6WG3, 6ZZ6, and 7OGT). Bottom: cohesin's HAWKs and regulatory proteins (AlphaFold 2 [AF] predictions).

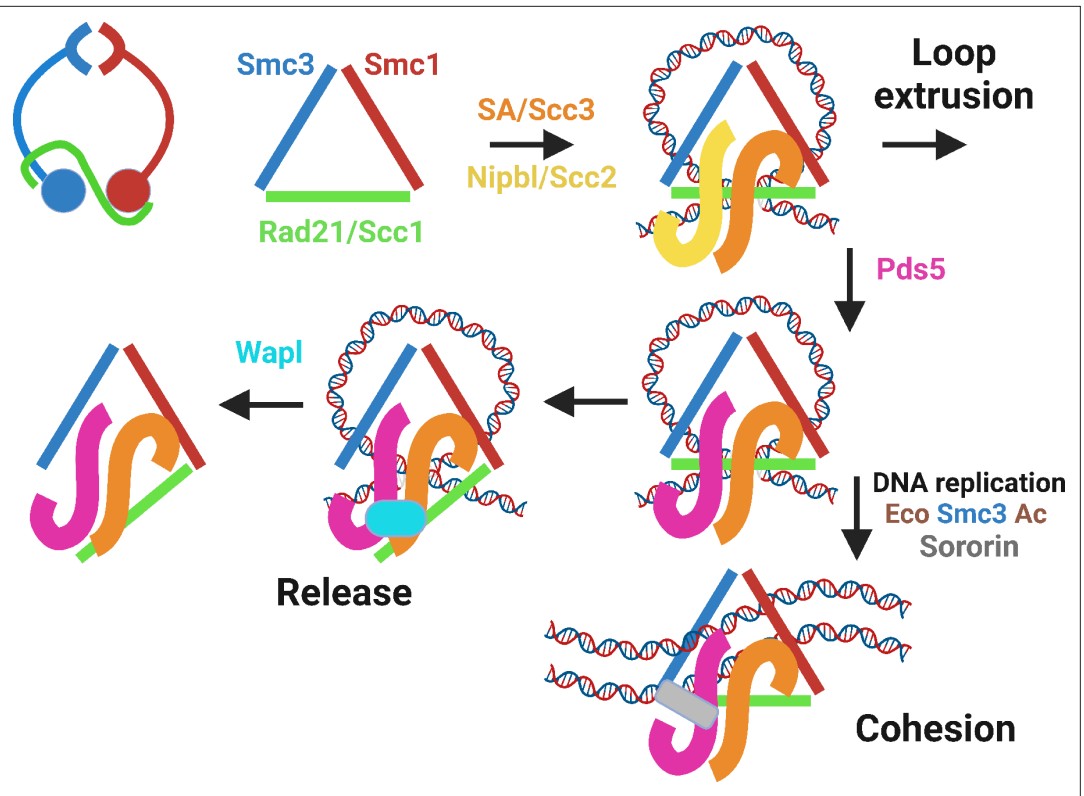

**Figure 2.** Cohesin's trimeric ring (top left), represented throughout the rest of the figure as a triangle, associates with DNA loops when occupied by SA/Scc3 and Nipbl/Scc2. Nipbl/Scc2 stimulates ATP hydrolysis, leading to DNA translocation and loop extrusion. This is halted by Nipbl's replacement by Pds5, creating a complex that can bind Wapl and detach from chromatin, in a process that depends on a pair of lysine residues (yeast: K112 and K113; humans: K105 and K106) on Smc3's ATPase head, whose acetylation by Eco1 (yeast) or Esco1/2 in vertebrates blocks release. Acetylated complexes containing Pds5 either form stable loops between CTCF sites or, during DNA replication, generate cohesion between sister DNAs, whose maintenance (in animal and plant cells) requires Sororin (as well as continued acetylation) to prevent Wapl-mediated release.

abrogates its cohesion protection and Wapl thereupon destroys most cohesion along chromosome arms but not at centromeres, where PP2A associated with Shugoshins counteracts Cdk1 (*Liu et al., 2013*). Wapl's destruction of all cohesion along chromosome arms when cells are arrested in mitosis by spindle poisons is responsible for the canonical image of mitotic chromosomes, with cohesion persisting solely at centromeres.

Cohesion is thought to be mediated through the topological entrapment of sister DNAs by individual tripartite Smc-kleisin rings (*Haering et al., 2008*; *Srinivasan et al., 2018*), either through co-entrapment of sister DNAs inside a single ring, or alternatively through formation of a Borromean ring that links sister DNAs (best thought of as infinitively large rings) and cohesin rings in a topology that does not involve catenation of any one by the other (*Collier et al., 2020*). In either case, ring opening through Scc1's cleavage by Separase destroys the topological entanglement, thereby destroying cohesion and triggering the metaphase to anaphase transition (*Oliveira et al., 2010*; *Uhlmann et al., 2000*). Unlike Separase, Wapl removes cohesin and destroys cohesion by opening the cohesin ring in a manner that does not involve proteolysis. Importantly, the finding that Wapl-mediated release is blocked by fusing Smc3's C-terminus to Scc1's N-terminus suggests that release depends on dissociation of Scc1's NTD from Smc3 (*Beckouët et al., 2016*; *Chan et al., 2012*), thereby creating a gate through which DNAs may pass. If sisters are held together by their co-entrapment within a cohesin ring, then the latter must also possess an entry gate that differs from the exit gate and probably resides at the hinge interface (*Collier and Nasmyth, 2022*), because, unlike release, fusion of Smc3 to Scc1 does not block cohesion establishment (*Gruber et al., 2006*; *Srinivasan et al., 2018*). Less is known about the topology of cohesin's association with DNA during LE. The observation that it is

unaffected, at least in vitro, by fusing all three of the ring's interfaces has led to the suggestion that LE does not require (single) passage of DNA through any of the ring's interfaces (*Davidson et al., 2019*), but this does not exclude the possibility of DNAs passing through cohesin rings in two directions (double passage).

X-ray crystallography and electron cryomicroscopy (cryo-EM) have revealed many details of the topology of Smc1/Smc3 heterodimers (*Figure 1*), how they dimerize via hinge domains (*Haering et al., 2002*), how their coiled coils are folded around an elbow (*Bürmann et al., 2019*), how their ATPase domains associate in the presence of ATP and bind the N- and C-terminal domains of Scc1 (*Gligoris et al., 2014*; *Haering et al., 2004*), the structure of cohesin's HAWK subunits (*Hara et al., 2014*; *Kikuchi et al., 2016*; *Lee et al., 2016*; *Ouyang et al., 2016*; *Roig et al., 2014*) and how they bind DNA (*Li et al., 2018*), how Scc2/Nipbl clamps DNA on top of engaged ATPase heads (*Collier et al., 2020*; *Higashi et al., 2020*; *Shi et al., 2020*), and lastly the structure of Wapl (*Chatterjee et al., 2013*; *Ouyang et al., 2013*). Despite this progress, many molecular aspects of cohesin's activities remain mysterious. These include how cohesin actually extrudes DNA loops, how Scc2/Nipbl and Pds5 exchange with each other, how cohesin's exit and entry gates are opened, how Wapl facilitates exit gate opening and thereby induces cohesin's dissociation from chromosomes, how Esco/Eco acetyl transferases are recruited to cohesin, how they recognize Smc3, and how this is facilitated by Pds5 and the Mcm helicase, and how Sororin hinders exit gate opening and thereby thwarts Wapl.

We report here a complementary approach to in vitro biochemistry and experimental structural biology, namely in silico 3D protein structure prediction by AlphaFold 2 (AF) (*Evans et al., 2021*; *Jumper et al., 2021*), based on amino acid co-evolution analyses and deep learning approaches to sequence/structure relationships, to predict structurally hitherto unreported interactions between cohesin subunits. When performed with subunits from multiple eukaryotic lineages and critically judged in the light of amino acid conservation (beyond that used by AF), these predictions have furnished insights into several of the outstanding questions posed above. It has revealed a possible role in opening cohesin's exit gate of interactions between the Smc1 and Smc3 coiled coils in the vicinity of their joints when ATPases engage in the absence of Scc2/Nipbl. AF also predicts how Eco/Esco acetyl transferases are recruited to cohesin and thereby bind to K112 and K113 (human: K105 and K106) within Smc3's ATPase domain. Crucially, it predicts that Wapl induces formation of a tripartite complex between itself, SA/Scc3, and Pds5. If this occurs after Scc1's NTD has already dissociated from Smc3, then the manner in which Scc1 associates with Pds5 places the kleisin's NTD close to a highly conserved cleft within Wapl's CTD, raising the possibility that the latter acts by sequestering Scc1's N-terminal 20 amino acids. Importantly, the tripartite SA/Wapl/Pds5 complex is predicted to form not only in animal but also in plant and fungal lineages, suggesting that it has deep evolutionary origins. Likewise, the prediction that Sororin's essential C-terminal domain binds to the interface between Smc3 and Scc1's NTD provides a mechanism by which it protects the latter from Wapl.

## Results

### Closing the Smc1 and Smc3 coiled coil joints plays a part in the dissociation of Scc1's NTD from Smc3's neck

Wapl is thought to induce cohesin's release from chromosomes by facilitating the passage of DNAs through the Smc3/Scc1 interface. It could do this either by inducing dissociation of the interface, by holding the gate open once Scc1 has already dissociated, or by a combination of both mechanisms. If it merely held the gate open, then there must exist a mechanism capable of opening it that is Wapl-independent. Evidence for this comes from the observation that inactivation of Scc2 during late G1 triggers cohesin's dissociation from chromosomes via a mechanism that is blocked by fusion of Smc3 to Scc1 or by mutations such as *SMC1(L1129V)* or *SMC1(D1164E)* that suppress *eco1Δ* lethality (*Srinivasan et al., 2019*). Crucially, this release is independent of Wapl.

A clue as to a mechanism for Wapl-independent release comes from observations on the related condensin complex. In the absence of DNA, ATP induces dissociation from Smc2 (equivalent to Smc3) of Brn1's NTD (equivalent to Scc1). Because it also occurs in the presence of ADP and BeF$_3$, Brn1's dissociation is presumably induced by the engagement of Smc2 and Smc4 ATPase heads (*Hassler et al., 2019*; *Lee et al., 2022*). This raises two related questions. Does a similar process occur in cohesin, and if so, how might head engagement induce dissociation? An important clue as to the

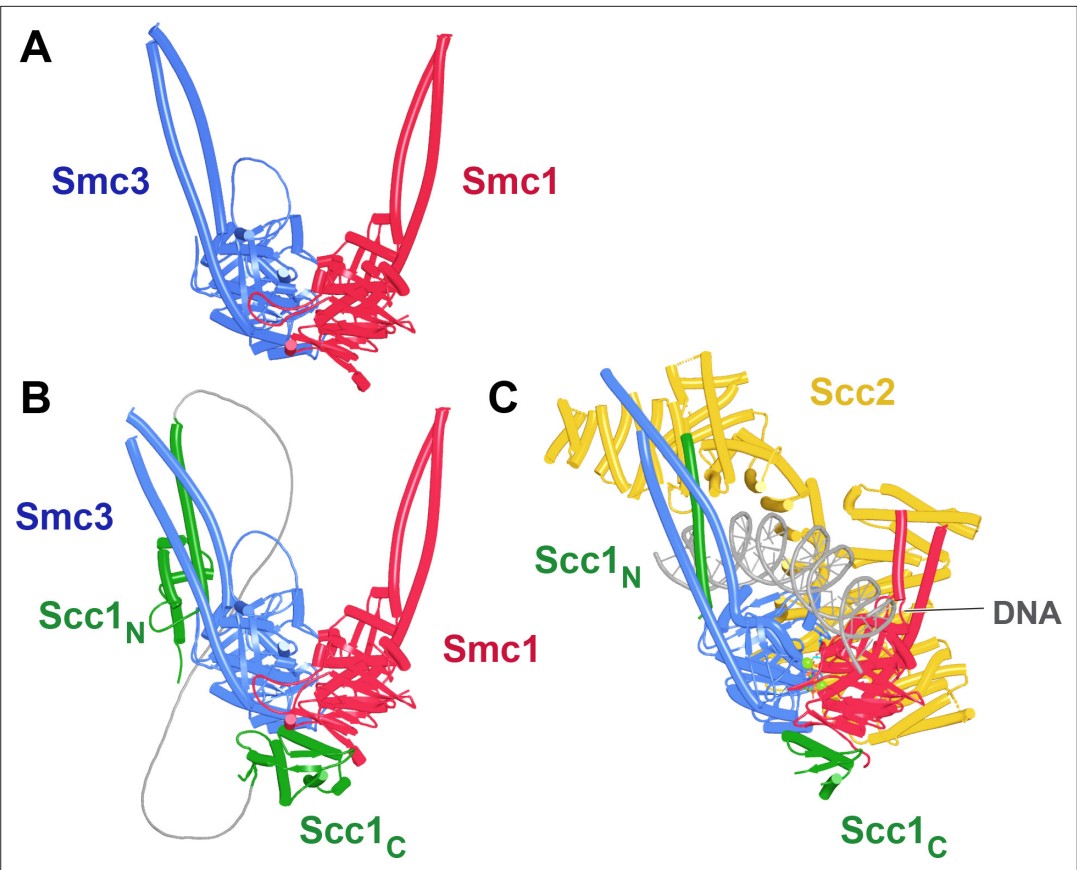

**Figure 3.** Enagagement of Smc1 and Smc3 heads in the absence and presence of Scc1 and Scc2. (**A**) AlphaFold 2 multimer (AF) prediction of yeast Smc1 and Smc3 ATPase heads with associated coiled coils up to, but not including their joints. The actual predictions and sequences used are available in folder f1 at https://doi.org/10.6084%20m9.figshare.22567318.v2. (**B**) As (**A**), but including Scc1's N- and CTD connected by a GS linker (f2). (**C**) Cryo-EM structure of DNA clamped on top of engaged Smc1 and Smc3 heads by Scc2 (PDB 6ZZ6).

possible mechanism stems from cryo-EM structures of both cohesin and condensin in the presence of DNA and non-hydrolyzable ATP analogues, which reveal that DNA is 'clamped' on top of engaged ATPase heads by Scc2 and Ycs4, respectively. Importantly, in both cases, Smc coiled coils are splayed open and the N-terminal domains of Brn1 and Scc1 remain associated with the necks of Smc2 and Smc3, respectively (*Collier et al., 2020*; *Lee et al., 2022*; *Shaltiel et al., 2022*). This raises the question as to why DNA's presence leads to retention of the Smc-kleisin interaction. Is it due to DNA binding per se or to an event that accompanies that binding, namely opening of the Smc coiled coils. Clearly, head engagement does not per se induce dissociation, at least when DNA is bound on top of them, as in the clamped cryo-EM structures.

To address these questions, we explored the potential of structural predictions using AF multimer (*Evans et al., 2021*; *Jumper et al., 2021*). Importantly, presumably due to the co-evolution of amino acids within the interface of engaged Smc ATPase heads, AF routinely attempts to engage them even though ATP is absent, which means that it is possible, to some extent, to observe the effects of head engagement on the structure of cohesin's coiled coils using AF (*Figure 3A*, f1). As with all further figures, the data files (f1–f98), including atomic coordinates and AF output files, can be found at https://doi.org/10.6084/m9.figshare.22567318.v2. To address whether head engagement is compatible with association of the N- and C-terminal domains of Scc1 with Smc3 and Smc1 ATPase heads, respectively, we asked AF to predict the structures of Smc1 and Smc3 heads associated with coiled coils extending up to but not including their joints in the presence of a truncated Scc1 polypeptide in which its conserved NTD and CTDs were joined by a GS (glycine, serine) linker (*Figure 3B*, f2). AF predicted with high confidence in five out of five models heads whose engagement was very similar to that of the clamped cryo-EM structures (*Figure 3C*), as was the orientation of their coiled coils.

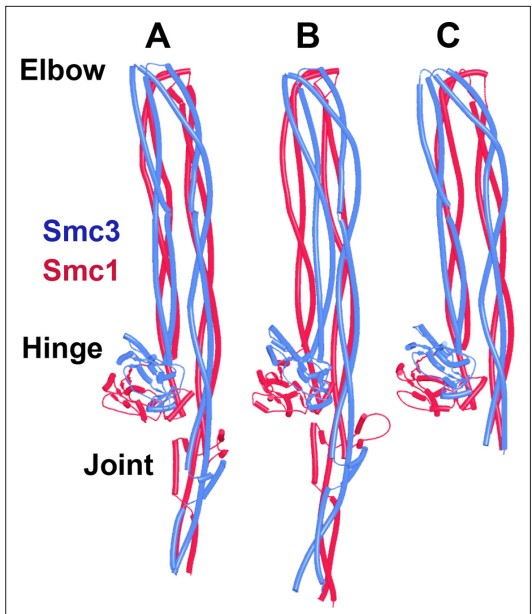

**Figure 4.** Folding of Smc1 and Smc3 coiled coils is accompanied by their juxtaposition. (**A**) AlphaFold 2 (AF) prediction for yeast Smc1 and Smc3 lacking their ATPase domains (f3). (**B**) As (**A**), but human Smc1 and Smc3 (f4). (**C**) Cryo-EM structure of yeast Smc1 and Smc3 hinge and associated coiled coils, PDB 7OGT.

Importantly, the N- and C-terminal domains of Scc1 were invariably bound to the neck of Smc3 and the base of Smc1, respectively, as observed in the clamped structures. As shown by cryo-EM and predicted by AF, head engagement is therefore not per se incompatible with Scc1's association with Smc3.

A very different arrangement of Smc1 and Smc3 coiled coils has been observed by cryo-EM in the absence of ATP. The coiled coils are juxtaposed from their emergence from the hinge all the way to their joints close to the ATPase domains, which is accompanied by folding of both coiled coils around their elbows, thereby locating the hinge domain close to a segment of Smc3's coiled coil just above its joint (*Petela et al., 2021*; PDB 7OGT). To address whether AF can reproduce this state, it was used to predict the structure of a yeast Smc1/Smc3 heterodimer containing all hinge and coiled coil sequences, but lacking both ATPase domains (*Figure 4A*, f3). This revealed, with high confidence in five out of five models, a structure resembling that seen by cryo-EM (*Figure 4C*), namely, with the Smc1 and Smc3 coiled coils juxtaposed in parallel throughout their length and folded around their elbows placing the hinge asymmetrically on top of the Smc3's coiled coil, just above its joint. Noticeably, in the AF prediction, the latter was intimately inter-digitated with Smc1's joint, creating a compact junction between the two (*Figure 4A*). AF predicted an almost identical structure for the equivalent parts of human Smc1/Smc3 heterodimers (*Figure 4B*, f4). Importantly, the asymmetric manner in which the Smc1 and Smc3 coiled coils emerge from the hinge and thereby associate with each other, as soon as they do so, resembles that seen by cryo-EM as well as a crystal structure of the condensin hinge associated with stretches of coiled coil (*Lee et al., 2020*; *Soh et al., 2015*; PDB 4RSI, 6YVU). This relationship between

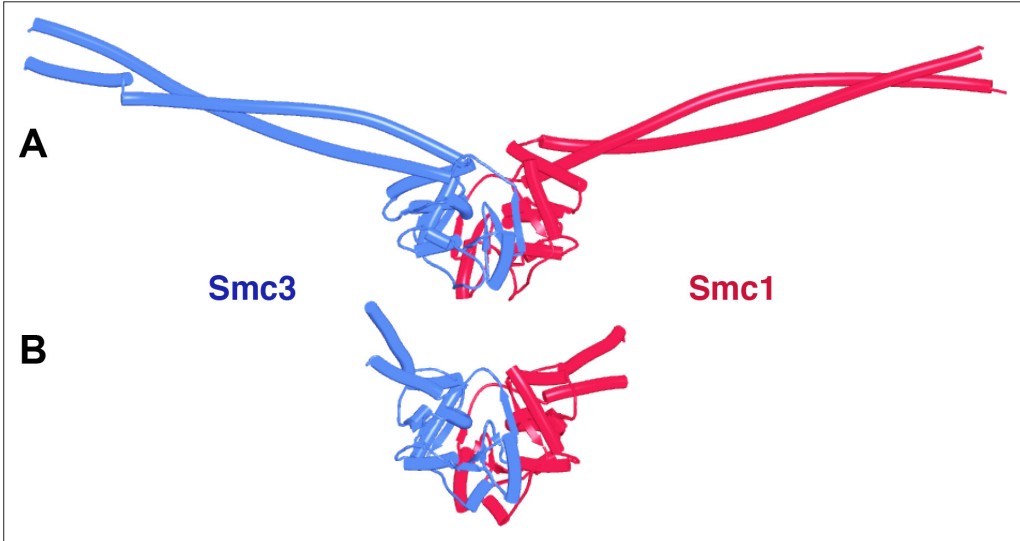

**Figure 5.** Juxtaposition of Smc1/3 coiled coils requires segments below the elbow. (**A**) AlphaFold 2 (AF) prediction for yeast Smc1 and Smc3 hinge domains with short associated coiled coils (f5). (**B**) Crystal structure of *T. maritima* hinge associated with short coiled coils, PDB 1GXL.

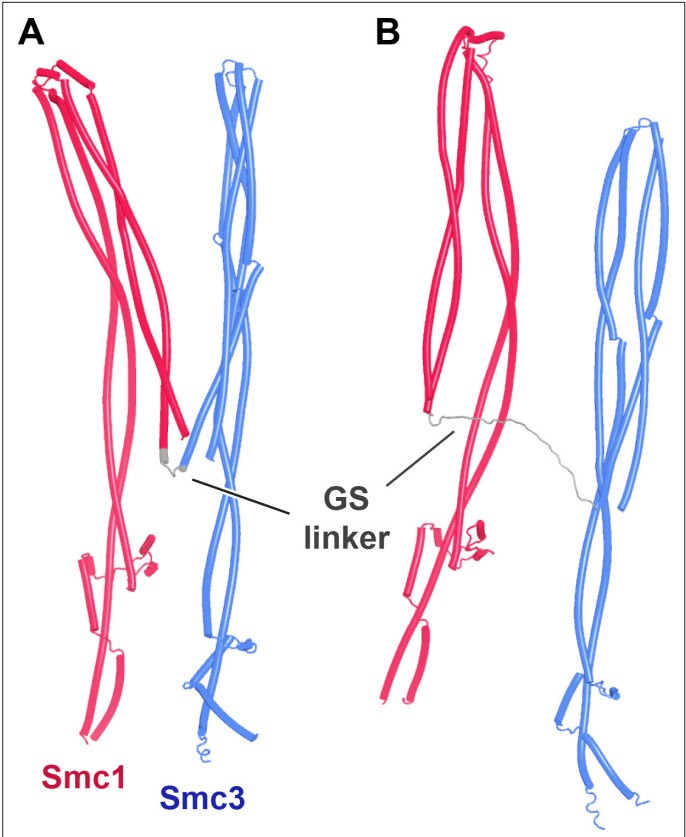

**Figure 6.** AlphaFold 2 (AF) predictions for yeast Smc1 and Smc3 coiled coils including their elbow and joints, but joined by GSx4 (**A**, f6) and GSx15 (**B**, f7) linkers instead of their hinges.

the hinge and its associated coils has the appearance of a wrist, folded back on its associated arm. Interestingly, according to AF, the folded wrist conformation is not predicted when the coiled coils associated with the hinge were cut short at the elbow (*Figure 5A* , f5). The shortened Smc1 and Smc3 coiled coils instead emerge symmetrically with opened arms at an angle of 120° from each other, in a manner resembling the crystal structure of the *Thermotoga maritima* hinge associated with short coiled coils (*Figure 5B*; *Haering et al., 2002*; PDB 1GXL). This suggests that folding at the elbow as well as the consequent packing of the hinge on top of the Smc3 coiled coil may have an important role in organizing how the coiled coils emerge from the hinge. Crucially, the association of full-length elbow-containing coiled coils and especially their joint junction depends on the hinge, as Smc1 and Smc3 coiled coils in which the hinge is replaced by either a GSx4 (*Figure 6A*, f6) or GSx15 (*Figure 6B*, f7) linker fail to associate in the predictions, even at their joints. Though the elbow appears to be a feature that is autonomous, at least according to AF, coiled coil association, including joint junction, is affected by a distant 'organizing locus,' namely the hinge in its wrist conformation.

Though influenced by elbow and hinge, the joints nevertheless have an affinity for each other, as they form compact junctions when the coiled coils distal to the elbow are excluded and do so despite the lack of any extensive association between coiled coils away from the joint (*Figure 7A*, f8). Indeed, AF predicted with high confidence a compact heterodimer formed between the joints of yeast Smc1 and Smc3, associated with only short sections of coiled coils (*Figure 7B*, f9). A very similar prediction, albeit in only the top ranked model, was made for the equivalent human proteins (f10). In both yeast and humans, the predicted interactions between isolated joints were very similar to those within Smc1/Smc3 heterodimers lacking merely their ATPase domains (*Figure 4*).

It is important to note that the junction between Smc1 and Smc3 joints was fully compatible with the simultaneous association of Scc2 with the Smc3 joint. This was demonstrated by running an AF prediction of Scc2's NTD either with an isolated Smc3 joint segment or with an Smc1/Smc3 joint dimer (*Figure 8A*, f12). Aligning the Smc3 moiety of the AF prediction with the equivalent residues in the

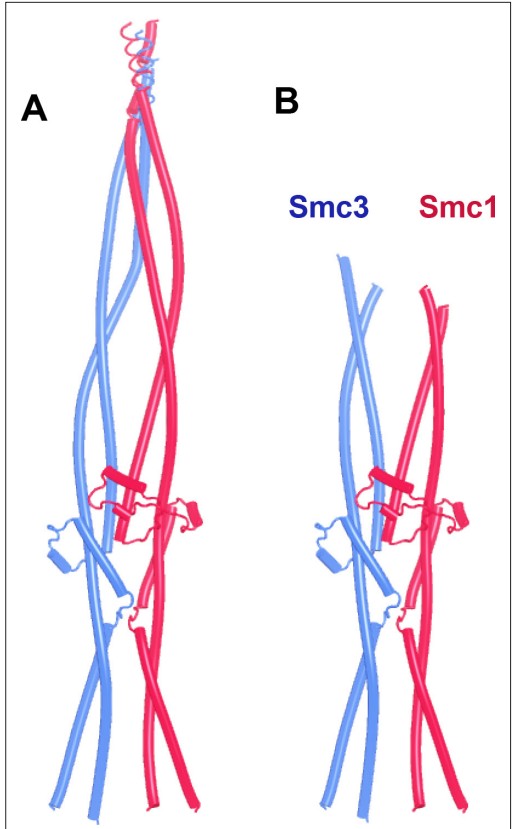

**Figure 7.** AlphaFold 2 (AF) predictions of Smc1 and Smc3 joints with short associated coiled coils. (**A**) (f8) and (**B**) (f9), yeast.

cryo-EM structure of DNAs clamped by Scc2 on top of engaged Smc1/3 heads (*Figure 8B*; PDB 6ZZ6) shows the prediction to recapitulate the same Scc2:Smc3 binding mode. Because of this compatibility, competition with Scc2 is unlikely therefore to have a role in joint disjunction (and thereby retention of Scc1's association with Smc3) upon clamping.

These predictions imply that dimerization at opposite ends of the Smc1 and Smc3 coiled coils has opposing influences on their organization. Engagement of the ATPase heads opens their associated coiled coils, while hinge dimerization has the opposite effect, forcing them together, including junction of their joints proximal to the ATPase heads. What happens then when both types of dimerization co-exist? Due to their size, AF predictions of full-length Smc1/Smc3 heterodimers are close to limits of what is possible for current implementations of AF multimer (*Evans et al., 2021*). Despite this, we obtained four models for yeast and five for human heterodimers. Four out of four models for the yeast complex predicted heterodimers whose coiled coils were folded and their joints interdigitated despite heads being fully engaged (*Figure 9A and B*, f13). In the human case, three out of five models (ranks 1, 2, and 5) had a similar configuration but in one model (rank 3) joint junction was dominant over head engagement (and the latter were disengaged), while in another head engagement was dominant over joint junction and the latter were disjoined (*Figure 9C*, f14). However,

as one might expect, coexistence of joint junction and head engagement is accompanied by distortions of the coiled coils connecting joints and heads, and in some predictions this is associated with extreme bending and/or actual unfolding of the alpha helices. The more extreme coiled coil distortions observed in some models of the full-length Smc1/Smc3 heterodimers models are absent when the hinge and distal coiled coils are removed from the yeast complex. Thus, four out of five (ranks 1–4) models of Smc1/3 ATPase heads associated with coiled coils extending to but not beyond the elbow predict with high-confidence junction of their joints co-existing with fully engaged heads (*Figure 10A*, f15).

Though some of the more baroque AF models may reflect unrealistic scenarios, the emerging principle that head engagement and joint junction can co-exist but only do so if the intervening coiled coils are distorted is consistent with a similar configuration in low-resolution cryo-EM structures of Smc1/Smc3 heterodimers in the presence of ATP (*Petela et al., 2021*). Indeed, there is evidence that the state also occurs in vivo. By introducing cysteine pairs in the interface between engaged Smc1/3 ATPases, it is possible to use the bifunctional thiol-specific crosslinking reagent BMOE (bis-maleimidoethane) to detect their proximity in vivo. A different pair of cysteines can be used to detect association between Smc1 and Smc3 coiled coils in the vicinity of their joints. In an observation surprising at the time, it was observed that BMOE induces appreciable double crosslinking (head and joint crosslinks) with a frequency not dissimilar to the product of the frequency of the individual pairs, which indicates simultaneous crosslinking (*Chapard et al., 2019*).

These observations lead us to propose that head engagement under conditions in which the hinge reinforces a junction of Smc1/Smc3 joints, namely in the absence of DNA or Scc2, will create distortions of the coiled coil (Smc3's neck) to which Scc1's NTD is normally associated and that this distortion

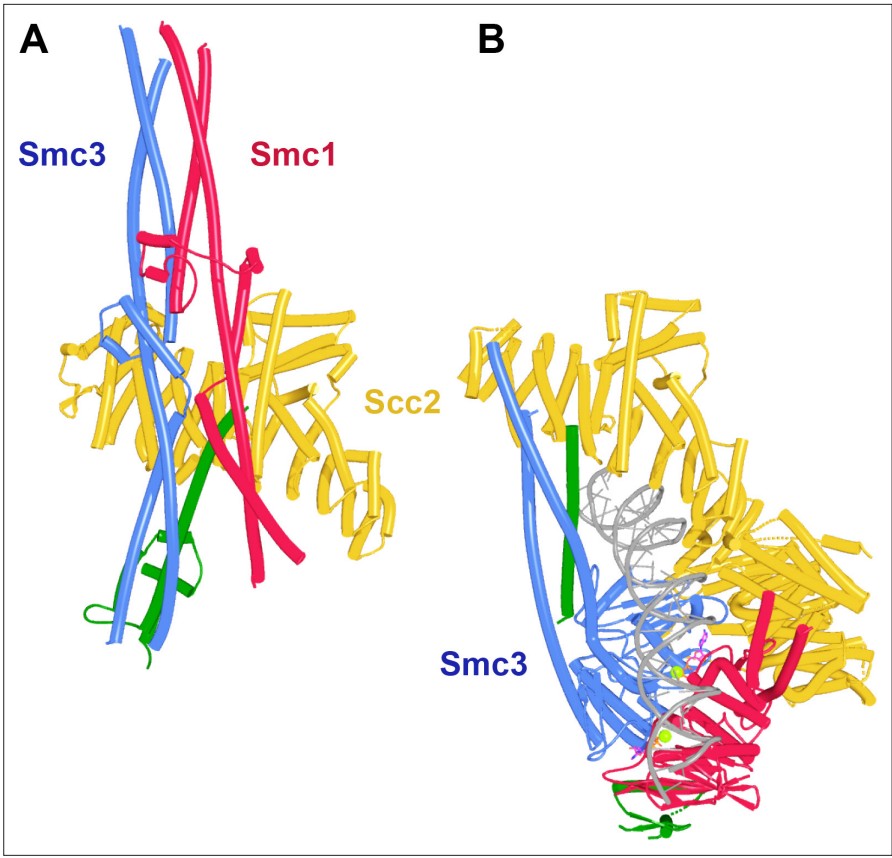

**Figure 8.** Scc2 binding does not per se disrupt junction of Smc1 and Smc3 joints. (**A**) AlphaFold 2 (AF) prediction for yeast Smc1 and Smc3 joints with associated coiled coils together with Scc1 and Scc2's NTDs (f12). Chains in the PAEs from f12 are Scc2 (A), Smc3 (B), Smc3 (C), Scc1 (D), Smc1 (E), and Smc1 (F). (**B**) Cryo-EM structure of DNA clamped by Scc2 on top of engaged Smc1 and Smc3 ATPase, PDB 6ZZ6.

may cause its dissociation. A corollary is that maintaining the association of Scc1's NTD with the Smc3 neck should prevent either joint junction or even head engagement. Consistent with this hypothesis, the addition of Scc1's NTD (*Figure 10B*, f16) but not its CTD (*Figure 10C*, f17) to the Smc1 and Smc3 ATPase heads associated with coiled coils extending to the elbow eliminated the joint junction previously observed in four out of five models. In this case, where joint junction is not reinforced by the hinge, AF clearly gave preference to the Smc3-kleisin interaction over joint junction. Likewise, the top three ranked models obtained with Scc1's NTD together with full-length Smc1/Smc3 heterodimers predicted engaged heads with Scc1's NTD associated with Smc3's neck but joints fully disjoined (f18). Though the in silico prediction that association between Scc1's NTD and the Smc3 neck is dominant over joint junction may not be an accurate reflection of nature, the notion that joint junction while heads are engaged may be the driving force for opening cohesin's exit gate nevertheless provides a plausible mechanism for this key event. Importantly, the hypothesis explains why non-lethal mutations that compromise head engagement (*Elbatsh et al., 2016*) abrogate releasing activity (RA). Crucially, it is eminently testable, as it predicts that dissociation of Scc1's NTD from the Smc3 neck depends on joints and probably also on distal coiled coils and their associated hinge.

## How does Wapl enhance exit gate opening?

If ATP-dependent dissociation of Scc1's NTD from the Smc3 neck is a feature intrinsic to Smc-kleisin trimers, then how might Wapl enhance this process? To address this, we used AF to explore how Wapl is first recruited to cohesin and second what effect it has on the configuration of cohesin's HAWK regulatory proteins and /or on the Smc3/Scc1 interface. Our starting point was existing knowledge based on biochemical and genetic assays. In vitro, Wapl is known to bind SA/Scc3 and this binding is abrogated by mutation of a highly conserved lysine residue (K404 in yeast; K330 in humans) within

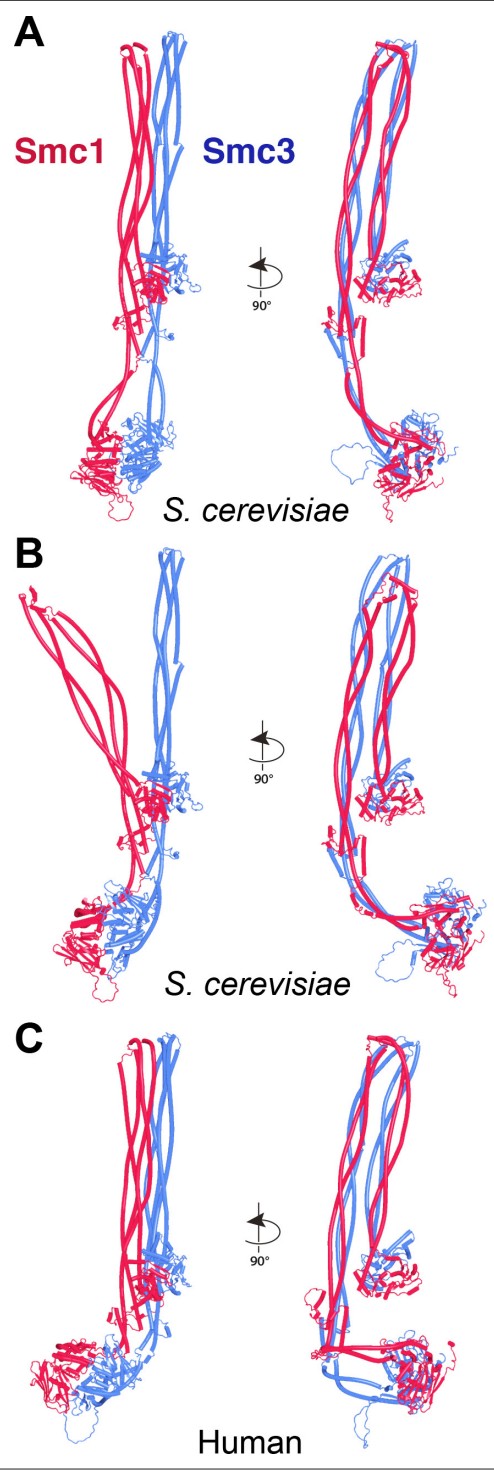

**Figure 9.** AlphaFold 2 (AF) predictions for full-length yeast (**A, B**, f13) and human (**C**, f14) Smc1/Smc3 heterodimers.

a highly conserved surface (known as the 'CES,' conserved and essential surface) that sits underneath the pronounced 'nose' of Scc3/SA. This binding must be of physiological importance as K to E mutations abrogate Wapl-dependent RA in vivo, both in yeast (*Beckouët et al., 2016*) and in cultured human cells (*Hara et al., 2014*). Exactly which part of Wapl binds to the CES has never been fully established. It has been proposed that sequences within Wapl's unstructured N-terminal domain are responsible, either one or other of its FGF motifs (*Shintomi and Hirano, 2009*) or a different segment between residues 510 and 570, known as Wapl-M (*Hara et al., 2014*; *Ouyang et al., 2013*). As will become clear below, AF predictions imply that neither proposal is correct. Meanwhile, a short YSR motif at Wapl's N-terminus binds a highly conserved loop (consensus sequence AP[D/E]AP) between helices 2 and 3 within Pds5's NTD, which is also bound by a YSR-like motif within the kinase Haspin (*Ouyang et al., 2016*). This locus within Pds5's NTD was in fact first identified as being crucial for Wapl-mediated release through mutations clustering at this location that suppress the lethality of *eco1* mutants in yeast (*Rowland et al., 2009*). However, Wapl's N-terminal YSR has never been rigorously demonstrated to be necessary for its RA. Most fungal Wapls also possess at or near their N-termini a similar motif whose consensus is [M/Φ]xxYG[K/R]. Importantly, its deletion in *Saccharomyces cerevisiae* does not suppress the lethality of *eco1Δ* mutants (*Figure 11A*), implying that unlike Pds5's AP[D/E]AP loop to which it is thought to bind, the [M/Φ]xxYG[K/R] motif is not essential for Wapl-mediated RA. Some other sequence must therefore bind the AP[D/E]AP loop in vivo. Interestingly, the [M/Φ]xxYG[K/R] motif is lacking in *Schizosaccharomyces pombe*, though a different one (TYxxxR[T/S]ΦL), which is much closer to Wapl's conserved CTD, is present and highly conserved in that location among all fungi (*Figure 11B*). Unlike the N-terminal motif, the more C-terminal TYxxxR[T/S]ΦL motif is essential for RA, as its deletion suppresses the lethality of *eco1Δ* mutants in *S. cerevisiae* (*Figure 11A*). Thus, the picture that emerges is primarily one of a loose association between Wapl and SA/Scc3 or Pds5 via short motifs within Wapl's unstructured N-terminal domain. Crucially, none of these interactions help explain how Wapl's more conserved CTD composed of eight HEAT repeats actually promotes release.

We first used AF to explore the association between Wapl and Pds5. For yeast, this predicted a single point of contact between a short Wapl motif ([F/W]xFLD), which is conserved in numerous

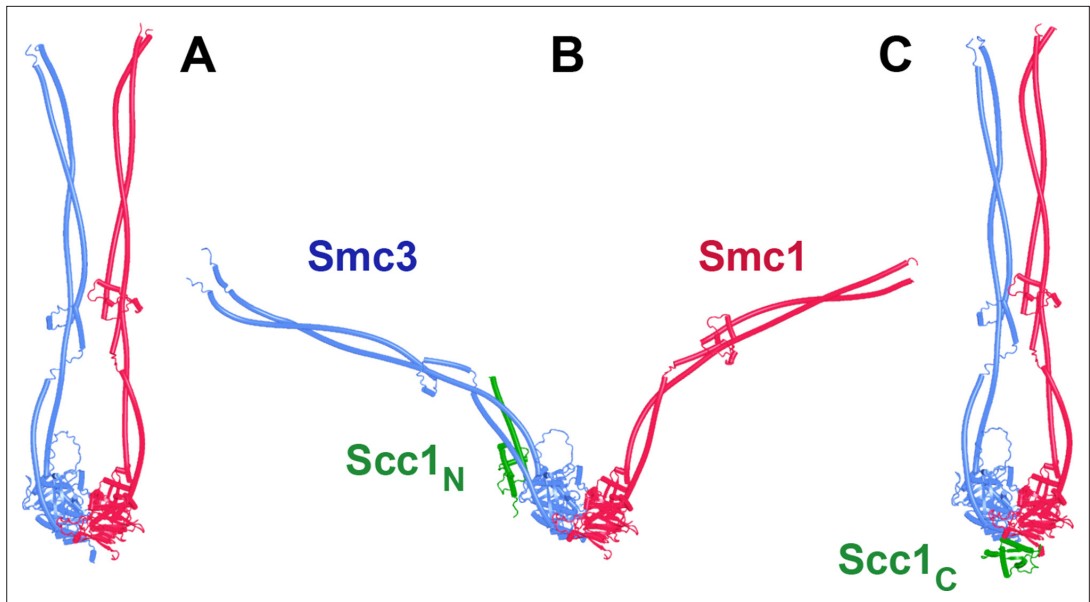

**Figure 10.** AlphaFold 2 (AF) predictions of yeast Smc1 and Smc3 ATPases with coiled coils extending to the elbow but not beyond, without Scc1 (**A**, f15), with Scc1's NTD (**B**, f16), and with Scc1's CTD (**C**, f17). Chains in the PAEs from f16 are Scc1 (A), Smc1 (B), Smc1 (C), Smc3 (D), and Smc3 (E).

ascomycetes including *S. cerevisiae* (FNFLD; see *Figure 11A*) and *S. pombe* (WNFLD), and a trough situated between a pair of HEAT repeats located within Pds5's C-terminal half, at the point where its longitudinal axis starts to curve (*Figure 12A*, f19). Because the very same trough is predicted also to bind FGF motifs from human Wapl (*Figure 13B*, f20) and Sororin (*Figure 13B*, f21), as well as a separate motif within Esco1 (*Figure 13B*, f22), we will refer to it as the Wapl Esco Sororin Trough or 'WEST.' Intriguingly, AF models of individual proteins in the AlphaFold Protein Structure Database (*Varadi et al., 2022*) predict that the *S. cerevisiae* and *S. pombe* FNFLD motifs of Wapl also bind a highly conserved cleft within Wapl itself, at the C-terminal end of its CTD (*Figure 12B*). However, given the choice between binding Pds5 and itself, AF chooses the former. In humans, as already alluded to, AF predicted in five out of five models an interaction between the central of three FGF motifs within Wapl's N-terminal extension and the equivalent C-terminal trough within Pds5A and B (*Figure 13B*, f20). In contrast, AF predicted association between Wapl's N-terminal YSR motif and Pds5's N-terminal APDAP loop in just one out of five models. In the case of human (*Figure 14*, f23), but not yeast Wapl (*Figure 12A*, f19), AF predicted with high confidence in five out of five models a rigid body association between its HEAT-repeat containing CTD and HEAT repeats at Pds5's N-terminus, close to but independent of Pds5's APDAP loop, an interaction that proved pivotal for our subsequent analyses (see below).

Though AF did not predict interaction of the *S. cerevisiae* TYxxxR[T/S]ΦL motif with full-length Pds5 (f19A), it predicted, in five out of five models, interaction of the equivalent motif from *Neurospora crassa* with Pds5's APDAP loop (*Figure 12C*, f19B). Subsequent predictions using truncated versions of Pds5 and Wapl revealed that the *S. cerevisiae* TYxxxR[T/S]ΦL motif also reproducibly bound to this site (f19C), as did (using a different Wapl truncation) its [M/Φ]xxYG[K/R] motif (f19D), in this case in a manner resembling a crystal structure of the equivalent complex from humans (PDB: 5HDT) (*Ouyang et al., 2016*). Given that both Pds5's APDAP loop and Wapl's TYxxxR[T/S]ΦL motif are essential for RA in yeast, we presume that the interaction predicted by AF is crucial for their function, a feature whose importance will become apparent later. It is reasonable to suppose that the Pds5 APDAP loop also binds Wapl's N-terminal [M/Φ]xxYG[K/R] motif, albeit not at the stage when the releasing reaction requires the APDAP loop's interaction with the TYxxxR[T/S]ΦL motif.

In contrast to yeast Pds5, AF did predict with high confidence in five out of five models a rigid body contact between a complex between yeast Scc3 and Scc1, and the first (α1) and second (α2) α helices within Wapl's CTD (*Figure 15A*, f24), which are highly conserved (*Figure 11*). The first of these binds across Scc3's HEAT repeats, including the helices at the base of its nose and those associated with

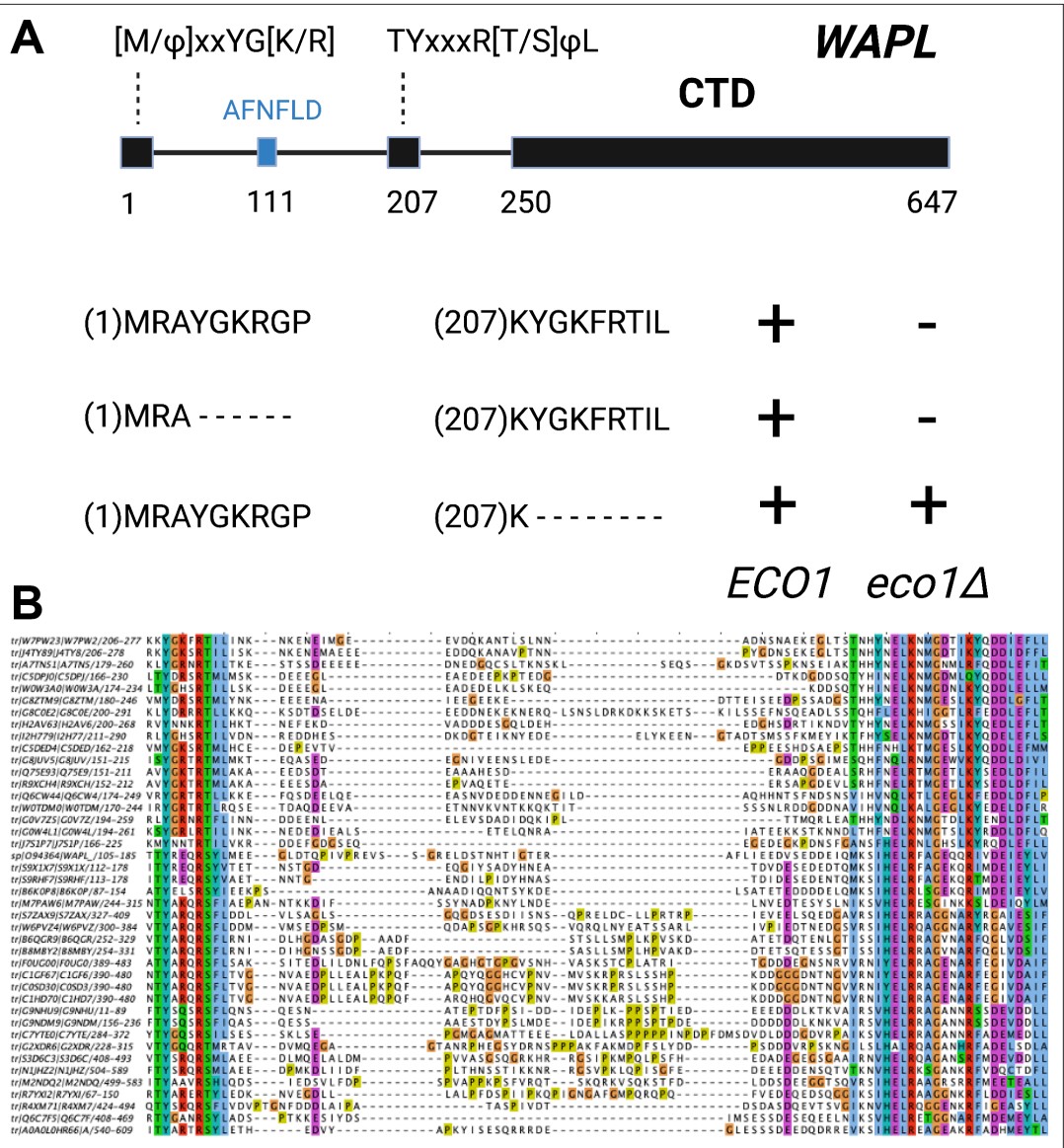

**Figure 11.** The role and conservation of a short motif within the N-terminal region of fungal Wapl proteins. (**A**) Deletion of Wapl's N-terminal [M/Φ]xxYG[K/R] motif from *S. cerevisiae* (waplΔ4–9) does not suppress the lethality of eco1 mutations while deletion of its more C-terminal TYxxxR[T/S]ΦL motif (waplΔ208–215) does so. Both deletions were initially created within a version of the WAPL gene tagged with EGFP (KN18714) and their ability to suppress the lethality of eco1Δ determined by crossing to a waplΔ eco1Δ strain (strain KN16432) followed by tetrad analysis. (**B**) A multiple amino acid sequence alignment of fungal Wapl sequences, showing the TYxxxR[T/S]ΦL motif on the left and the first highly conserved alpha helix within Wapl's CTD on the right. Yeast Wapl is the top sequence (W7PW2).

the CES. Though the Wapl helix runs parallel to, and contacts, the Scc1 polypeptide, exclusion of the latter made little or no difference to AF predictions. Since the axis of Wapl's first helix is nearly orthogonal to those of Scc3's HEAT repeats, so too are most of the HEAT repeat helices within Wapl's CTD and these therefore rise above Scc3 like a 'tower' that follows the line of Scc3's nose, albeit leaning away from the latter's axis by 30°. Toward the top of this tower is a hydrophobic cleft (*Chatterjee et al., 2013*), lined by the most conserved α helix within fungal Wapls. This is the same cleft that in the absence of other proteins is occupied by Wapl's own N-terminal F/WxFLD motif (*Figure 12B*), not only in fungi but also in a wide variety of eukaryotes (see below).

A very similar if not identical interaction was predicted between the human SA2:Scc1 complex and a fragment of human Wapl containing its entire N-terminal extension as well as half of its CTD

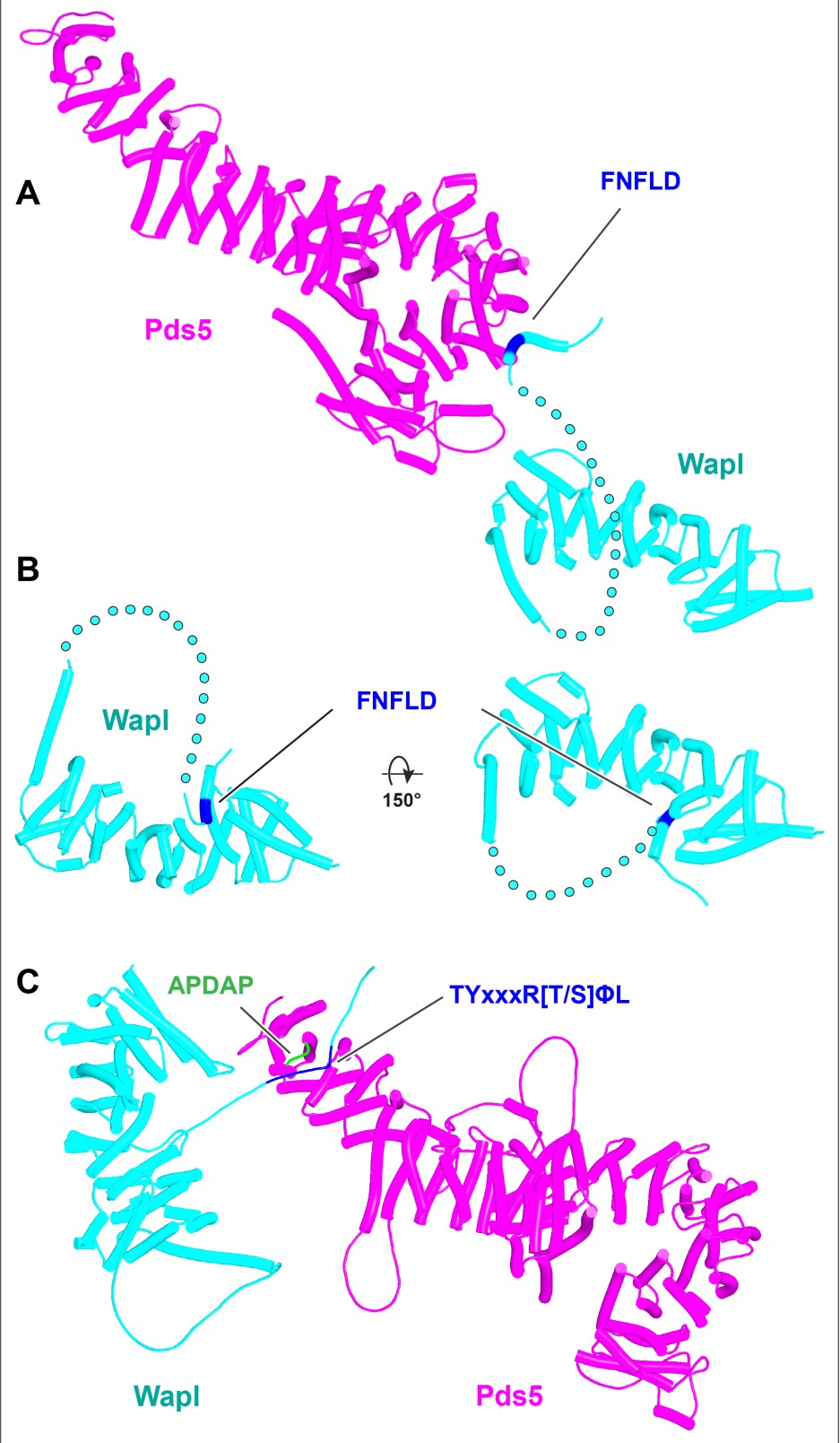

**Figure 12.** Interactions between motifs within Wapl's N-terminal sequences and Pds5. (**A**) AlphaFold 2 (AF) prediction for yeast Wapl's association with Pds5 (f19A). (**B**) AF prediction of yeast Wapl alone (AF-Q99359-F1, AlphaFold Protein Structure Database). (**C**) AF predicts interaction between Wapl's TYxxxR[T/S]ΦL motif and Pds5's APDAP loop in *Neurospora crassa* (f19B).

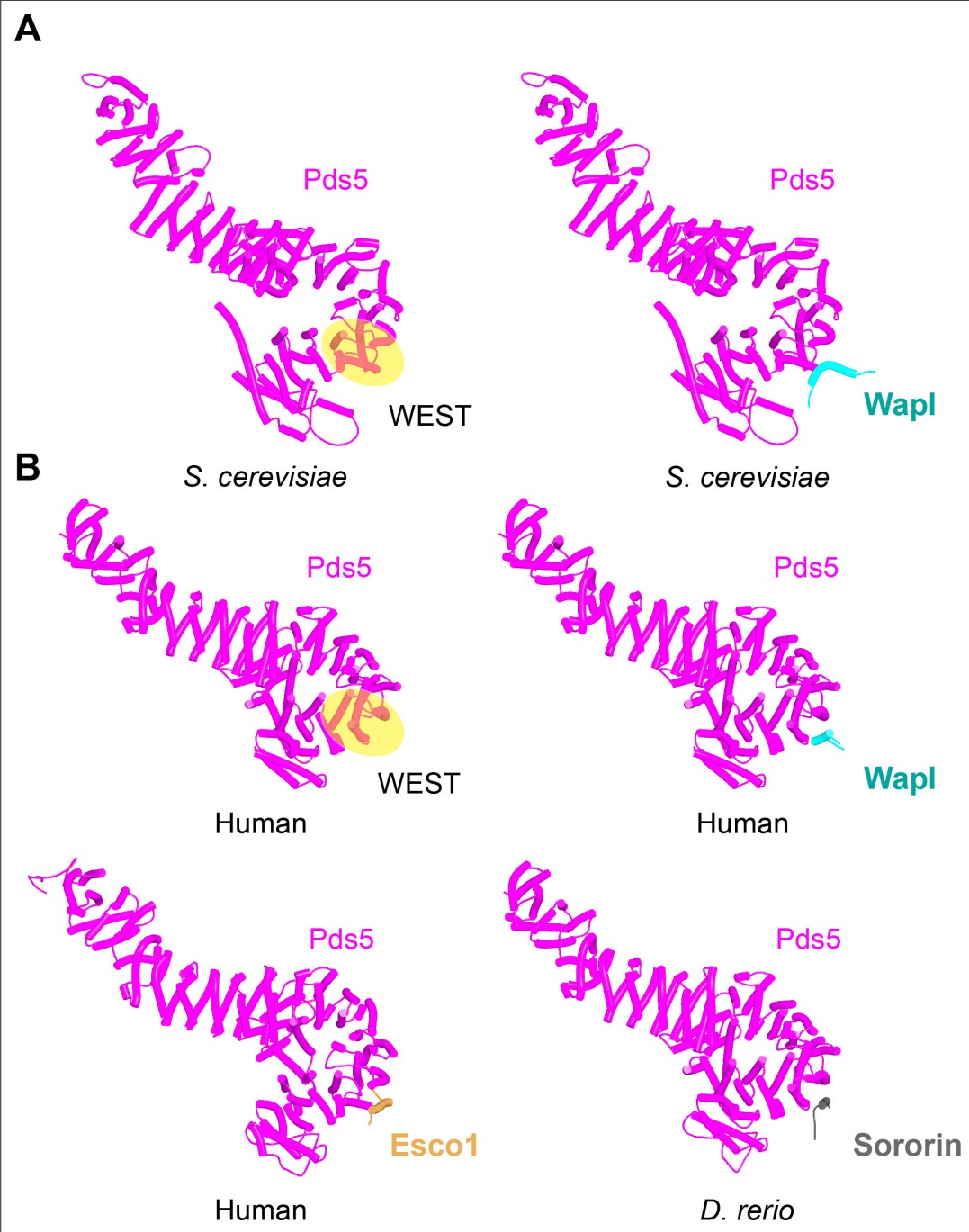

**Figure 13.** Interaction between a trough in Pds5's CTD called 'WEST' and an FNFLD motif within the N-terminal part of yeast Wapl (**A**, left and right, f19), FGF motifs within human Wapl (**B**, upper left and right, f20) and *D. rerio* Sororin (**B**, lower right, f21), and a highly conserved, partly helical, ILxLCEEIAGEIESD motif within human Esco1 (**B**, lower left, f22).

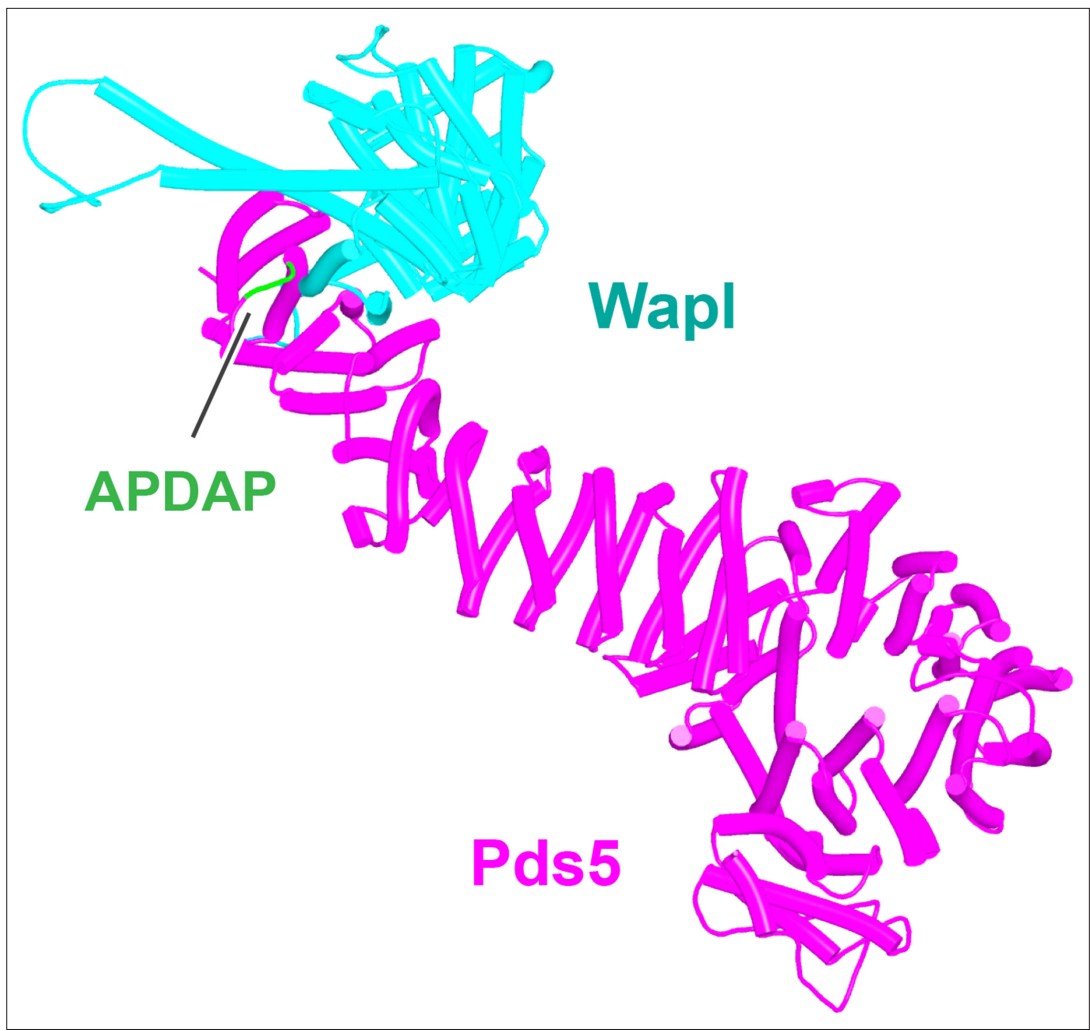

**Figure 14.** AlphaFold 2 (AF) prediction for the interaction between Pds5A and the N-terminal half of Wapl's highly conserved CTD (f23). The same interaction was observed with both Pds5A and B.

(a prediction with full-length SA2 and human Wapl would have been close or over AF's size limit) (*Figure 15B* , f25). In addition, a long motif (IFSGPKRSPTKAVYNARHWN) within Wapl's unstructured N-terminal extension associated with a quite separate segment of SA2, in this case binding parallel to one of its central HEAT repeats (*Figure 15B*). Contact of this motif with the SA2:Scc1 complex is extensive, running along the entire length of the relevant SA2 HEAT repeat, crossing and in so doing, contacting the Scc1 polypeptide and finally interacting via its conserved tryptophan residue C-terminal HEAT repeats on the other side of SA2's hook. This part of Wapl corresponds to the Wapl-M sequences (510–570) important for its association with SA2:Scc1 and previously thought to bind the CES (*Hara et al., 2014*; *Ouyang et al., 2013*). AF predicts a very different interaction, and besides which, it suggests that the CES is occupied instead by Wapl's CTD (*Figure 15B*). The AVYNARHWN part of the motif is conserved in animals (consensus A[V/L]Yxx[R/K][H/-]W) and its mode of binding may explain why Wapl binds to Scc1/Rad21 alone in vitro (*Shintomi and Hirano, 2009*). Significantly, AF failed to predict any interaction between SA2:Scc1 and Wapl's FGF motifs, at least when the N-terminal helices within Wapl's CTD are present and predicted to bind to SA2's CES.

The first alpha helix of Wapl's CTD (α1) is highly conserved (Figure 17A, right; see also *Figure 11B*), especially, but not exclusively (for reasons that will become apparent) residues facing SA2. Crucially, its interaction with SA/Scc3 appears to be of physiological importance. Scc3(K404) likely forms a conserved salt bridge with yeast Wapl's D272, while the equivalent residue in SA2, K330 does likewise with human Wapl's D657. Remarkably, mutation of all four residues (individually) has been shown

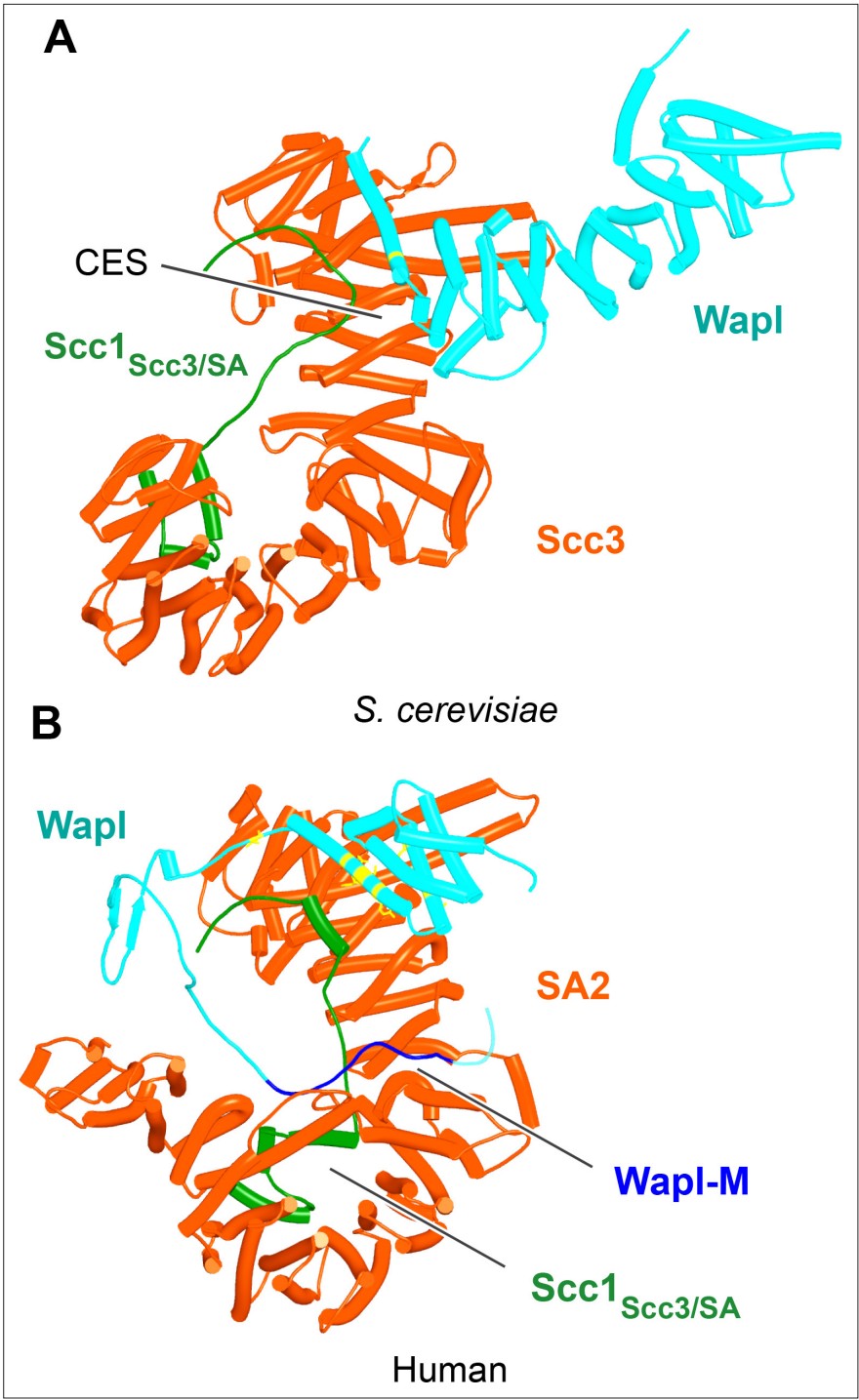

**Figure 15.** Interaction of Wapl's CTD with Scc3 and SA proteins. (**A**) AlphaFold 2 (AF) prediction for the interaction between Scc3/Scc1 and the CTD of yeast Wapl (f24). Chains in the PAEs from f24 are Scc3 (A), Scc1 (B), and Wapl (C). (**B**) AF prediction for the interaction between SA2/Scc1 and a fragment of human Wapl containing its N-terminal sequences as well as the N-terminal half of its helical CTD (f25). Chains in the PAEs from f25B are Scc1 (A), SA2 (B), and Wapl (C). Mutations that abolish releasing activity (RA) are marked in yellow: E653, D656, D657, L676, and K684.

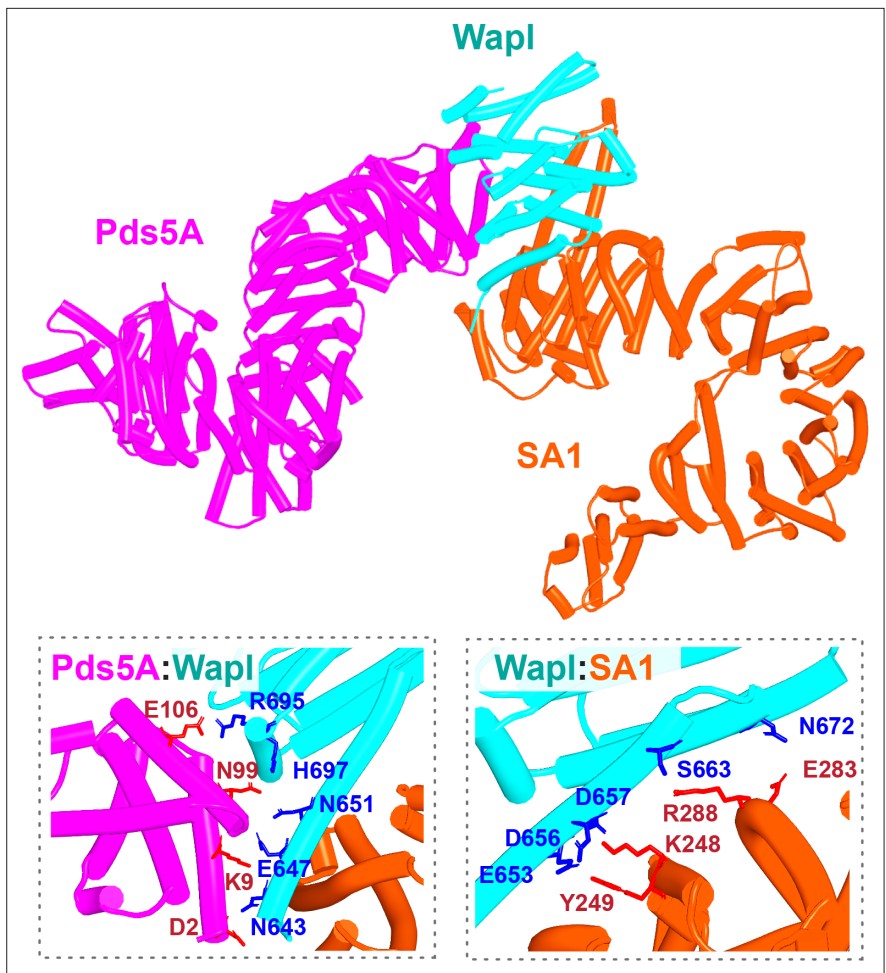

**Figure 16.** AlphaFold 2 (AF) prediction of a complex between human Pds5A, SA1, and a fragment of Wapl's CTD (f27). Chains in the PAEs from f27 are Wapl (A), SA1 (B), and Pds5A (C). Pds5 and SA1 were not predicted to interact in the absence of Wapl (f26).

to either abrogate or to greatly reduce RA mediated by Wapl (*Beckouët et al., 2016*; *Ouyang et al., 2013*; *Rowland et al., 2009*). *S. cerevisiae* Wapl(D272G) was in fact isolated as a spontaneous suppressor of *eco1* lethality in yeast. The AF prediction for the human proteins explains the greater conservation of the surface of Wapl's α1 facing SA2 as well as the more N-terminal V639, which also contacts SA2. Crucially, the AF model explains why individual mutation of E653, D656, D657, or Y660 within the CTD's α1 helix abrogates RA, as does mutation of L676 or K684 in its α2 helix (*Ouyang et al., 2013*). All these residues are predicted by AF to contact SA2 (*Figure 15B*, residues marked in yellow).

The prediction that in humans Wapl's CTD forms a complex with Pds5 as well as with SA2:Scc1 raises the possibility that both interactions might occur simultaneously and that Wapl would therefore glue all three proteins together in a ternary complex. This seems indeed to be the case, as AF failed to predict any direct interaction between full-length SA1 and Pds5A (f26), but did so with high confidence in the top two ranked models created in the presence of the part of Wapl's CTD that it predicts to bind both SA and Pds5 (*Figure 16*, f27). The prediction statistics indicated that the most accurate images of the ternary complex were obtained when only the N-terminal domains of SA1 and Pds5A were included. In this case, AF predicted a ternary complex with high confidence in five out of five models (*Figure 17A*, f28). Using this template, we screened for the formation of ternary SA:Wapl:Pds5 complexes in a variety of eukaryotic lineages. Identical ternary complexes were predicted with high confidence in plants (*Arabidopsis thaliana*), insects (*Drosophila melanogaster*), and a basal fungus, namely the chytrid *Spizellomyces punctatus* (*Figure 17B*, f29, f30, f31). Crucially, formation of the

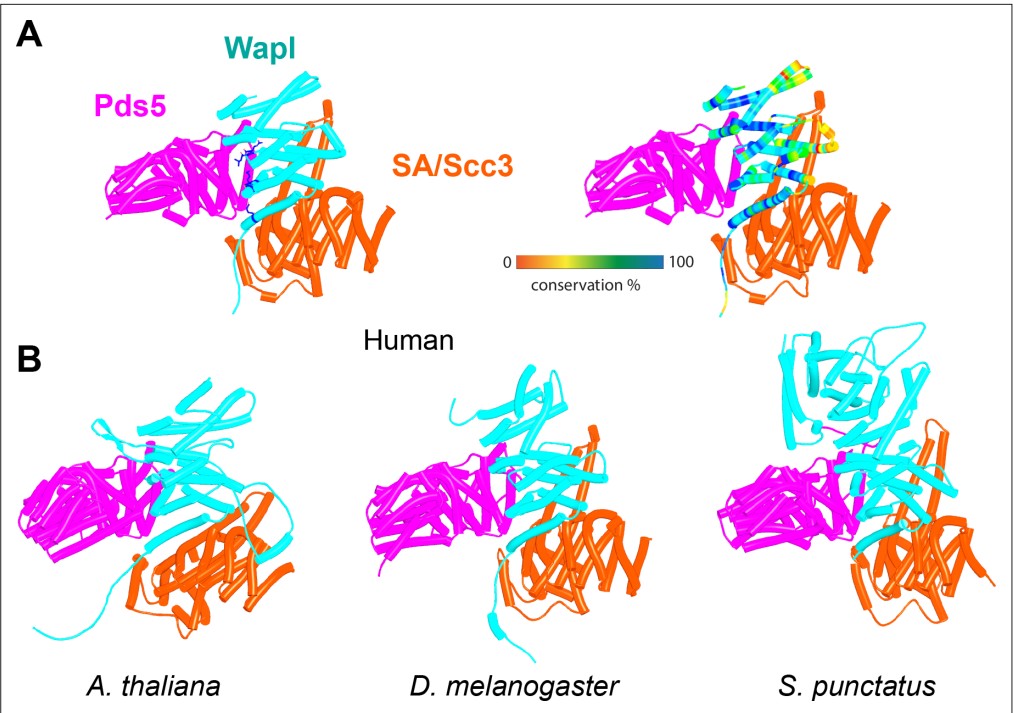

**Figure 17.** AlphaFold 2 (AF) predictions for complexes between N-terminal fragments of Pds5 and SA/Scc3 and the N-terminal half of Wapl's CTD for humans (**A**, f28), *A. thaliana* (**B**, f29), *D. melanogaster* (f30), and *S. punctatus* (f31). Also shown is Wapl's conservation mapped on the human complex (**A**, right). Chains in the PAEs from f28 are Pds5 (A), Wapl (B), and SA (C). Chains in the PAEs from f29 are Scc3 (A), Pds5 (B), and Wapl (C). Chains in the PAEs from f31 are Pds5 (A), Scc3 (B), and Wapl (C).

ternary complex explains the conservation and functional importance of E647 (*Ouyang et al., 2013*) within human Wapl's α1 helix despite not facing SA2, as it contacts Pds5's NTD. Likewise, contact with Pds5's NTD in the AF model explains abrogation of Wapl-mediated RA by R695E as well as conservation of M692, R695, R731, and L732 in animals (*Figure 17A*, right). We will refer to this as the canonical Pds5:SA/Scc3:Wapl rigid body ternary complex.

In contrast, despite the robust formation of binary complexes between Scc3 and Wapl orthologs, AF failed to predict the same type of ternary complexes with the equivalent protein fragments from *S. cerevisiae* (f32A). Further analysis of the way Wapl interacts with Pds5 in *S. punctatus* revealed a possible explanation. When the former's CTD was attached to N-terminal residues containing its TYxxxR[T/S]ΦL motif, AF predicted that Pds5's NTD formed a rigid body complex with the CTD, whose conformation was identical to that observed in the canonical Pds5:SA/Scc3:Wapl rigid body ternary complex (*Figure 17B*), while its APDAP loop simultaneously bound Wapl's TYxxxR[T/S]ΦL motif (*Figure 18*, f19E), in the manner observed in *S. cerevisiae* (f19C) and *N. crassa* (f19B). This raised the possibility that an interaction between Wapl's TYxxxR[T/S]ΦL motif and Pds5's APDAP loop might actually facilitate formation of ternary complexes and that this might explain why the TYxxxR[T/S]ΦL motif is essential for RA in *S. cerevisiae*. We therefore repeated the AF predictions for the previously described *S. cerevisiae* Pds5, Scc3, and Wapl fragments (f32A) but in this case included sequences containing Wapl's TYxxxR[T/S]ΦL motif. Remarkably, the top three ranked models now predicted with high confidence a rigid body ternary complex (*Figure 19A*, f32B), whose conformation closely resembles the canonical Pds5:SA/Scc3:Wapl rigid body ternary complex observed in animals, plants, and basal fungi. Moreover, this was accompanied by the interaction between the TYxxxR[T/S]ΦL motif and Pds5's APDAP loop, as observed in *S. punctatus*. The dramatic effect of including the TYxxxR[T/S]ΦL motif is best visualized by comparing the PAEs of the top three ranked models with and without the motif (*Figure 19B*).

AF also predicted with high confidence in two out of five models a very similar type of canonical ternary complex in *N. crassa* (f33A) and the basidiomycete *Ustilago maydis* (f33B). As in *S. cerevisiae*,

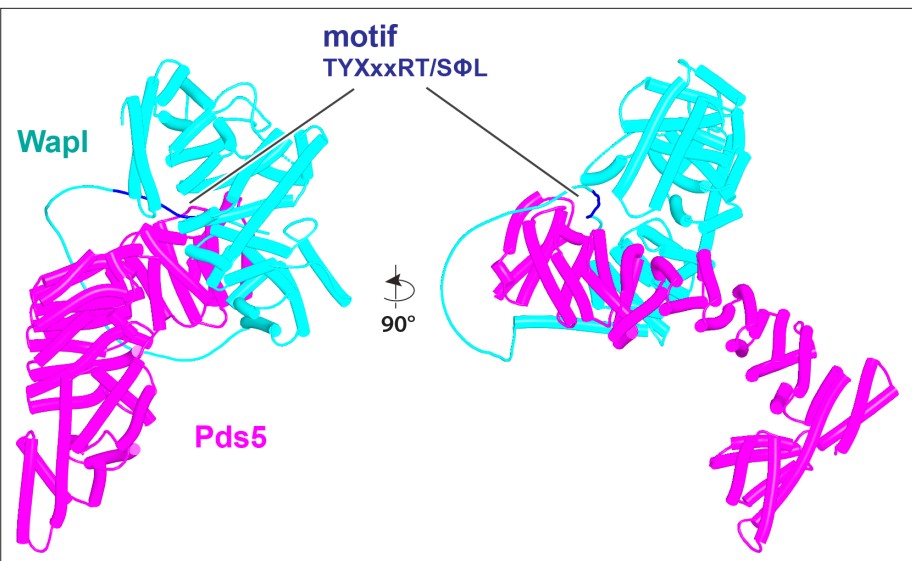

**Figure 18.** AlphaFold 2 (AF) prediction for the interaction between the N-terminal half of *S. punctatus* Pds5 and a fragment of Wapl containing its TYxxxRT/SΦL motif connected to its CTD (f19E). Both the motif and the CTD bind to Pds5. Chains in the PAEs from f19E are Wapl (A) and Pds5 (B).

formation of the canonical Pds5:Scc3:Wapl ternary complex in *Neurospora* was also strictly dependent on inclusion of its TYxxxR[T/S]ΦL motif (compare f33A and f33C). These observations not only suggest that formation of a canonical Pds5:SA/Scc3:Wapl rigid body ternary complex is conserved in fungi as well as animals and plants, but also that its formation may be an essential feature of RA, because deleting the TYxxxR[T/S]ΦL motif necessary for its in silico prediction in *S. cerevisiae* (f32) abolishes RA in vivo (*Figure 11B*). It is remarkable that the TYxxxR[T/S]ΦL motif is not only highly conserved among fungi but that it is connected to α1 in Wapl's CTD by a short polypeptide linker whose length (25–50 residues) does not vary greatly. In other words, the shortness of this tether may be important for ensuring the particular type of interaction between Wapl's CTD and Pds5's NTD observed in canonical Pds5:SA/Scc3:Wapl rigid body ternary complexes.

Though the structure of the ternary Pds5:SA/Scc3:Wapl RA complex in animals (*Figure 20*, f35), plants, and fungi is clearly intriguing and presumably important for RA, it does not per se offer insight into the actual mechanism by which these three proteins facilitate cohesin's release. This is especially true for the conserved cleft at the top (C-terminal end) of Wapl's CTD. Despite differences in the precise patterns of conservation in animals, plants, and fungi, it is remarkable that in all three lineages, motifs within Wapl's unstructured N-terminal extension are predicted by the AF structure database to bind their own clefts (yeast shown in *Figure 12B*). As will become apparent later, AF also predicts that Sororin's conserved FGF motif (which is related to the yeast Wapl's FNF) also binds the cleft. The fact that Sororin is an antagonist of Wapl suggests that the binding by N-terminal Wapl sequences represents a form of autoinhibition.

Potential insight into the function of the highly conserved C-terminal Wapl cleft, which is predicted to be bound by motifs within Wapl's N-terminal sequences, was revealed by a separate set of predictions, namely investigation of how Scc1 binds to Pds5, either in the presence or absence of Smc3's neck. Functional studies in yeast combined with X-ray crystallography of a Pds5:Scc1 complex from *Lanchea thermotolerans* revealed association of an extended Scc1 peptide, from *Lt* S125 to V141, along the axis of Pds5's spine, half way down from its N terminus (*Lee et al., 2016*). This mode of interaction is essential as mutations affecting the interaction, namely Scc1(V137K) and Pds5(Y458K), abrogate binding of the two proteins in vivo and are lethal (*Lee et al., 2016*). Pds5 is required to maintain cohesion in post-replicative cells (*Hartman et al., 2000*; *Panizza et al., 2000*), when Scc1's NTD is bound to Smc3's neck (*Gligoris et al., 2014*), as well as being involved in RA (*Rowland et al., 2009*), when Scc1's NTD must dissociate from Smc3, at least transiently (*Beckouët et al., 2016*; *Chan et al., 2012*). With the initial aim of revealing the orientation of Pds5 connected to the Smc3 neck via Scc1, we used AF to predict how the N-terminal half of Pds5 interacts with the N-terminal 160 amino

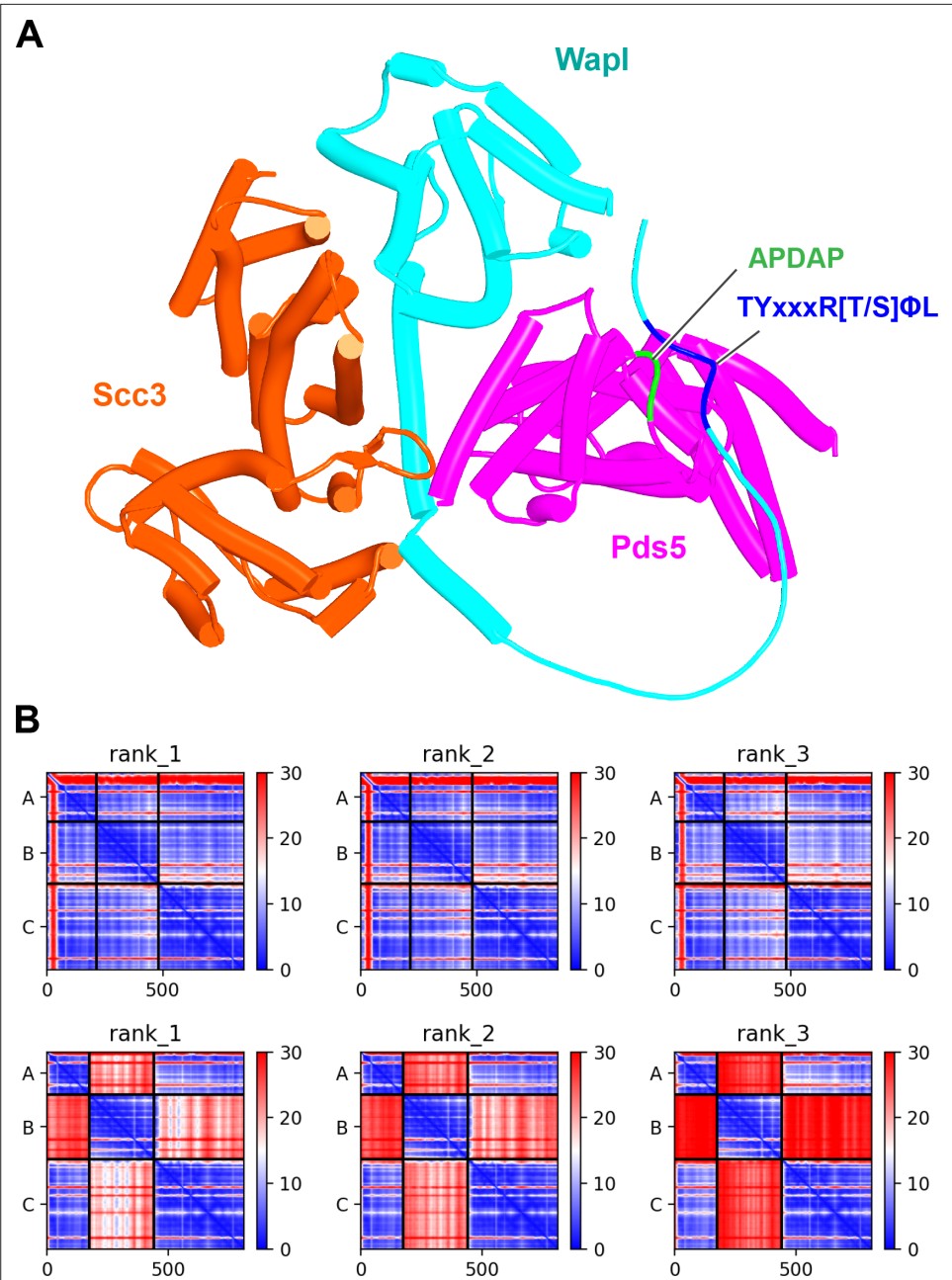

**Figure 19.** Formation of a canonical Pds5:Scc3:Wapl ternary complex in yeast. (**A**) AlphaFold 2 (AF) prediction for a *S. cerevisiae* Pds5:Scc3:Wapl rigid body ternary complex accompanied by interaction between Wapl's TYxxxR[T/S]ΦL motif and Pds5's APDAP loop (f32B). (**B**) PAEs for the top three ranked models for *S. cerevisiae*'s Pds5:Scc3:Wapl ternary complex with (top row, f32B) and without (bottom row, f32A) Wapl's TYxxxR[T/S]ΦL motif. Note that the lower confidence complexes predicted by AF for those lacking Wapl's TYxxxR[T/S]ΦL motif are not of the canonical variety. In both (**A**) and (**B**), chains A, B, and C correspond to Wapl, Pds5, and Scc3, respectively. Interaction between Wapl's TYxxxR[T/S]ΦL motif and Pds5 is represented by the thin blue lines on the left and upper squares of the top three maps in (**B**). Chains in the PAEs from f32B are Wapl (A), Pds5 (B), and Scc3 (C).

acids of Scc1 in the presence of Smc3's ATPase and neck. Not surprisingly, all five models predicted Scc1's simultaneous association with both the Smc3 neck and with Pds5, in both cases in the manner expected from crystallographic studies (PDB 5F0N and 4UX3) (*Figure 21*, f36). More unexpected were the predictions for the sequences that connected the Scc1 S125-V141 interval to the NTD bound to Smc3. This revealed that Scc1's association with Pds5's spine extended, in an N-terminal direction on

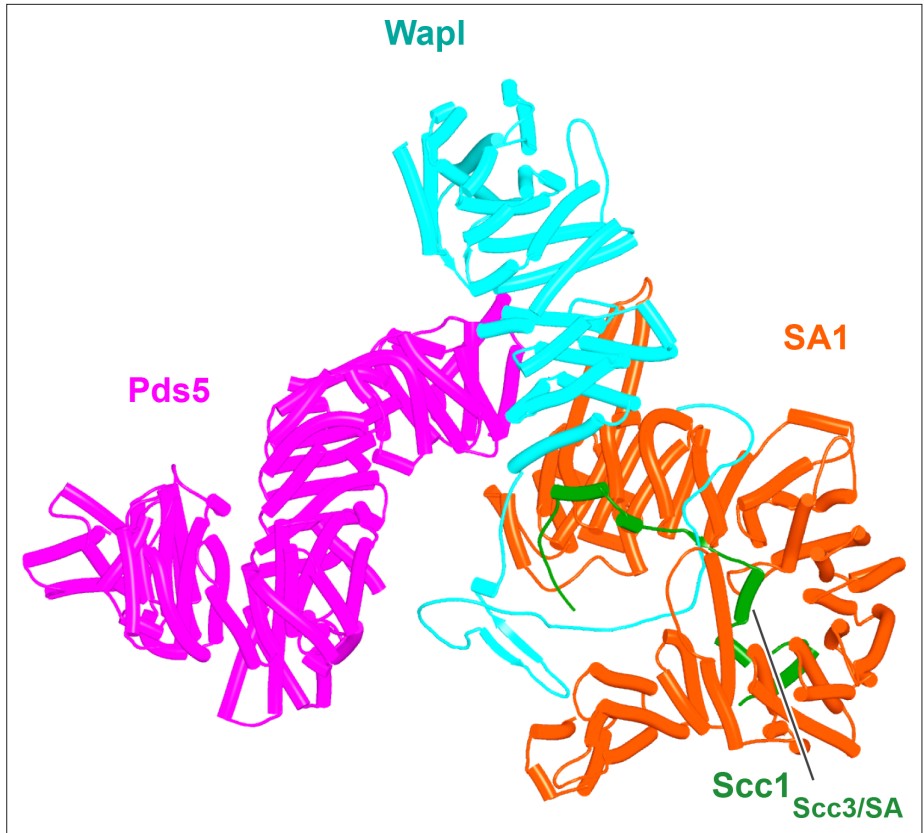

**Figure 20.** Composite model of the human Pds5:SA1:Wapl:Scc1 quarternary complex (f35) created using by superimposing rank 1 models from f23, f25, and f28.

both proteins, from the previously characterized site of interaction almost but not quite to the C-terminal end of the long α3 helix that forms a three-helical bundle with Smc3's coiled coil (*Figure 21*). In all five models, the short peptide linking α3 with the part of Scc1 that is associated with Pds5's spine adopts the same conformation, which ensures that Pds5's NTD points toward the Smc3 ATPase head. Though we have previously suggested that such an orientation might help to direct Eco1 to its K112 and K113 targets on the Smc3 ATPase (*Lee et al., 2016*), subsequent AF predictions (see below) imply that this orientation would not in fact facilitate acetylation and the orientation presumably has some other function. Given that the sequences linking $Scc1_N$'s α3 with its canonical Pds5 binding motif are poorly conserved between fungi and animals, it was remarkable that AF predicted a very similar structure with the orthologous human proteins (*Figure 21*, f37).

Because RA depends on Wapl's association with Pds5, $Scc1_N$ is presumably bound to Pds5 upon its dissociation from Smc3 during a release reaction. We therefore used AF to predict how $Scc1_N$ alone binds to Pds5. In the absence of Smc3, Smc3's coiled coil no longer poses a constraint on how $Scc1_N$ binds with Pds5, and AF predicts that yeast Scc1's association now extends to the C-terminal end of α3, whose interaction with the ends of a pair of Pds5's HEAT repeats leads to a pronounced rotation, as well as levering backward of the Scc1 NTD (*Figure 22*, f38). This change is not accompanied by any alteration of its actual structure or in the manner in which more C-terminal Scc1 sequences bind the Pds5 spine. In humans, α3's association with the Pds5 spine was less affected by Smc3's removal but it was nevertheless levered further backward, becoming more orthogonal to the spine's axis (*Figure 22*, f39). Crucially, in both cases, as well as in *A. thaliana* (*Figure 22*, f40), the C-terminal end of $Scc1_N$'s α3 is anchored on the Pds5 spine and juts out from it at a defined angle. The change in the manner of $Scc1_N$'s association with Pds5 upon dissociation from Smc3 might account for the observation that Pds5 inhibits $Scc1_N$'s association with Smc3 in vitro, but does not affect its association once bound (*Ouyang et al., 2016*). Given that Pds5 is intimately involved in RA and that the latter requires dissociation of Scc1's NTD from Smc3, it is plausible to imagine that upon its dissociation (presumably in

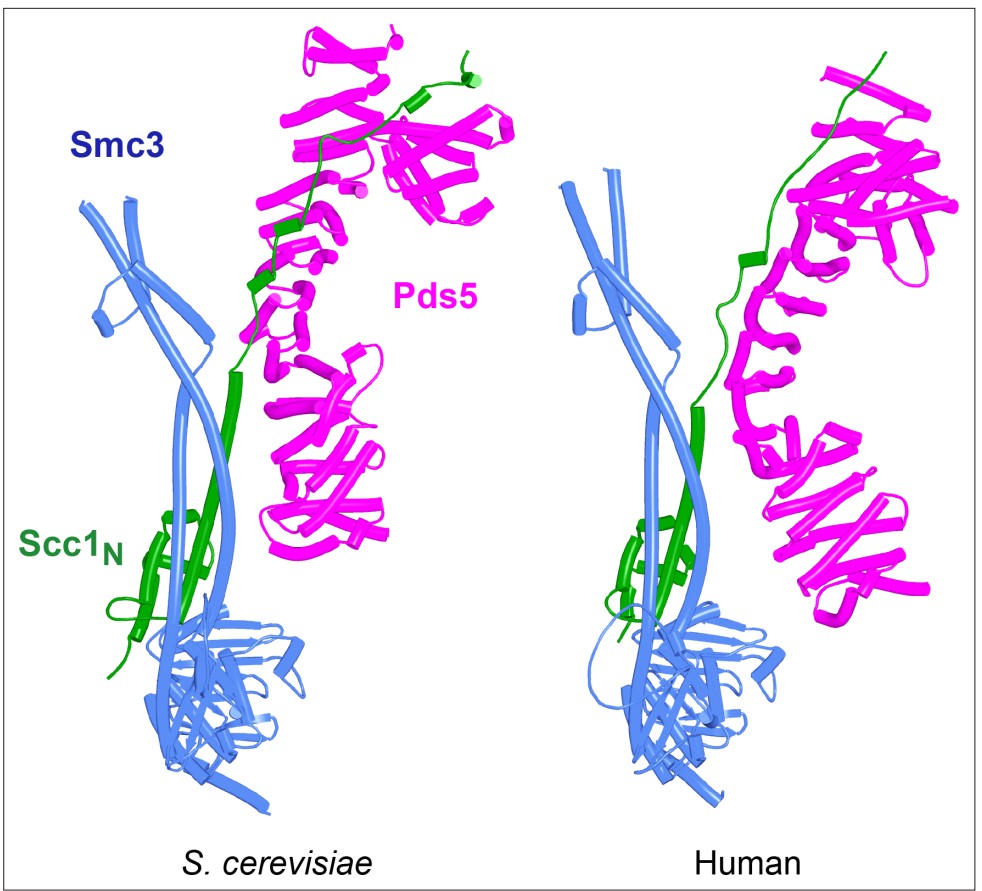

**Figure 21.** AlphaFold 2 (AF) predictions for the interaction of Scc1's N-terminal 160 residues with Smc3 and with an N-terminal fragment of Pds5 for both *S. cerevisiae* (f36) and human (f37). Chains in the PAEs from f36 are Smc3 (A), Smc3 (B), Scc1 (C), and Pds5 (D). Chains in the PAEs from f37 are Scc1 (A), Pds5 (B), Smc3 (C), and Smc3 (D).

response to head engagement and joint junction), Scc1's NTD remains associated with Pds5 in the conformations observed in the above AF models. Crucially, AF predicts that under these conditions, Scc1's NTD adopts a defined position relative to Pds5's NTD that is similar in fungi, animals, and plants.

Having realized that Pds5 organizes the spatial position of Scc1's NTD, we superimposed the human Pds5:Scc1$_N$ model (*Figure 22*, middle, f39) onto the canonical Pds5:SA/Scc3:Wapl rigid body ternary complex (*Figure 23*, f41). Remarkably, this revealed that the N-terminal end of Scc1's NTD is placed, through its association with the spine of Pds5, in the vicinity of the aforementioned cleft at the C-terminal end of Wapl's CTD. Supposition of the canonical Pds5:Scc3:Wapl rigid body ternary complex from yeast (f32B) with AF predictions for Scc3's association with Scc1 and full-length Wapl (f24) and full-length Pds5 associated with Scc1$_N$ (f38B) revealed a very similar looking quarternary complex (Figure 31B).

As mentioned, in humans and most other metazoa, as well as in many fungi (*Figure 12B*) and plants, this cleft is predicted to be occupied by a motif (PSCLSVCNVT) within Wapl's N-terminal sequences (AlphaFold Protein Structure Database: AF-Q7Z5K2-F1). The revelation that Scc1's NTD is positioned close to this cleft in a putative Pds5:Scc3:Wapl:Scc1 complex suggested that the role of this site might be to sequester Scc1's NTD following its dissociation from Smc3's neck. Testing this hypothesis was not possible using a full quarternary complex due to its size. However, predictions of Wapl's position relative to Pds5's NTD are in fact similar whether or not Wapl's CTD is also bound to the SA subunit and we therefore used AF to predict the structure of a complex containing Wapl's CTD, the first 150 residues of Scc1(Scc1$_N$), and the N-terminal straight section of Pds5 whose spine binds Scc1. All five models predicted with high confidence a ternary complex between Wapl, Pds5, and Scc1$_N$ (*Figure 24*, f42). In three of the five, including ranks 1, 2, and 5, Scc1's NTD was placed in close vicinity of the Wapl

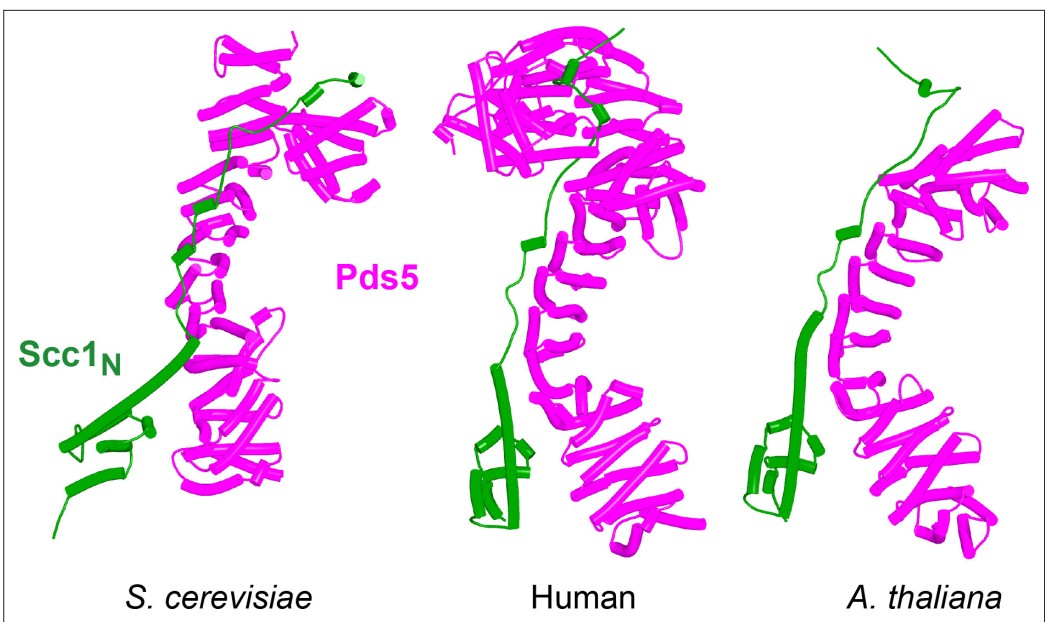

**Figure 22.** AlphaFold 2 (AF) predictions for the interaction between Scc1's N-terminal 160 residues and Pds5 in *S. cerevisiae* (f38), humans (f39), and *A. thaliana* (f40).

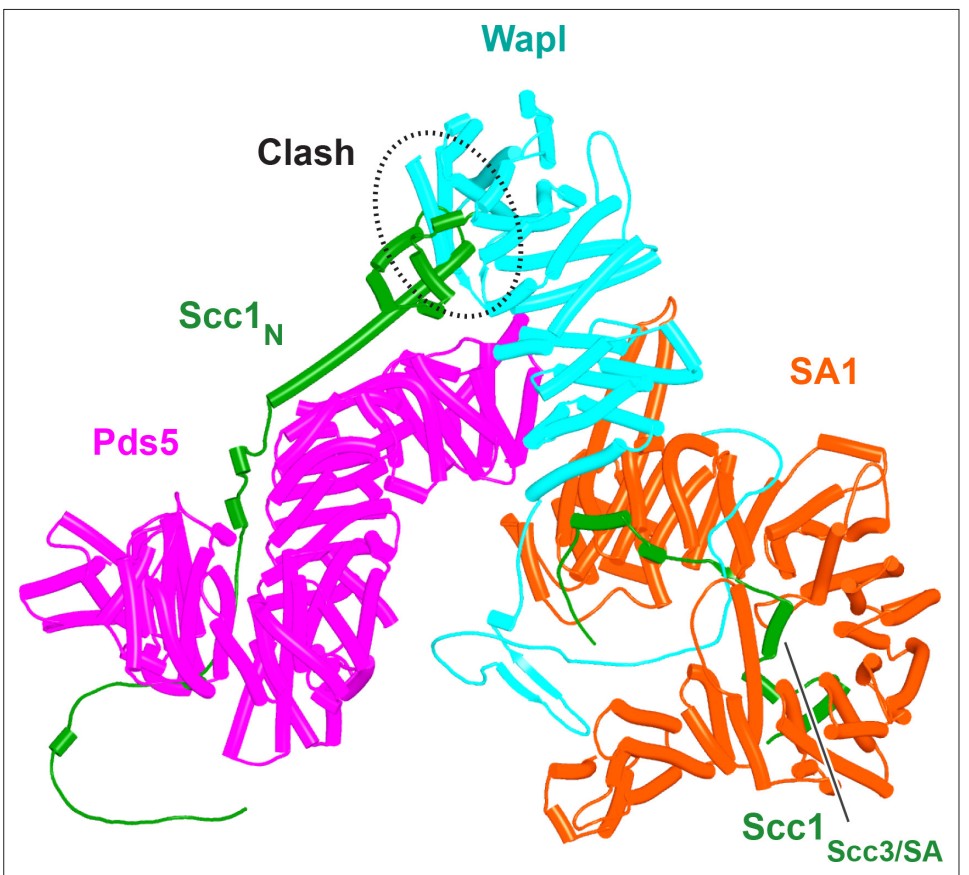

**Figure 23.** A putative quarternary complex between human Scc1$_N$ Pds5, SA1, and Wapl, created by super-imposing f23, f25, f28, and f39. The Scc1's NTD points toward, but clashes with the highly conserved cleft in Wapl's CTD that is otherwise predicted to bind a motif PSCLSVCNVT (conserved among metazoa) located in Wapl's N-terminal sequences (see *Figure 26B*).

cleft. The most N-terminal (α-1) of its four short helices (α-1, α0, α1, and α2), normally packed on top of the α3 N-terminal helix, dissociates from α3 and binds in the cleft (f42). However, these peripheral associations between Scc1$_N$'s α-1 and Wapl were not predicted with high confidence (f42), did not look convincing, and more seriously, differed in their details between models.

To test whether the positioning and hence orientation of Scc1's NTD through its association with Pds5 hinders its association with Wapl's CTD, we asked how the latter interacts with Scc1's NTD alone, that is, when not constrained in space through its association with Pds5 within the rigid body ternary complex. In this case, AF predicts with high confidence and in four out of five models (*Figure 25*, f34A) that Scc1's N-terminal helix (α-1) binds the cleft in Wapl's CTD that otherwise binds Wapl's own SCLSVCNVT peptide (*Figure 26B*), while its second helix (α0), lying at 90° to α-1, is sandwiched between two long antiparallel helices that form a pronounced nose at the C-terminal end of Wapl and a second pair that cross over at 45°. AF predicts similar interactions for *D. melanogaster* (f34B), *Caenorhabditis elegans* (f34C), and for *A. thaliana* (f34D). Moreover, a slightly different but clearly related interaction was also predicted for *N. crassa* (f34E). Remarkably, only modest changes in the conformation of Scc1's NTD are required to accommodate binding of its α-1 and α0 helices to Wapl's CTD in this manner, principally a slight bending of α3 to avoid a clash with the pair of Wapl CTD helices that cross its nose. As a consequence, α3 lies on top of and almost parallel with Wapl's nose and this kind of alignment is presumably why AF failed to predict Wapl's interaction with Scc1's α-1 and α0 when Scc1's NTD is connected to Pds5 when the latter is part of a ternary complex with Wapl (*Figure 24*, f42). In the latter case, Scc1's NTD approaches Wapl's CTD from an angle incompatible with the stereo-specific engagement depicted in *Figure 25*. Additional AF predictions confirmed that Scc1's first three mini-helices (α-1, α0, and α1) are sufficient for this type of interaction not only for humans (*Figure 26A*, f43) but also for *D. melanogaster* (*Figure 26C*, f44), for *C. elegans* (*Figure 26D*, f45), for the sponge *Amphimedon queenslandica* (*Figure 26E* and f46), for the green alga *P. salinarum* (*Figure 26F*, f47), and for the vascular plant *A. thaliana* (*Figure 26G*, f48). Likewise, despite considerable sequence variation, AF predicted a very similar interaction between Wapl and the equivalent helices of a fragment of human Rec8 containing its first 36 amino acids (*Figure 26H*, f49).

These results suggest that there may indeed be an intrinsic affinity between Wapl and Scc1's α-1 and α0 helices but the interaction is not possible when the helices are packed onto α3 while the latter is simultaneously connected to Pds5 bound to Wapl's CTD. If positioning Scc1's NTD in the immediate vicinity of Wapl's CTD, through formation of a stereo-specific ternary complex between Pds5, Wapl, and Scc1$_N$ is to promote sequestration of Scc1's α-1 α0 helices by Wapl's CTD, then these helices would have to dissociate from α3. Crucially, AF would not predict the unfolding necessary if it gave higher priority to packing of Scc1's N-terminal helices with α3 than to partial unfolding and association with Wapl. That this appears to be the case should not be surprising, given that Scc1's NTD spends most of its life associated with Smc3 during which the α0 helix is packed onto top of α3. Even when not associated with Smc3, Scc1 may not actually be engaged in a release reaction, which after all is probably a rare event in the life of cohesin, only occurring once every 20 min in G1 cells (*Gerlich et al., 2006*; *Wutz et al., 2020*).

There are two possible explanations for the conundrum that Scc1's NTD appears to approach Wapl's CTD from the wrong direction: either the hypothesis that Wapl sequesters Scc1's NTD is wrong or Wapl does indeed sequester Scc1's NTD when delivered precisely to the right location by the Pds5 moiety of the Pds5:Scc3:Wapl:Scc1 quarternary complex, but to do so in the manner depicted in *Figure 26* requires a partial unfolding of Scc1$_N$, in particular dissociation of Scc1$_N$'s α-1, α0, and α1 helices from α3, a feature that AF may not on its own be able to predict. We therefore explored whether the packing of the three most N-terminal helices around α3 could be weakened by changing key residues within α3, without altering the latter's structure. After a series of trials testing the effects of α3 mutations on the structure predictions of Scc1's NTD alone, we settled on a pentuple α3 mutant Scc1$_N$(H58A, L59S, G62D, R65A, R69A) that, according to the PAE and pLDDT plots (*Figure 27*, f50, f51), weakens the association between Scc1$_N$'s three most N-terminal helices and α3, without greatly affecting the folding of α3 itself. Importantly, AF prediction for Scc1's interaction with the Smc3 neck was also unaltered by the pentuple mutant, not only with regard to α3's interaction with the Smc3 coiled coil, but also with regard to the packing on and around α3 of Scc1's more N-terminal helices. The latter may be surprising given the extensive changes but AF predictions are remarkable resilient to mutations and do not reflect the underlying physico-chemical changes. We then tested the

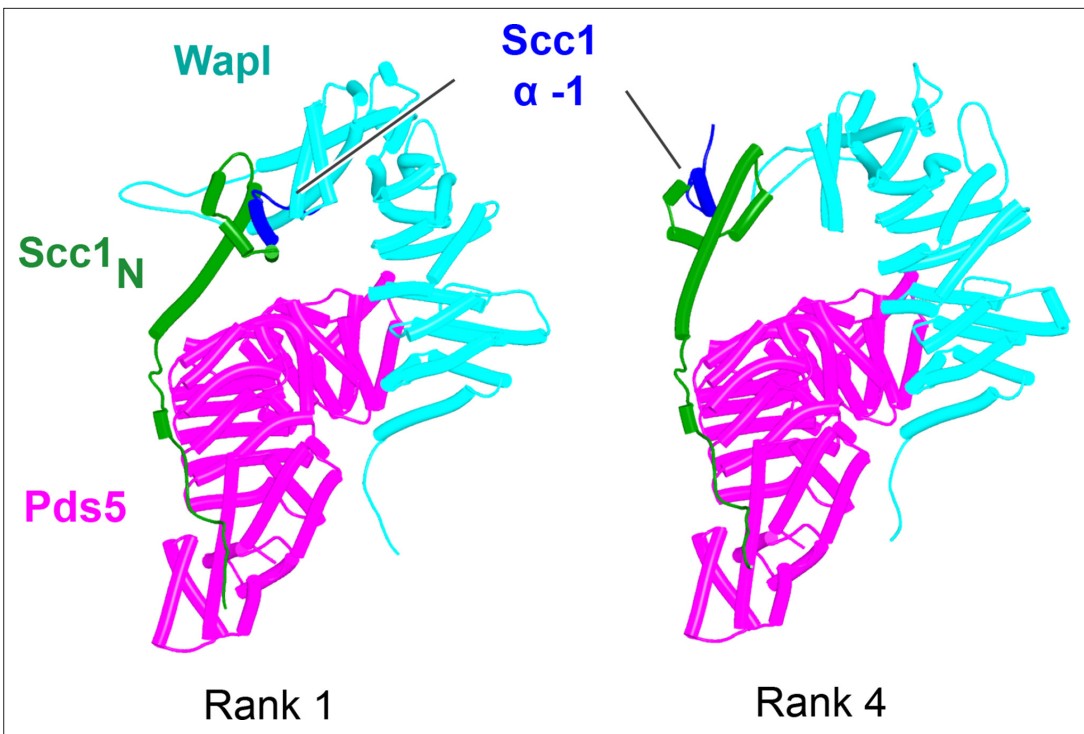

**Figure 24.** AlphaFold 2 (AF) predictions for ternary complexes (models ranks 1 and 4) containing Scc1$_N$, Pds5's NTD, and Wapl's CTD (f42). Chains in the PAEs from f42 are Scc1 (A), Pds5 (B), and Wapl (C).

effect of these mutations on AF predictions for the complex containing Wapl's CTD, the first 150 residues of Scc1, and the N-terminal straight section of Pds5. Remarkably, the mutations led to AF predicting with high confidence and in five out five models the unpacking of Scc1$_N$'s first two helices and their docking with Wapl's CTD (*Figure 28*, f52A) in a manner very similar, if not identical, to that observed when the first two or three helices are run alone with Wapl's CTD (*Figure 26A*) or in AF predictions for Wapl's association with isolated Scc1 NTDs (*Figure 25*, f34). AF even managed to

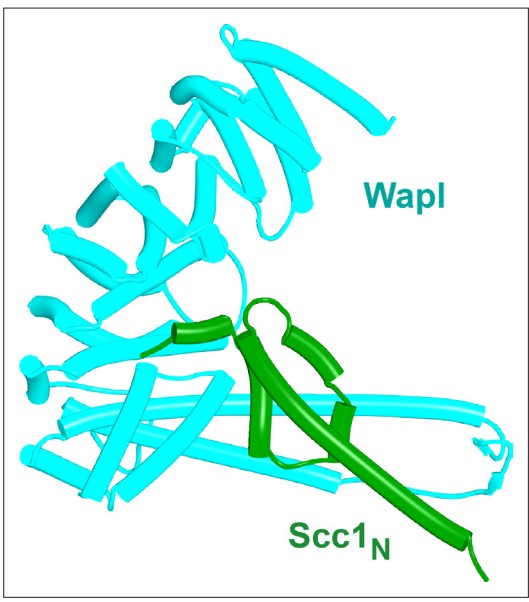

**Figure 25.** Interaction between Scc1's NTD and Wapl's CTD. AlphaFold's prediction for the interaction of human Scc1's NTD with Wapl's CTD (f34A).

predict a similar outcome when the appropriate fragment of SA was included, thereby creating a quarternary complex whose Scc1$_N$ is sequestered by Wapl (f52B). It is important to note that the mutations did not affect any of the residues that actually contact Wapl in these models. Being in α3 itself, which does not associate with Wapl, the mutations merely affect how the two N-terminal helices associate with α3 (*Figure 27*).

If sequestration of Scc1's N-terminal (α-1 and α0) helices by Wapl's CTD is of physiological importance, then the Wapl residues involved in sequestration should be conserved. This consideration cannot be applied to Scc1's NTD as it has multiple other functions. Highly conserved (amongst animals) human Wapl surface residues involved in the interaction include N939 and N990, D979 and E1117, which could interact with Scc1[R11], L983 (which could interact with Scc1[L8]), A1112 and H1115 (which could interact with Scc1[W23]), M1116 (whose sulfur atom could interact simultaneously with Wapl[F1165] and Scc1[W18]), and lastly M1166 (which could

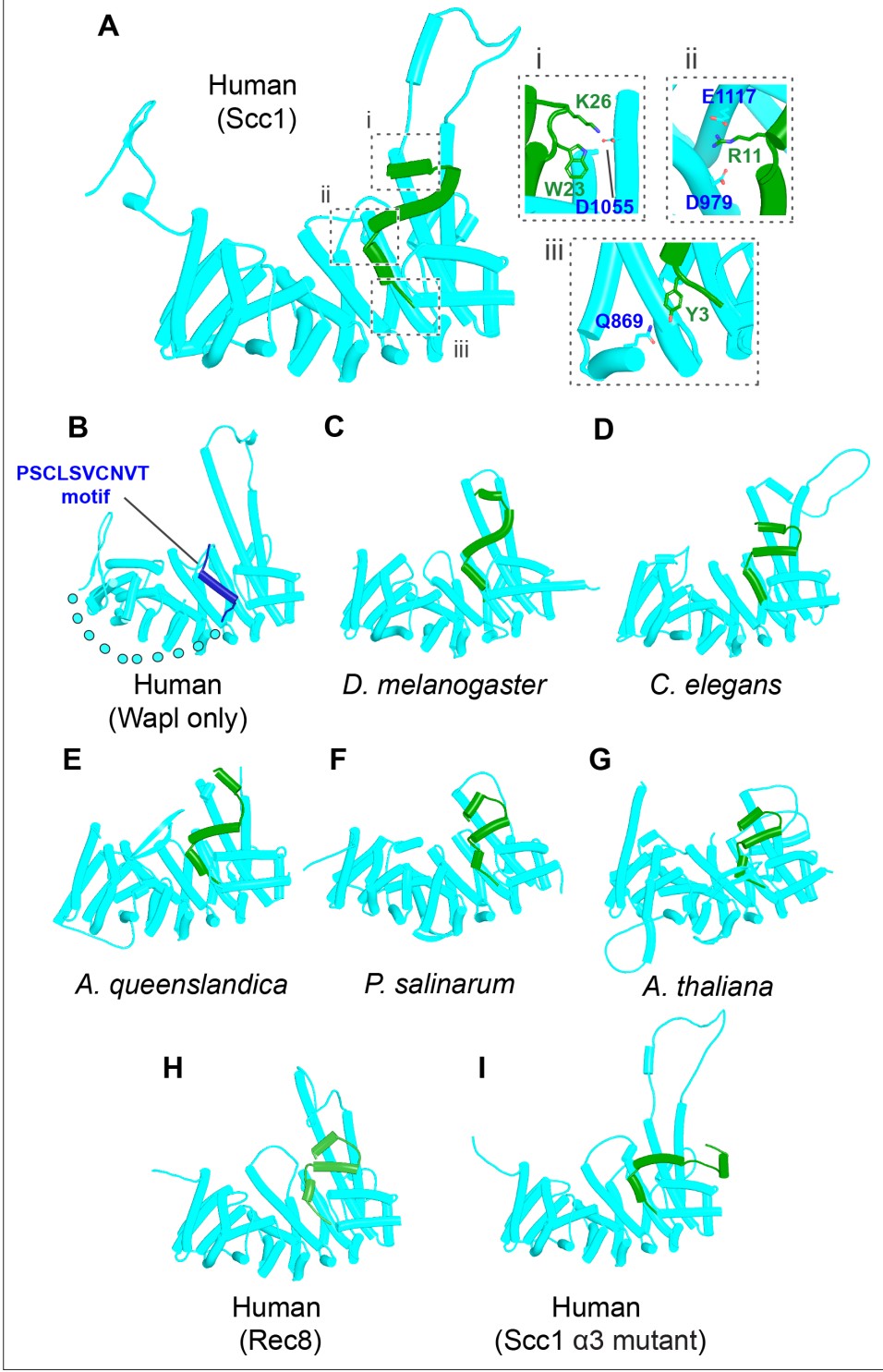

**Figure 26.** AlphaFold 2 (AF) predictions for the interactions between Wapl's CTD and the N-terminal 35 amino acids of Scc1 for humans. (**A**, f43), *D. melanogaster* (**C**, f44), *C. elegans* (**D**, f45), the sponge *A. queenslandica* (**E**, f46), the green alga *P. salinarum* (**F**, f47), and *A. thaliana* (**G**, f48). (**B**), AF prediction for human Wapl alone (AlphaFold Protein Structure Database AF-Q7Z5K2-F1). (**H**) AF prediction for the interaction between Wapl's CTD and the N-terminal 36 amino acids of human Rec8 (f49). (**I**) Interaction between Wapl's CTD and the N-terminal 35 amino acids of Scc1 in a ternary complex containing Scc1$_N$(H58A, L59S, G62D, R65A, R69A), Pds5's NTD, and Wapl's CTD (f52A). See *Figure 28* for the same structure in the complete ternary complex.

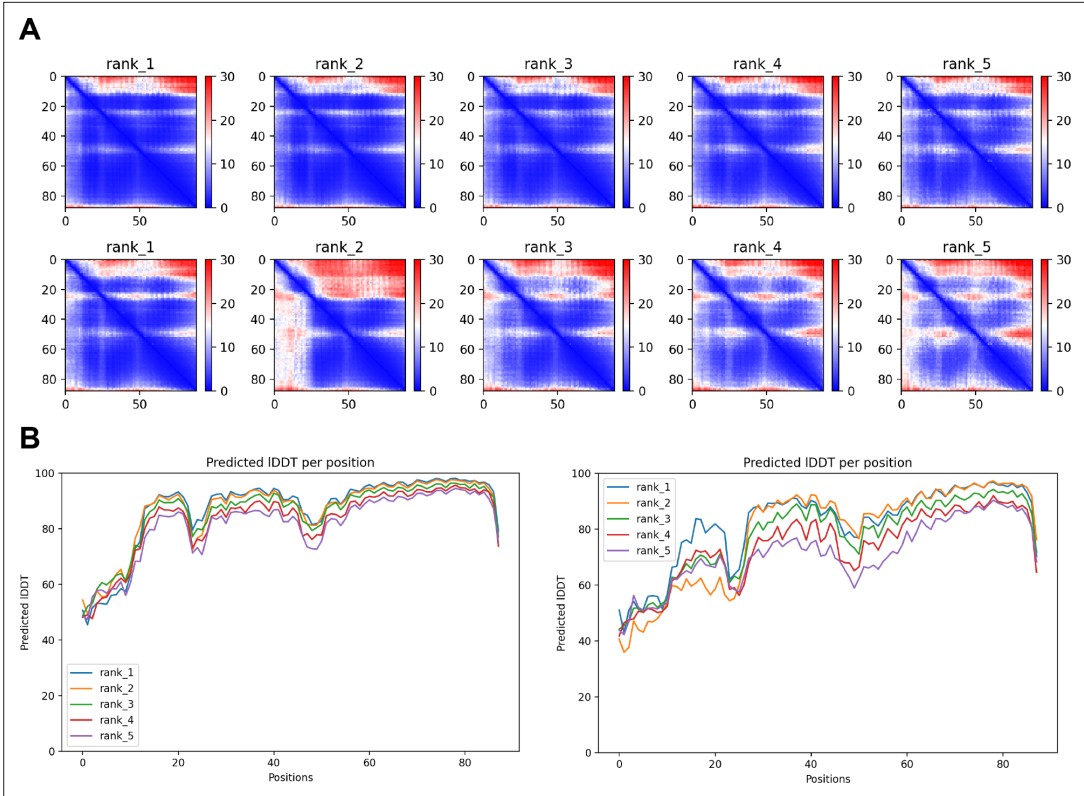

**Figure 27.** Mutations that weaken interaction between Scc1's first two alpha helices with its alpha 3 helix. (**A**) PAEs for WT $Scc1_N$ (top, f50) and mutant $Scc1_N$(H58A, L59S, G62D, R65A, R69A) (bottom, f51). (**B**) pLDDT plots for WT $Scc1_N$ (left, f50) and mutant $Scc1_N$(H58A, L59S, G62D, R65A, R69A) (right, f51).

interact with Scc1[Y3]). Remarkably, several of these residues have been mutated and shown to abrogate Wapl's RA activity in vivo, namely Wapl(D979K), M1116A, E1117K, and F1165A (*Ouyang et al., 2013*). In conclusion, sequestration not only helps explain the pattern of conservation but also the physiological consequences of mutating key residues. Crucially, AF predicts that sequestration can also occur when $Scc1_N$ and Wapl's CTD are juxtaposed within a quarternary complex with SA and Pds5 (f52B). Because very similar rigid body ternary complexes are formed between Pds5, Scc3, and Wapl in plants as well as fungi (*Figure 23B*), the N-terminal helices of their $Scc1_N$s will be presented to Wapl's CTD in a similar manner to that observed in humans. It is therefore entirely plausible that the proposed sequestration mechanism applies to cohesin RA in all three kingdoms.

The cleft within Wapl's CTD putatively bound by $Scc1_N$'s α-1 (from animals, plants, and even certain fungi such as the *Sordariales*) has previously been highlighted by Chatterjee and co-workers (*Chatterjee et al., 2013*), who described a crystal structure (PDB 3ZIL) in which the equivalent cleft from the yeast *Ashbya gossypii* was occupied by a peptide from Smc3's ATPase domain, a phenomenon that seemed implausible from a structural point of view and has subsequently not been confirmed. *A. gossypii*, like *S. cerevisiae*, belongs to the *Saccharomycetacae*, a group of yeasts in which α-1 is missing, a feature unique amongst eukaryotes, and replaced by a short string of non-conserved residues. In *Ashbya*'s case, AF predicts that the cleft is bound by α0 instead of α-1 (f58), a feature that might also apply to *S. cerevisiae*.

## Yeast genetics implies that formation of the canonical Pds5:Scc3:Wapl ternary complex involves multiple steps

Our AF predictions suggested that formation of canonical rigid body Pds5:Scc3:Wapl ternary complexes in ascomycetes depends on a juxtaposition of Wapl's CTD and Pds5's NTD facilitated by Wapl's TYxxxR[T/S]ΦL motif (*Figure 19B*), which is essential for RA in vivo. It is, however, curious that despite its clear effect on canonical ternary complex formation and despite highly reproducible

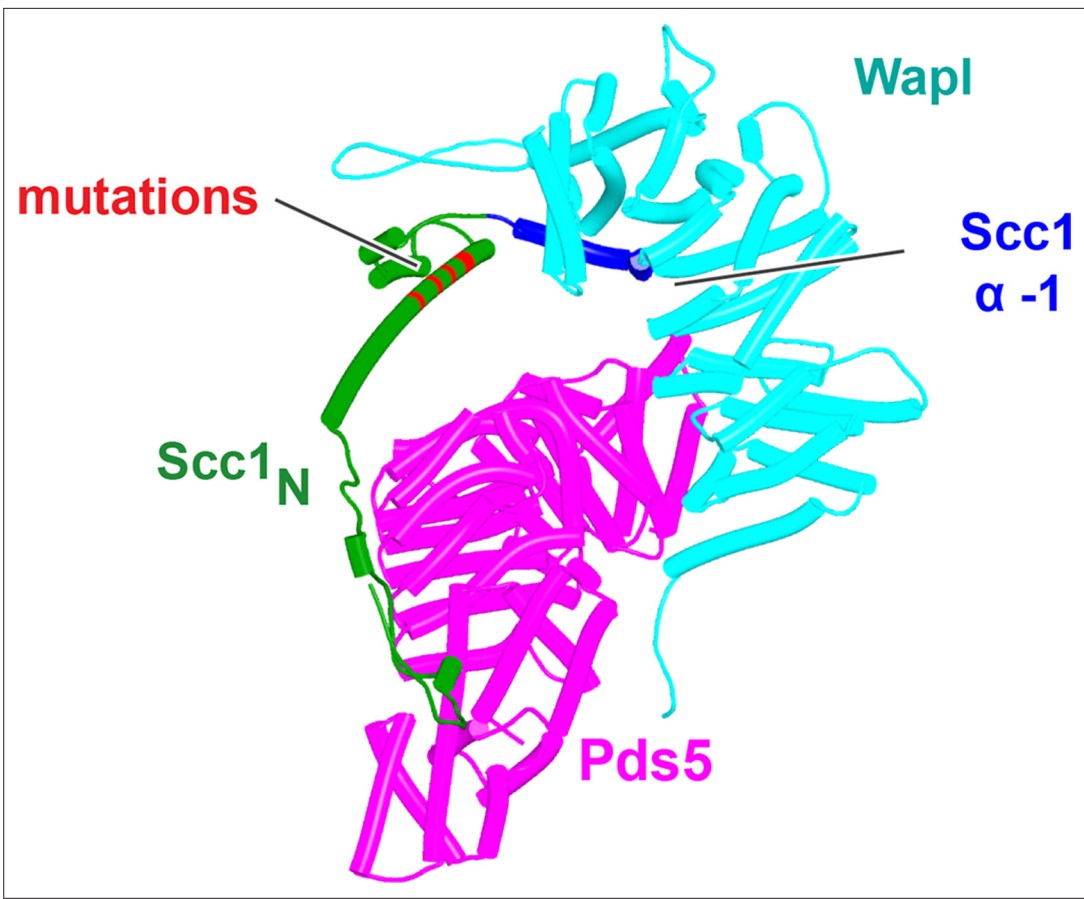

**Figure 28.** AlphaFold 2 (AF) prediction of a ternary complex containing $Scc1_N$(H58A, L59S, G62D, R65A, R69A), Pds5's NTD, and Wapl's CTD (f52A). The location of $\alpha 3$ $Scc1_N$ mutations that distinguish it from f42 shown in *Figure 24* is marked in red. Chains in the PAEs from f52A are Scc1 (A), Pds5 (B), and Wapl (C).

AF predictions for the interaction of this motif with Pds5's APDAP loop using versions of Wapl that lack sequences N-terminal to the motif, AF fails to predict its association with the APDAP loop of full-length yeast Pds5 when Wapl's entire unstructured N-terminal sequences are present (f19A, f19D). A possible reason for this is that AF robustly predicts that the FNFLD motif within yeast Wapl's N-terminal sequences binds to the WEST locus situated at the opposite end of Pds5 and the limited distance between the FNFLD and TYxxxR[T/S]$\Phi$L motifs reduces the likelihood of the latter's interaction with the APDAP loop. Under these circumstances, AF in fact prefers to dock the N-terminal [M/$\Phi$]xxYG[K/R] motif to the APDAP (f19D), possibly because it is further from the FNFLD motif and therefore less restricted by the latter's association with WEST. How then does association between the APDAP loop and the TYxxxR[T/S]$\Phi$L motif come about? A clue emerges from AF predictions for the interaction between Scc3 and Pds5 in yeast. Unlike humans where there is no direct interaction between SA and Pds5, AF predicts with high confidence in five out of five models an interaction between Scc3's C-terminal face ($Scc3_C$) and the pronounced protrusion within the middle of Pds5 ($Pds5_P$) (*Figure 29A*, f54A). Crucially, both surfaces involved are the sites of mutations (*Figure 29B*) that abrogate RA in vivo, namely R996S and R1043L within Scc3's C-terminal face and R578I, C599F, and E602K within the face of $Pds5_P$ predicted to interact with $Scc3_C$ (*Rowland et al., 2009*). The remarkable congruence between the AF prediction and mutations isolated as spontaneous suppressors of *eco1Δ* is unlikely to be a coincidence and therefore implies that the interaction must be of physiological importance and necessary for RA, at least in yeast. There is a simple explanation why no such interaction was detected in humans, namely that SA proteins have seven extra HEAT repeats at their C-termini, which means that they lack the $Scc3_C$ face that interacts with $Pds5_P$ in yeast. These extra repeats are shared by Scc3 orthologs in plants, basal fungi, and basidiomycetes, but lacking in

*S. pombe* and *N. crassa* as well as *S. cerevisiae*. AF also predicted, albeit with low confidence a similar association between Scc3$_C$ and Pds5$_P$ in *N. crassa* (f54E).

Though the Scc3$_C$ face involved in the interaction appears unique to ascomycetes, it has a potentially important implication for how canonical rigid body Pds5:Scc3:Wapl ternary complexes are generated in *S. cerevisiae*. This emerges from the fact that the Scc3$_C$:Pds5$_P$ interaction aligns the axes of the two proteins in parallel and thereby juxtaposes their N-termini, albeit modestly spaced apart (**Figure 29A**). As a consequence, any Wapl CTD bound to Scc3's CES will now be placed in the vicinity of Pds5's NTD, which may favor the binding to the latter's APDAP loop of the TYxxxR[T/S]ΦL motif (associated with Wapl's CTD) over the more distant N-terminal [M/Φ]xxYG[K/R] motif.

To test this, we used AF to predict the structure created by a fragment of Wapl containing its TYxxxR[T/S]ΦL motif together with the part of its CTD that binds Scc3's CES together with full-length Scc3 and a long fragment of Pds5 that contains its entire N-terminal part as well as its Pds5$_P$ protrusion predicted to bind Scc3$_C$. Remarkably, AF predicted with high confidence in five out of five models (f54B) a complex with Scc3$_C$ bound to Pds5$_P$, Wapl's CTD bound to Scc3's CES, a close juxtaposition of the latter with Pds5's NTD, and interaction of Wapl's TYxxxR[T/S]ΦL with Pds5's APDAP loop (**Figure 30**). A similar result was obtained when all of Wapl's N-terminal sequences were included and crucially under these circumstances Wapl's TYxxxR[T/S]ΦL and not its N-terminal [M/Φ]xxYG[K/R] motif bound to Pds5's APDAP loop (f54C).

This finding suggests that the Scc3$_C$:Pds5$_P$ interaction facilitates juxtaposition of Wapl's CTD with Pds5's NTD in manner that would place them close to but not identical to that adopted in the canonical configuration. It also promotes the interaction of Pds5's APDAP loop with Wapl's TYxxxR[T/S]ΦL motif that is ultimately necessary for formation of the canonical rigid body Pds5:Scc3:Wapl ternary complex in yeast. However, the latter is not in fact generated because its precise adoption is incompatible with the manner in which the axes of Scc3 and Pds5 are oriented due to their Scc3$_C$:Pds5$_P$ interaction. As a consequence, in this alternative state, Scc1$_N$ associated with Pds5 is not placed in the position necessary for its sequestration by the cleft in Wapl's CTD (**Figure 31A**). For this to occur, the Scc3$_C$:Pds5$_P$ interaction must subsequently be broken, which allows Pds5's axis to rotate in a manner that slots it into the canonical configuration, which then delivers Scc1$_N$ to the cleft in Wapl's CTD (**Figure 31B**).

Though speculative, the scheme has the merit of explaining why Wapl's TYxxxR[T/S]ΦL motif as well as the Scc3$_C$:Pds5$_P$ interaction are essential for RA in yeast but not in animals, and yet RA in both types of organism appears to share the fundamental mechanism, namely delivery of Scc1N to Wapl's CTD when part of the canonical Pds5:Scc3/SA:Wapl ternary complex. If this mechanism is indeed correct and universal, then we must conclude that not all of the interactions revealed by AF and

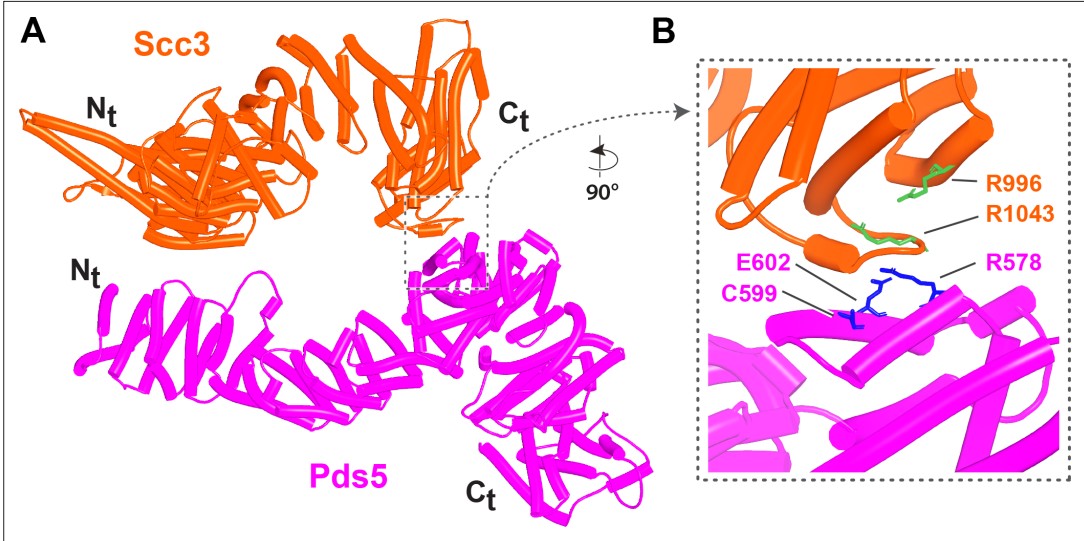

**Figure 29.** Interaction between Scc3 and Pds5 in yeast. (**A**) AlphaFold 2 (AF) prediction for the interaction between Scc3 and Pds5 in *S. cerevisiae* (f54A). (**B**) Closeup showing residues whose mutation abrogates releasing activity (RA) and suppresses eco1Δ. Chains in the PAEs from f54A are Scc3 (A) and Pds5 (B).

by yeast genetics can occur simultaneously. The Scc3$_C$:Pds5$_P$ interaction is clearly incompatible with formation of the canonical ternary complex and yet the former is essential for RA in yeast. In other words, the proteins involved in RA in yeast must exist in multiple states and the process of release must involve transitions between these that presumably depend on a specific combination of on and off rates for the various interactions involved.

## Yeast genetics and AF suggest a novel function for Smc3(K112) and K113

Though our current AF predictions explain many of the existing mutations known to abrogate RA in yeast, as well as providing the first plausible explanation for how Wapl facilitates release, they do not explain how acetylation of Smc3(K112) and K113 by Eco1 in yeast blocks release in a manner that does not require Sororin, nor why release is abrogated by mutations of Smc3's KKD loop, Smc3(K113T, K113N, K112Q K113Q, and D114Q/Y) and its surroundings Smc3(S75R, R107I/S, G110R/V/W) or by Scc3(D189E) and E202K (*Rowland et al., 2009*), situated on the N-terminal face of Scc3 (Scc3$_N$). Because it was unclear whether or not Smc3's KKD loop binds another protein or merely binds DNA, we focused on what proteins might bind Scc3$_N$. Because the predictions described above excluded both Wapl (f24) and Pds5 (f54A), we tested whether Scc3$_N$ might bind Smc3's ATPase domain. Remarkably, AF predicted with high confidence, in the top four ranked models, an interaction between the face of Scc3$_N$ containing D189 and E202 and the part of Smc3's ATPase containing K112 and K113 (*Figure 32A*, f56A). Not only do both Smc3(K112) and Smc3(K113) participate in the interaction but so does Smc3(S75), whose substitution by R abrogates RA in yeast (*Rowland et al., 2009*). Again, the striking congruence between the predicted interaction and residues known to be essential for RA implies that the interaction is of physiological importance. We therefore presume that Scc3(D189E) and E202K abrogate RA because they compromise the interaction between Scc3$_N$ and Smc3's KKD loop. Because Scc3(E202K) suppresses the lethality caused by unregulated RA caused by Smc3(K112R, K113R) double mutants as well as *eco1Δ* (*Rowland et al., 2009*), we can also be confident that the AF prediction (f56A) recognizes K112 and K113 in their unmodified state. Acetylation of K112 and K113 would presumably also abrogate the interaction in vivo and given that the phenotype of Scc3(D189E)

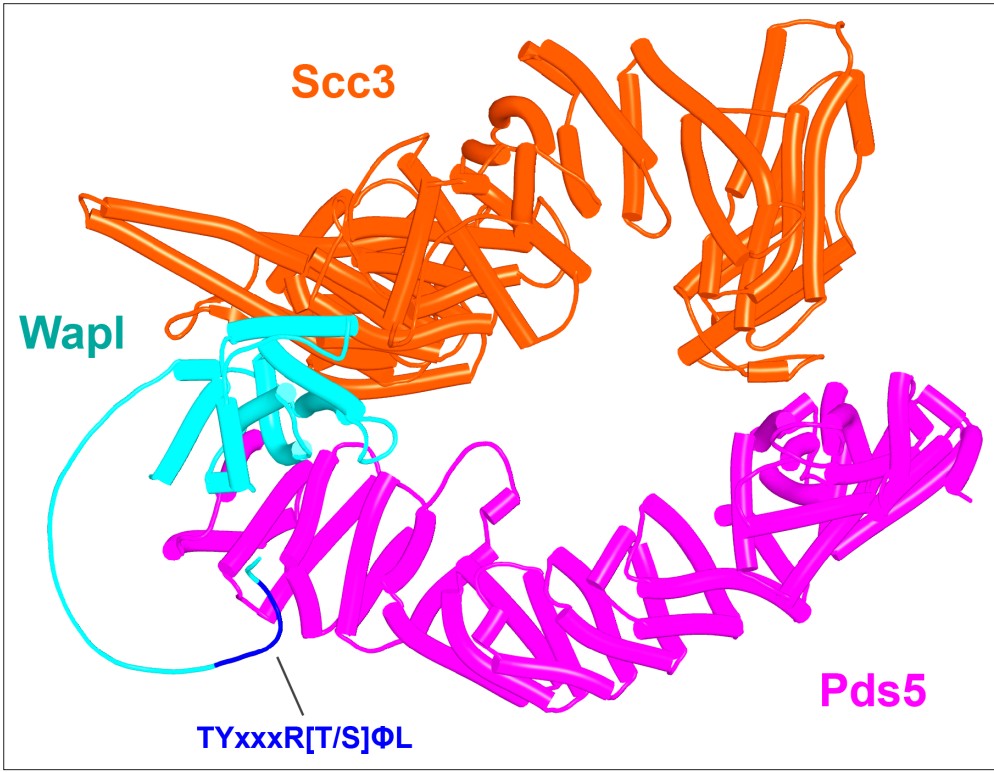

**Figure 30.** AlphaFold 2 (AF) prediction for a complex between *S. cerevisiae* Scc3, Pds5, and Wapl's CTD with its associated TYxxxR[T/S]ΦL motif (f54B). Chains in the PAEs from f54B are Wapl (A), Pds5 (B), and Scc3 (C).

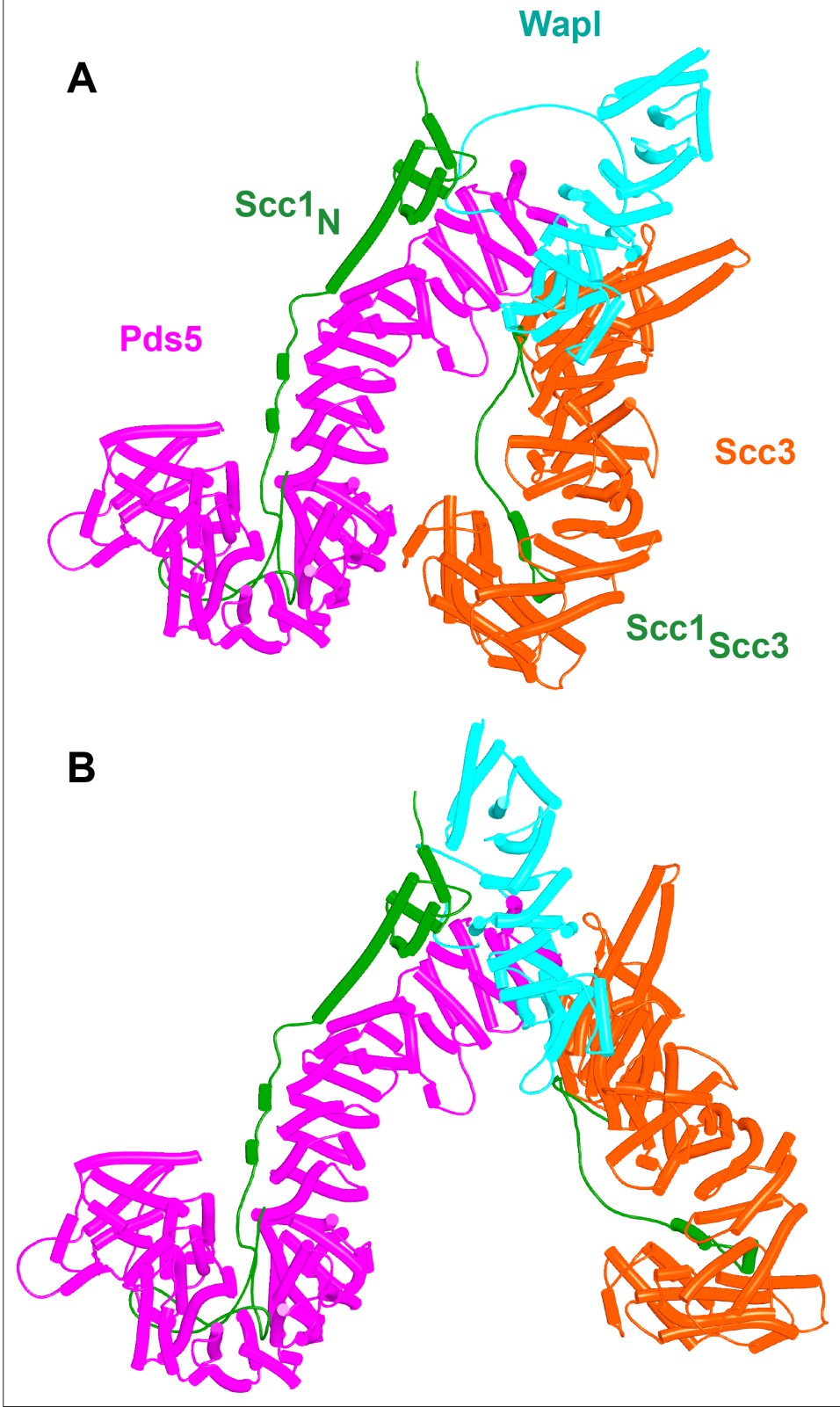

**Figure 31.** Putative quarternary complexes between yeast Scc1$_N$, Pds5, Scc3, and Wapl. (**A**) When formed with Scc3$_C$ interacting with Pds5$_P$ (f55), created by superimposing f24 and f38B onto f54B. (**B**) When formed without any interaction between Scc3$_C$ and Pds5$_P$ (f53), which involves a canonical rigid body Pds5:Scc3:Wapl ternary complex. Model created by super-imposing f24 and f38B onto f32B. A movie showing the transition between (**A** and **B**) at https://doi.org/10.6084/m9.figshare.22567525.v1.

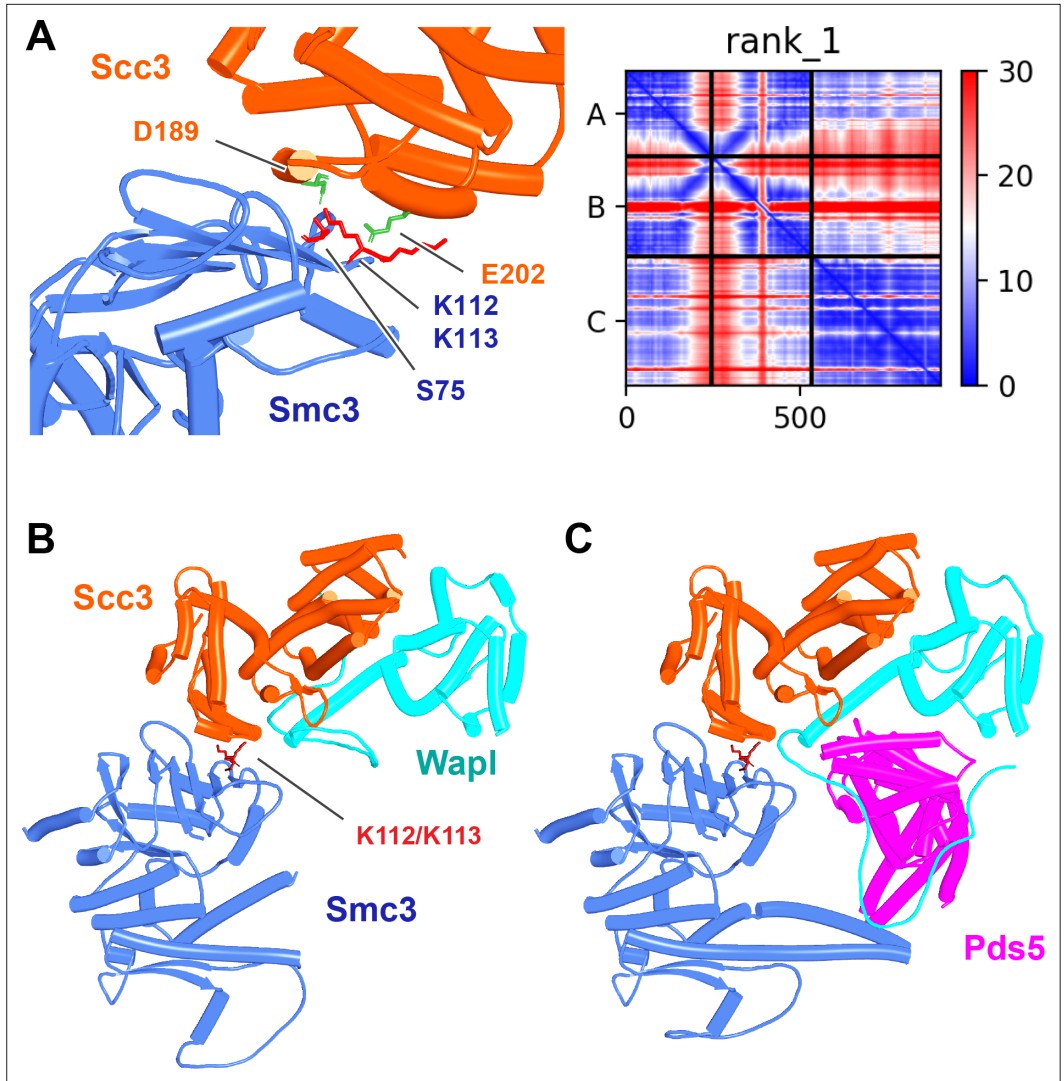

**Figure 32.** Interaction between Smc3's ATPase and Scc3's N-terminal domain. (**A**) Left: detail of the interaction between Smc3 and Scc3 (f56A) showing Scc3(D189) and Scc3(E202) (marked green) and Smc3(S75, K112, K113) (marked red). (**A**) Right: PAEs for rank 1 model in A. Chains A and B correspond to Smc3 while C to Scc3. (**B**) Interaction between Smc3's ATPase, Scc3's N-terminal domain, and Wapl's CTD (f56B). Chains in the PAEs from f56B are Smc3 (A), Smc3 (B), Scc3 (C), and Wapl (D). (**C**) Interaction of the canonical rigid body Pds5:Scc3:Wapl ternary complex with Smc3's ATPase domain (f56C). Chains in the PAEs from f56C are Wapl (A), Pds5 (B), Scc3 (C), Smc3 (D), and Smc3 (E).

and E202K mutations imply that the interaction is necessary for RA, the AF model finally provides an explanation for how acetylation would be sufficient to block RA in yeast, without any need for Sororin.

The face of Scc3$_N$, including D189 and E202, is highly conserved in fungi and AF predicted a similar interaction between Scc3$_N$ and Smc3's ATPase domain in *Candida albicans* (f57A) but not, strangely, in *N. crassa* (f57B). D189 and E202 are not conserved in metazoa and AF also failed to detect any interaction between SA/Scc3$_N$ and Smc3's ATPase domain in humans (f57C), where the physiological role of Smc3(K105) and K106 acetylation has never been rigorously investigated.

Finally, AF predicted a very similar interaction between Scc3$_N$ and Smc3's ATPase when the former was bound by Wapl's CTD (**Figure 32B**, f56B) and even when it was part of a canonical ternary complex with Wapl and Pds5 (**Figure 32C**, f56C), implying that the sequestration of Scc1's NTD by Wapl's CTD could in principle occur when Scc3$_N$ is bound to Smc3's KKD loop. If so, Scc1$_N$ would be held at 110° from Smc3's neck, which may help prevent their re-association during the release reaction (**Figure 33**). However, it is also possible that the interaction between Scc3$_N$ and Smc3 has a role at some other

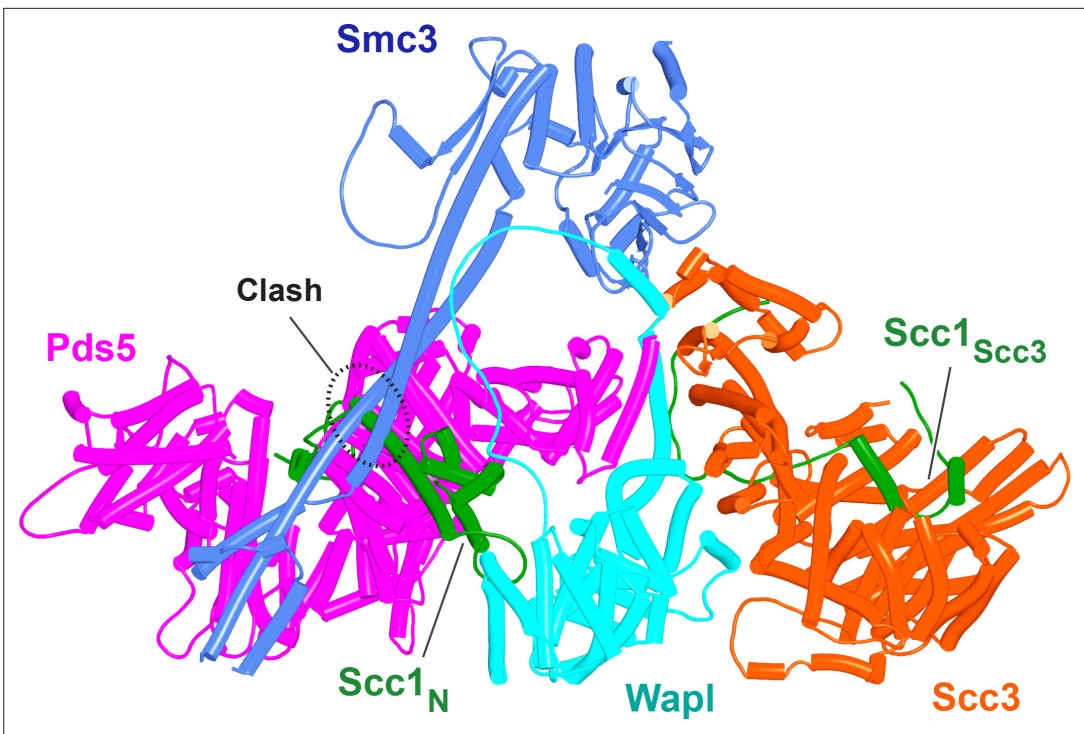

**Figure 33.** A model for the yeast quarternary Pds5:Scc3:Wapl:Scc1 complex in the canonical configuration when its Scc3 moiety is bound to the Smc3 ATPase head as predicted by AlphaFold 2 (AF), created by super-imposing f56A with the model in *Figure 31B*. There is a modest clash between Scc1 and the Smc3 coiled coil that would presumably have been avoided if AF were able to predict the entire complex, which is beyond its capability.

stage of the release process. Furthermore, it is presently unclear how the complex portrayed in *Figure 31* facilitates cohesin's release from DNA. If the latter is entrapped within the Smc-kleisin ring, then it must pass through the opened $Scc1_N$:Smc3 interface, which is hard to envisage if Scc3 remains bound at this stage to Smc3's KKD loop. Though essential, the binding of $Scc3_N$ to Smc3's KKD loop is not necessarily the latter's sole function during release. For instance, K112 and K113 might at some other stage during the reaction facilitate the binding of DNA destined to be released. Interestingly, there would be sufficient space between Smc1 and Smc3 coiled coils for DNA to bind to engaged heads even when this is accompanied by junction of Smc1 and Smc3 joints, which we suggest here may trigger dissociation of Scc1's NTD from Smc3. Importantly, Smc3(K112) and K113 would be in an ideal position to contribute to such DNA binding.

## Regulation of release by Smc3 acetylation

Acetylation of Smc3(K112) and K113 by Eco1 is essential for blocking RA in G2 cells, a process dependent on Pds5 (*Chan et al., 2013*). How does Eco1 recognize Smc3's KKD loop and how might Pds5 facilitate this? We therefore asked whether AF can predict how Eco1 binds to Smc3, Pds5, or other cohesin subunits. In two (ranks 1 and 2) out of five models with *S. cerevisiae* (f67) and in five out of five models with *Candida glabrata* (f68), AF predicts with high confidence an interaction between yeast Eco1 and Smc3's ATPase with Scc1's NTD bound to the coiled coils emerging from it (*Figure 34A*). Importantly, Eco1 binds in a manner in which Smc3(K113) would enter the enzyme's catalytic site as observed in a crystal structure of a K106-CoA conjugate bound to xEco2 from *Xenopus laevis* (PDB: 5N22) (*Chao et al., 2017*) (f69) (*Figure 34B*, left). The prediction envisions a very different type of association to that previously proposed on the basis of attempts to dock the acetyl transferase onto Smc3 heads (*Chao et al., 2017*). AF predicts, again with high confidence, a very similar mode of association between Smc3 and Esco1's catalytic domain (f70), either in the absence or presence of Scc1's NTD, and a similar mode of interaction was predicted in the top ranked model for Esco2 (f71).

Passage through S phase is necessary for acetylation by Eco1 (*Beckouët et al., 2010*; *Rolef Ben-Shahar et al., 2008*) and Esco2 (*Ivanov et al., 2018*) and likely also stimulates that by Esco1. S

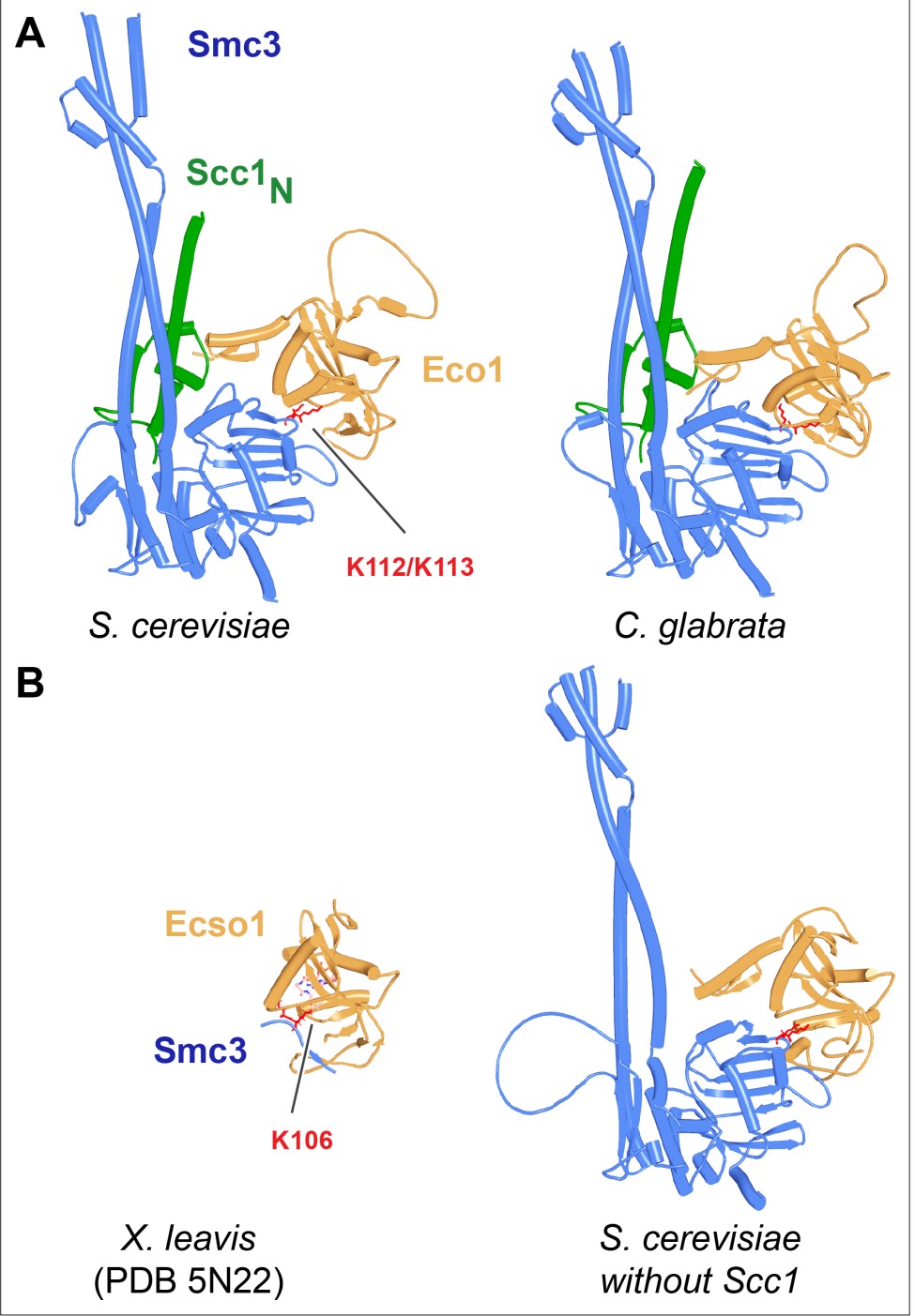

**Figure 34.** Interaction between Eco1/Esco1 and Smc3. (**A**) AlphaFold 2 (AF) predictions for Smc3's interaction with Eco1 from *S. cerevisiae* (f67) and *C. glabrata* (f68) with the Scc1 NTD present. (**B**) Left: similarity of this interaction with a crystal structure (PDB: 5N22) of Esco1 with a K106-CoA conjugate. A superimposition of f67 and 5N22 is shown in f69. Right: as in (**A**), left, without Scc1 NTD. Chains in the PAEs from f67A are Eco1 (A), Scc1 (B), Smc3 (C), and Smc3 (D).

phase-specific acetylation by all three enzymes requires a small motif (PIP box) within unstructured sequences immediately N-terminal to their zinc fingers that bind to PCNA (*Moldovan et al., 2006*), and this association was confirmed by AF (f72, f73, and f74).

In addition to predicting binding of Smc3 to the catalytic cleft on the front face of Eco1's catalytic domain, AF predicts with high confidence, in five out of five models, binding of three loops on the

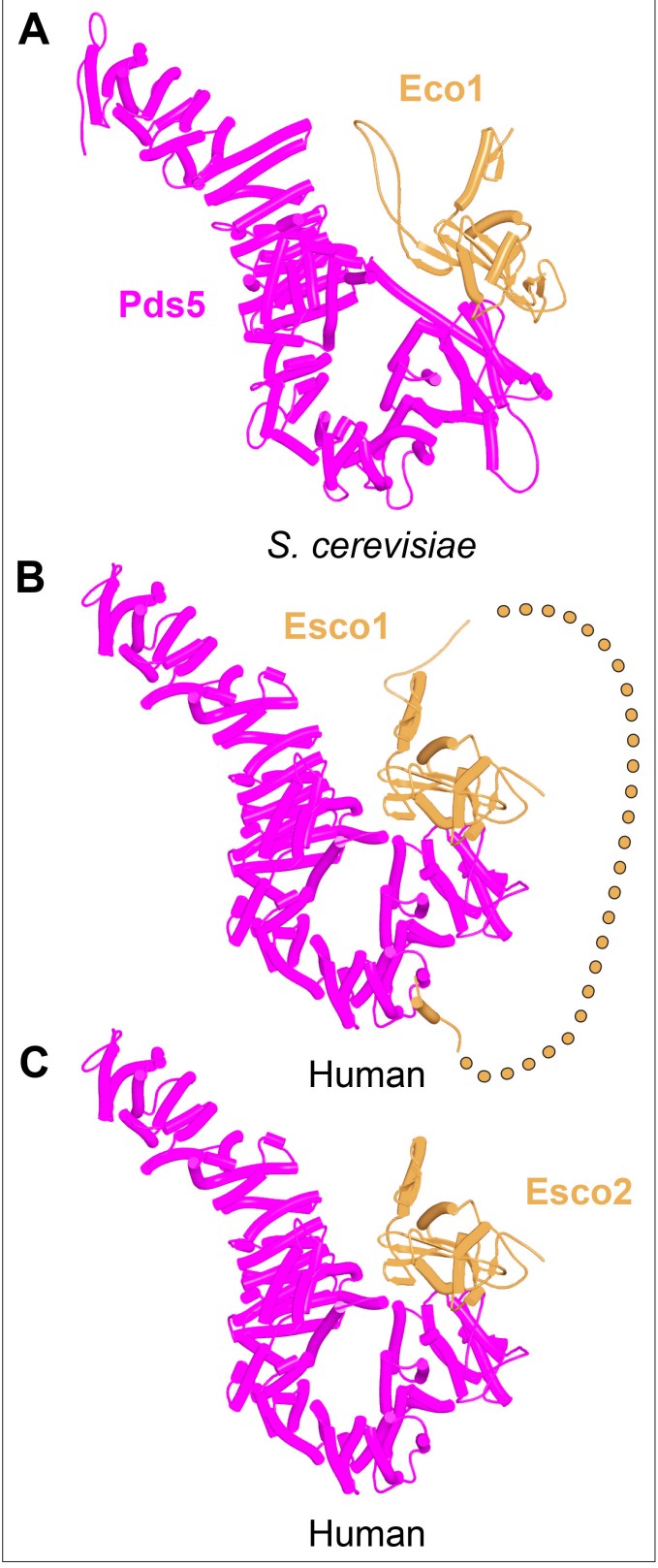

**Figure 35.** AlphaFold 2 (AF) predictions for interactions between Pds5 and *S. cerevisiae* Eco1 (**A**, f75), human Esco1 (**B**, f22), and human Esco2 (**C**, f76).

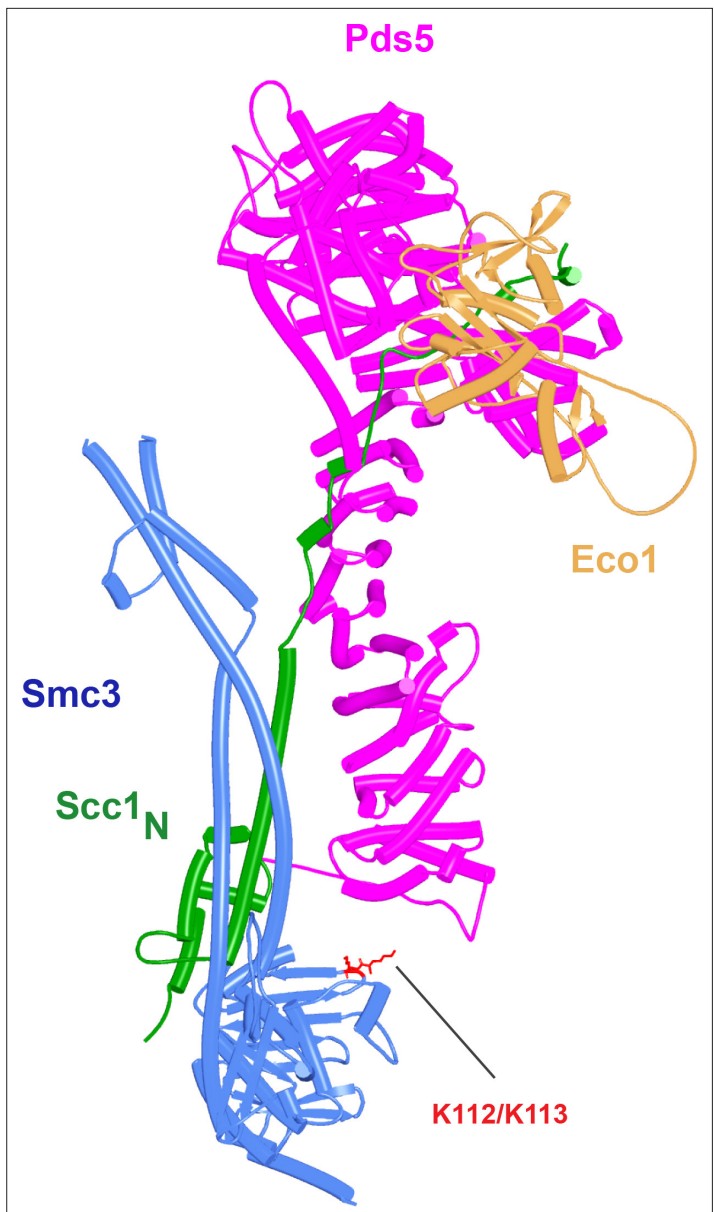

**Figure 36.** Projected position of Eco1 bound to Pds5 if Scc1$_N$ bound to the latter adopts the orientation predicted by AlphaFold 2 (AF) (f36). The model was created by superimposing f36 and f75 via their Pds5 moieties. For Eco1 to bind Smc3 K112 and K113, while still bound to Pds5, the Scc1 sequences between Scc1$_N$ α3 and the part bound to Pds5 must bend by 180°, or Scc1's NTD must dissociate from Smc3.

back side of the enzyme to a conserved loop associated with *S. cerevisiae* Pds5's most C-terminal HEAT repeats (*Figure 35A*, f75). A similar but not identical type of association is predicted for Esco1 (*Figure 35B*, f22) and Esco2 (*Figure 35C*, f76), both to Pds5A (*Figure 35*) and Pds5B (not shown). In these cases, an additional contact is made by a pair of antiparallel β sheets within the catalytic domain with the α helices that protrude from the center of Pds5A and B, which due to Pds5's hook shape are in close proximity to its C-terminal helices. Given that the association between *S. cerevisiae* Pds5's CTD with the back side of Eco1 is their sole form of interaction, it is reasonable to assume that this interaction alone recruits Eco1 to cohesin and thereby facilitates its association with the Smc3 ATPase. This has an implication. If Smc3 is acetylated by Eco1 when bound simultaneously to Pds5's CTD as well as to Scc1's NTD, then Pds5 must adopt a very different orientation, relative to the Smc3 ATPase, to that predicted by AF in the context of RA, where its NTD and not its CTD points toward the ATPase (*Figure 36*). Either the segment of Scc1 linking Scc1's NTD to Smc3 is sufficiently flexible for Pds5 to

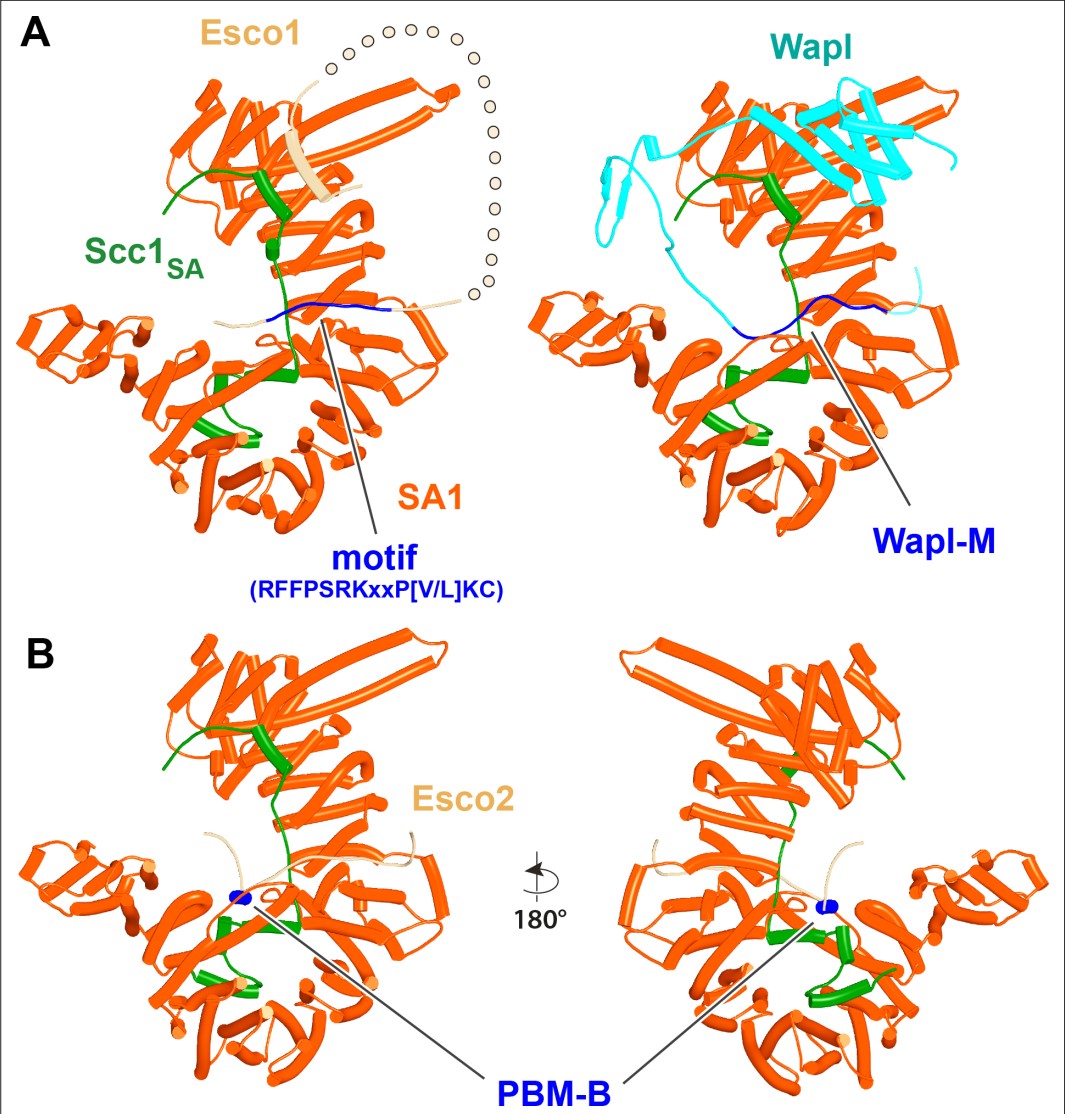

**Figure 37.** Interaction of Esco1 and Esco1 with SA1. (**A**) AlphaFold 2 (AF) predictions for the interactions between SA1:Scc1 and the N-terminal extension of Esco1 (left, f77) and between SA2:Scc1 and Wapl-M (right, f25). (**B**) AF prediction for the interaction between SA2:Scc1 and Esco2's PBM-B (f78). Chains in the PAEs from f77 are Scc1 (A), SA (B), and Esco1 (C). Chains in the PAEs from f78 are Esco2 (A), Scc1 (B), and SA (C).

adopt more than one orientation relative to the Smc3 ATPase or acetylation promoted by Pds5 would have to take place after Scc1's NTD has dissociated from the Smc3 neck.

Unlike yeast Eco1, Esco1 and Esco2 have long unstructured N-terminal extensions and AF predicts numerous interactions involving specific motifs within these, not only with various cohesin subunits (*Figures 13 and 37*), but also with the MCM helicase (*Figure 38*). Thus, in addition to the interaction between Esco1's catalytic domain and Pds5's CTD, AF predicts with high confidence and in five out five models, interactions of a highly conserved, partly helical, motif (ILxLCEEIAGEIESD) within Esco1 with the WEST site on both Pds5A (f22) and Pds5B, namely the very same site bound by Wapl's central FGF motif (*Figure 13*). Wapl and Esco1 have opposing effects on release and presumably compete for binding.

The theme of competition between Wapl and Esco1/2 is also observed with SA1:Scc1, where AF predicts, albeit with less confidence, that the aforementioned ILxLCEEIAGEIESD helical motif from Esco1 predicted to bind Pds5's WEST has an affinity also to SA1's CES, binding orthogonal to SA1's HEAT repeats, in a manner similar to the first helix from Wapl's CTD (*Figure 37A*, f77). More importantly, AF predicts that yet another conserved motif (RFFPSRKxxP[V/L]KC) binds the same

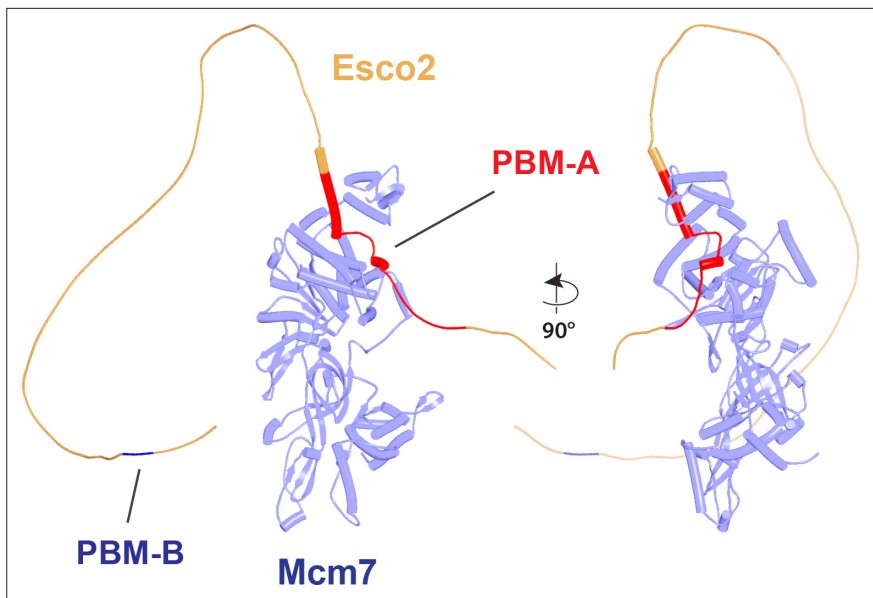

**Figure 38.** AlphaFold 2 (AF) prediction for the interaction of Esco2's PBM-A and Mcm7 (f80).

groove within the central section of SA1 bound by Wapl (*Figure 37A*, f77). Like the latter, binding extends across the Scc1 polypeptide. Despite binding to the same locus, the motif has only modest resemblance to the one used by Wapl. Interestingly, a largely unrelated but broadly conserved motif within Esco2's N-terminal domain is predicted to bind in a similar fashion to the equivalent locus on a SA2:Scc1 complex (*Figure 37B*, f78). This is adjacent to an even more conserved sequence, the PBM-B motif (GAAFF) required for the recruitment of *Xenopus* Eco2 (xEco2) to chromatin in a pre-RC dependent manner (*Higashi et al., 2012*), subsequently postulated to bind cohesin (*Ivanov et al., 2018*), and predicted by AF to bind as a very short helix to SA2's C-terminal HEAT repeats (*Figure 37B*). A second conserved motif (PBM-A) is required for xEco2's recruitment to chromatin and for its ability to acetylate Smc3 (*Higashi et al., 2012*). This was subsequently shown by co-immunoprecipitation to bind the MCM helicase. Because analysis of amine-reactive chemical crosslinking using mass spectrometry indicated Mcm4 or Mcm7 as the most likely partners (*Ivanov et al., 2018*), we used AF to investigate Esco2's interaction with each of these subunits. No interaction was observed with Mcm4 (f79), but three out of five AF models predicted with high confidence an interaction between PBM-A and Mcm7 (*Figure 38*, f80). The motif is predicted to form two short helices connected by a conserved loop. The first of these binds to the surface of Mcm7's CTD while part of the loop together with the second helix is sandwiched between the CTD and a C-terminal winged helical domain that binds the surface of ORC1 and Cdc6 within the pre-RC (*Zhai et al., 2017*). This interaction together with its PIP box may impart at least some of the S phase specificity of Smc3 acetylation by Esco2.

## Regulation of release by Sororin

Though Smc3 acetylation may be sufficient to protect sister chromatid cohesion from Wapl-mediated release in yeast, it is necessary (*Nishiyama et al., 2010*) but not sufficient in vertebrate, insect, and plant cells, where proteins called Sororins (vertebrates and plants) or Dalmatians (insects) (*Mota et al., 2022*; *Rankin et al., 2005*; *Yamada et al., 2017*) are also required. Sororin is absent from G1 mammalian cells because it is degraded at the hands of APC[Cdh1] and therefore only accumulates shortly before S phase. However, accumulation alone is insufficient for its interaction with chromosomal cohesin as its association also depends on DNA replication in *Xenopus* extracts that lack Cdh1 (*Nishiyama et al., 2010*). Association with cohesin only takes place when Smc3 is acetylated in an S phase-specific manner, largely by Esco2, and on DNA that has actually been replicated, whereupon it is essential for maintaining cohesion between sisters (*Ladurner et al., 2016*). Sororin is thought to bind to and protect from Wapl only cohesin engaged in cohesion, but whether this is really the case is not known. Importantly, though Smc3 acetylation is necessary for Sororin's association, it is insufficient and some other change in the state of cohesin that occurs during replication may be necessary. Sororins are

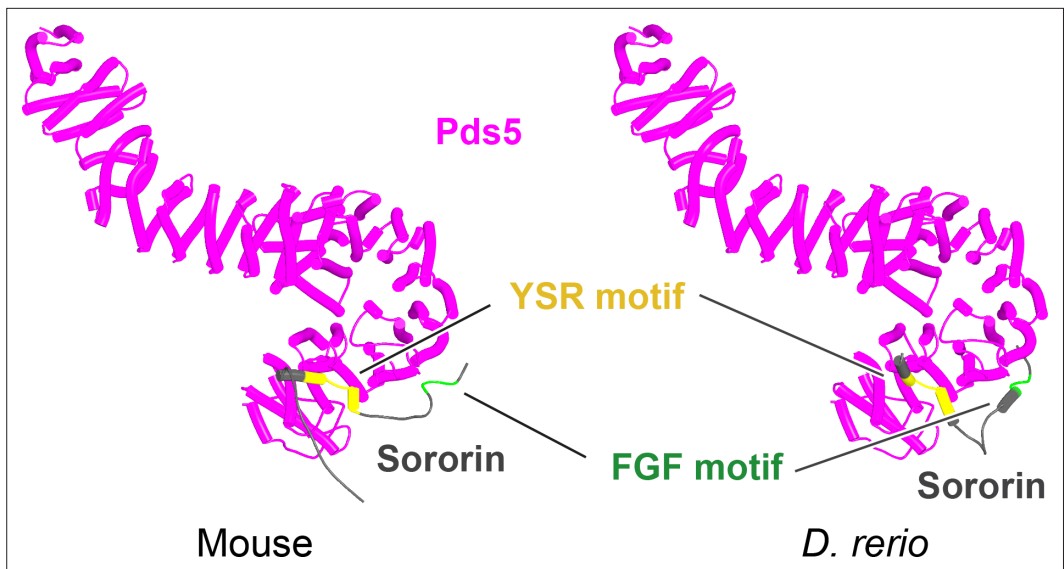

**Figure 39.** AlphaFold 2 (AF) predictions for Sororin's interaction with Pds5B in mouse (f82) and zebrafish (f21).

largely unstructured and for the most part poorly conserved. Common to all members of the family are C-terminal so called Sororin boxes predicted to contain a short (six turns) α helix and whose consensus in animals is [W/F/Y]xxxΦNxxFxE[A/Φ][E/D]x[F/Y]ELxΦE.

Three other features are common to vertebrate Sororins, namely a short stretch of basic amino acids immediately N-terminal to the Sororin box, which is preceded by a FGF motif (consensus [S/T]ΦFGFExL), separated by 10 residues from a so-called YSR motif (consensus [M/W]S[K/Q]KVRRSYSRL). Functional studies indicate that the Sororin box is essential, while the YSR and FGF motifs are not, at least when the protein is overexpressed (*Wu et al., 2011*). Because the cohesion defects associated with sororin inactivation are largely suppressed by Wapl mutants (*Ladurner et al., 2016*; *Nishiyama et al., 2010*), Sororin is believed to be required merely to suppress Wapl-mediated release, a function it fulfills despite a residence time on cohesin of about 60 s (*Ladurner et al., 2016*). It could perform this function either by inhibiting Wapl's sequestration of Scc1's NTD or by preventing the latter's dissociation from Smc3 in the first place. Though AF cannot address why Sororin only associates with replicated DNA, it has the potential to reveal which cohesin subunits interact with its essential Sororin box and/or FGF and YSR motifs. Accordingly, we used AF to predict the interaction of full-length mouse Sororin with cohesin subunits known to be involved in Wapl-mediated RA. AF predicted a potential interaction between Sororin's FGF motif and the CES of an SA:Scc1 complex, albeit in only one out five models (f81) and not in the same manner as CTCF's YDF motif (*Li et al., 2020*). In contrast, it predicted with high confidence an interaction between its YSR and FGF motifs and Pds5B's C-terminal domain (*Figure 39*), both in mice (f82) and in zebrafish (f21). Like Wapl, Sororin's FGF was predicted to bind to Pds5's WEST locus while the YSR bound to an adjacent (more C-terminal) pair of helices. In both cases, the highly conserved aromatic residues in each motif were inserted into grooves between adjacent HEAT repeats and, notably, all five models predicted simultaneous association. Crucially, none of them predicted binding of the YSR to Pds5's N-terminal APDAP loop, which was initially surprising as it has been shown to compete with the binding of Wapl's YSR to this site (*Ouyang et al., 2016*). The reason for this becomes apparent when one takes into account that Sororin's YSR and FGF motifs are separated by no more than 15 amino acids, which is too short to permit simultaneous binding of the YSR and FGF motifs to Pds5's N-terminal APDAP loop and C-terminal WEST, respectively.

Given that AF failed to predict any association between the C-terminal Sororin box and either SA2, Pds5B, or indeed Wapl itself (f83), we considered whether it might instead bind to the Smc3:Scc1 interface. Remarkably, AF predicted with high confidence in two out of five human models interaction with an Smc3 ATPase domain whose coiled coils were bound by Scc1's NTD (*Figure 40*, f84). A very similar interaction was observed with high confidence in five out of five models using orthologous proteins from *X. laevis* (f85), *Danio rerio* (zebrafish) (f86), *D. melanogaster* (f87), *Tribolium casaneum*

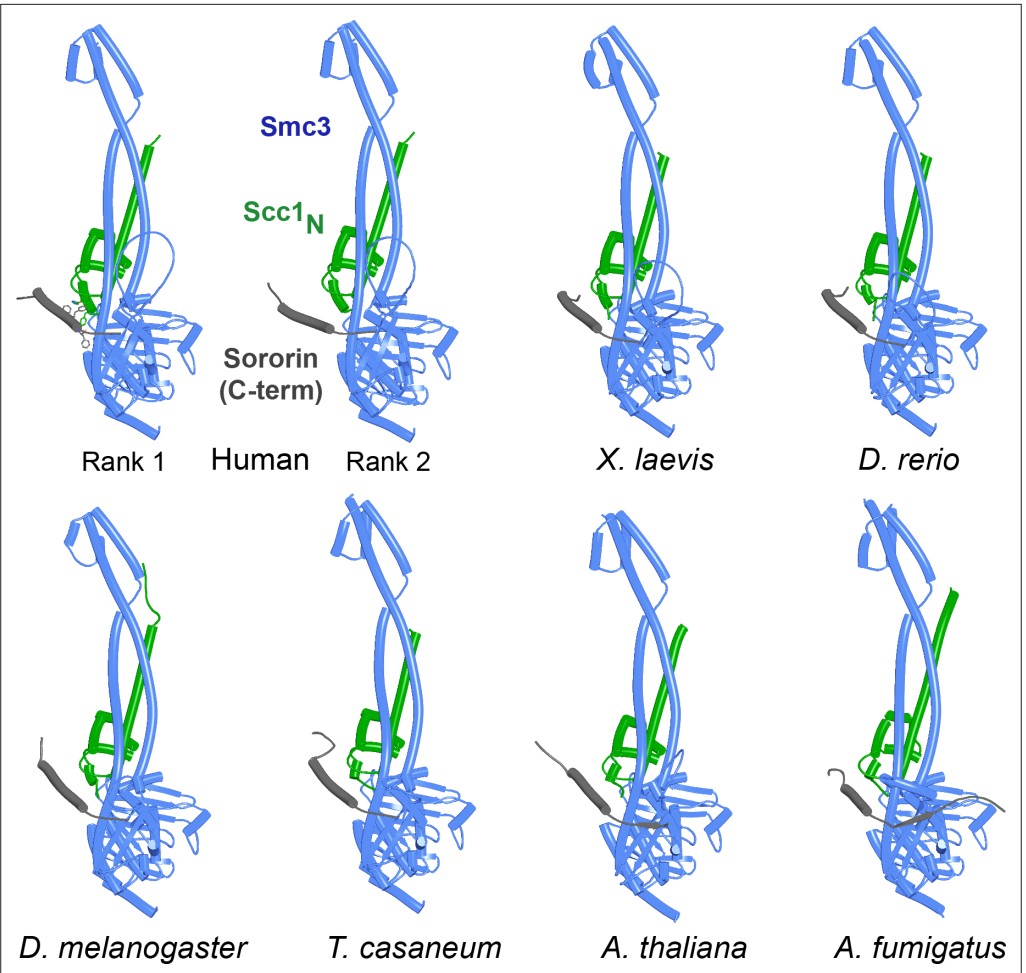

**Figure 40.** AlphaFold 2 (AF) predictions for Sororin's interaction with Smc3 bound to Scc1's NTD in humans (f84), *X. laevis* (f85), *D. rerio* (f86), *D. melanogaster* (f87), *T. casaneum* (f88), *A. thaliana* (89), and *A. fumigatus* (f90). Chains in the PAEs from f84 are Sororin (A), Smc3 (B), Smc3 (C), and Scc1 (D).

(red flour beetle) (f88), *A. thaliana* (f89), and even the ascomycete *Aspergillus fumigatus* (f90). The similarity of the predictions is remarkable given that the Sororin orthologs of vertebrates, insects, plants, and fungi have very little sequence homology and even exhibit considerable variation between their Sororin boxes (*Mota et al., 2022*). In contrast, no interaction was predicted between Sororin and a complex between Smc1 and Scc1's CTD in *D. melanogaster* (*Figure 41*, f91), *A. thaliana* (*Figure 41*, f92), and *A. fumigata* (*Figure 41*, f93). Intriguingly, AF suggests that Sororin's C-terminal helix might interact with the longer of the two helices within Scc1's CTD in mammals (*Figure 41*, f94). Thus, AF's prediction that the Smc3:Scc1$_N$ interface is a highly conserved partner of Sororin's CTD is not shared by Smc1:Scc1$_C$.

According to all of the AF models, Sororin's C-terminal α helix straddles Scc1's most N-terminal helix and the loop connecting its fourth helix to the long α helix that forms a three-helical bundle with the Smc3 coiled coil (*Figure 40*). Meanwhile, the C-terminal end of the Sororin helix contacts the C-terminal α helix emerging from the Smc3 ATPase just before it forms a coiled coil with its partner from the N-terminal part of the protein, while the terminal 4–5 amino acids straddle a pair of adjacent β sheets. Aromatic residues play a key part in the interface. In vertebrates, a conserved tryptophan (W233 in humans) sits on top of Scc1's most N-terminal helix, while a pair of highly conserved phenylalanines (F241 and F247) interact with the aforementioned Smc3 α helix, respectively (*Figure 40*, human rank 1, f84). The nature of Sororin's interaction with the Smc3:Scc1 interface is not only universal across eukaryotic kingdoms but also helps explain how Sororin hinders release. By binding Scc1's N-terminal helix, Sororin would prevent its unfolding from Scc1$_N$'s long α3 helix,

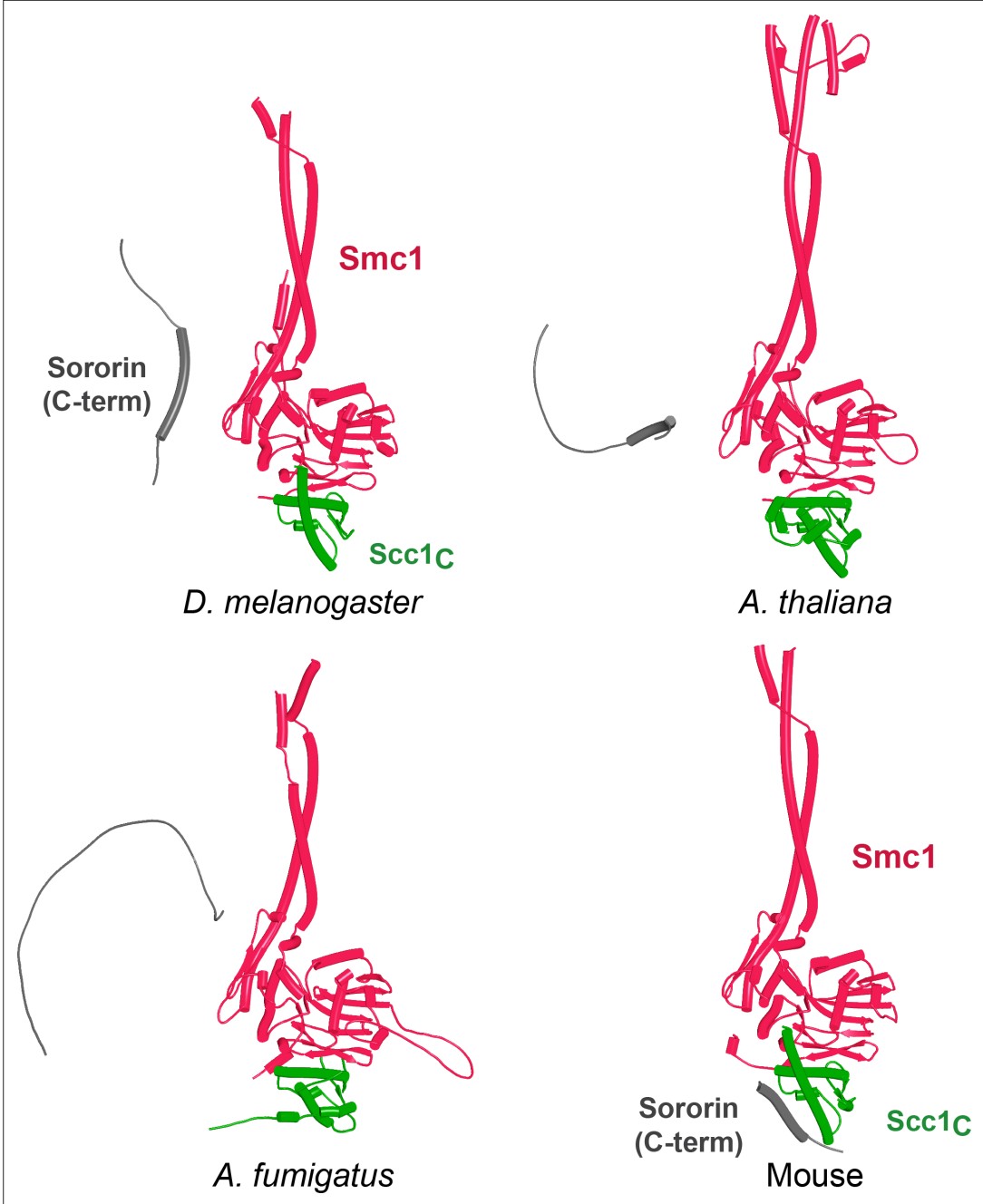

**Figure 41.** AlphaFold 2 (AF) predictions for the Sororin box's interaction with a complex between Smc1 and Scc1's CTD in *D. melanogaster* (f91), *A. thaliana* (f92), *A. fumigatus* (f93), and mouse (f94).

thereby preventing sequestration occurring even before the NTD has dissociated from Smc3. Perhaps more importantly, by binding over Scc1's N-terminal helices and simultaneously binding Smc3, Sororin could actually stabilize the interaction between Scc1's NTD and the Smc3 neck, thereby thwarting any subsequent sequestration of its N-terminal helices by Wapl. Such stabilization might also prevent the distortions in Smc3's neck produced by simultaneous head engagement and joint junction, thereby preventing such an event. Despite these insights, AF predictions provided no clues with regards to the role of Smc3 acetylation.

Finally, we tested whether Sororin might also bind Wapl's CTD. Remarkably, AF predicted with high confidence in five out five models that Sororin's [S/T]ΦFGFExL motif would bind to exactly the same cleft within Wapl's CTD that sequesters the most N-terminal α helix from Scc1 and does

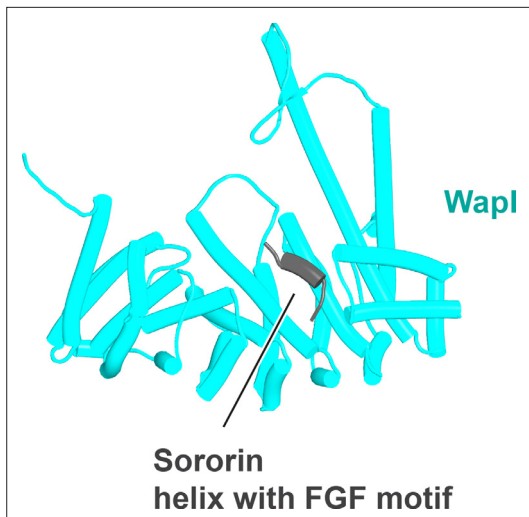

**Figure 42.** AlphaFold 2 (AF) prediction for the interaction between Sororin and Wapl's CTD (f83).

so in a manner that would be mutually exclusive (*Figure 42*, f83). As already mentioned, this motif is also predicted to bind Pds5's WEST (*Figure 13B*, f21 and *Figure 39*, f82) and by doing so ensures that Sororin's [M/W]S[K/Q]KVRRSYSRL motif binds to an adjacent groove in Pds5's CTD (*Figure 39*, f82) and not to its N-terminal APDAP loop. However, if Sororin's [S/T]ΦFGFExL motif was instead bound to Wapl's CTD (via the latter's proposed Scc1$_N$ sequestration cleft), when part of the canonical ternary RA complex with SA and Pds5, then the [K/Q]KVRRSYSRL motif would be free to bind to Pds5's APDAP, which is brought into close proximity to the aforementioned cleft in Wapl's CTD by ternary complex formation. In other words, if Sororin's [S/T]ΦFGFExL and [M/W]S[K/Q]KVRRSYSRL motifs are to bind cohesin simultaneously, the short linker connecting them will constrain the distance allowed between their binding sites. If the [S/T]ΦFGFExL motif binds to Pds5, its partner must also bind to an adjacent site on Pds5 (*Figure 39*) but if [S/T]ΦFGFExL instead binds to Wapl's CTD, its [M/W]S[K/Q]KVRRSYSRL partner could in principle bind simultaneously to Pds5's APDAP loop.

To test this, we asked AF to predict the interaction between Sororin and an N-terminal fragment of Pds5 that lacked its WEST [S/T]ΦFGFExL binding site. In this case, AF predicted with high confidence and in five out of five models an interaction exclusively between Sororin's Dxx[M/W]S[K/Q]KVRRSYSRL motif and Pds5's APDAP (*Figure 43*, left, f95). In this case, the interaction involved conserved aspartic acid (D135) and methionine (M138) residues at the N-terminal end of the motif, which are predicted to be part of a short helix (marked in gray in *Figure 43*), as well as the YSR sub-motif at its C-terminal end, whose interaction with Pds5's APDAP is predicted by AF to be identical to that observed in a

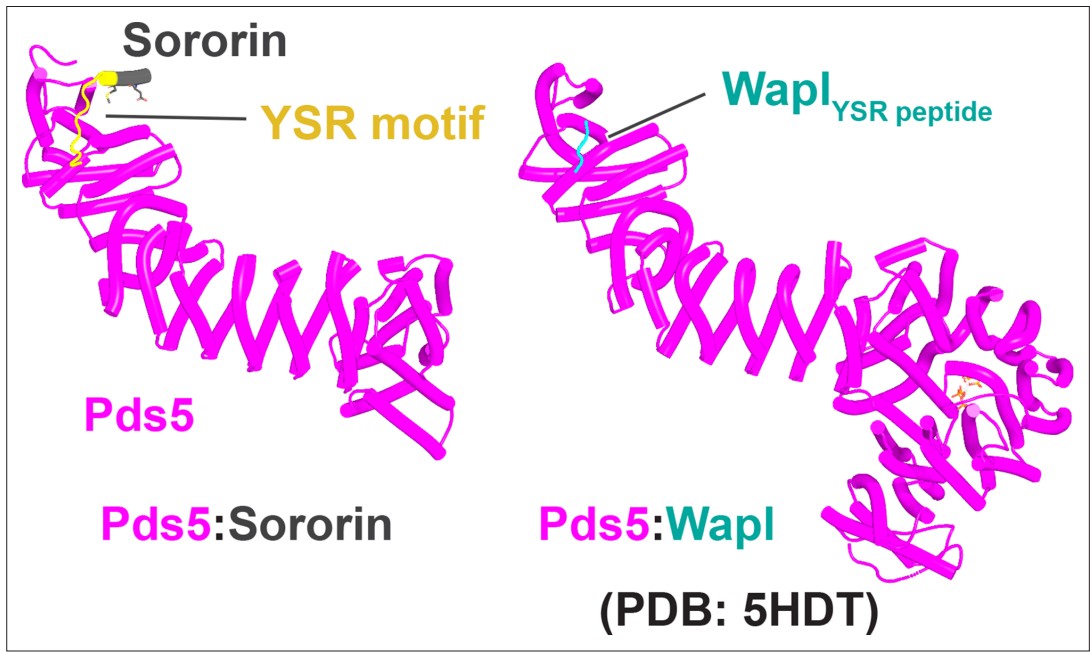

**Figure 43.** AlphaFold 2 (AF) predicts that Sororin's [M/W]S[K/Q]KVRRSYSRL (YSR) motif binds to Pds5's APDAP loop (f95) (left) in a manner similar to Wapl's YSR peptide (right, PDB 5HDT), but only if binding of Sororin's [S/T] ΦFGFExL motif is prevented from binding Pds5's WEST site by deleting the C-terminal part of Pds5 containing this WEST site.

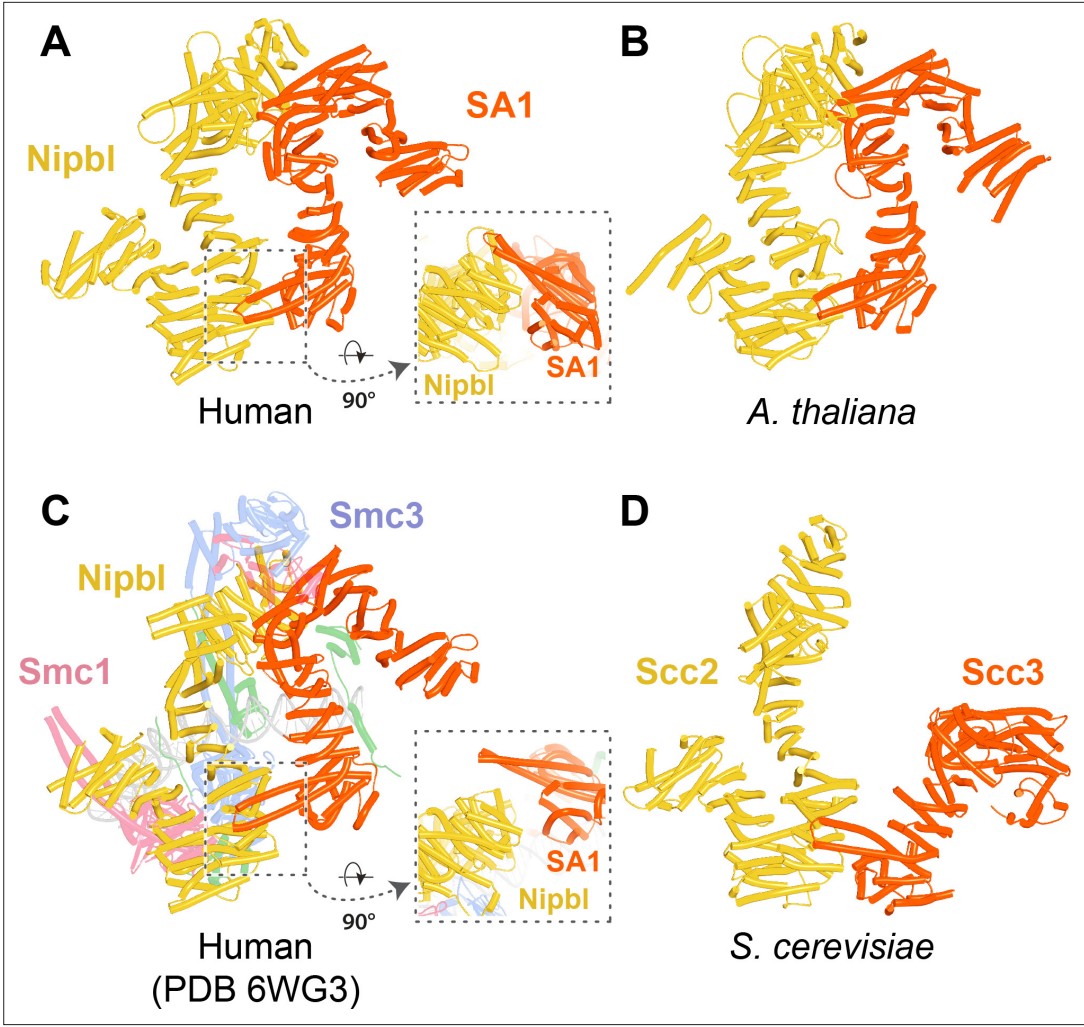

**Figure 44.** AlphaFold 2 (AF) predictions for interactions between Nipbl/Scc2 and SA/Scc3 for humans (**A**, f96), A. thaliana (**B**, f97),and S. cerevisiae (**D**, f98); (**C**) cryo-EM structure PDB 6WG3. Chains in the PAEs from f96 are Nipbl (A) and SA (B). Chains in the PAEs from f98 are Scc2 (A)and Scc3 (B).

crystal structure containing Wapl's N-terminal YSR motif bound to Pds5 (*Ouyang et al., 2016*; PDB 5HDT; *Figure 43*, right). These predictions therefore raise the possibility that Sororin may have a second line of defense against Wapl in case dissociation of Scc1's NTD from Smc3 is accompanied by formation of the canonical ternary (SA:Wapl:Pds5) RA complex, namely occupation of the key cleft in Wapl's CTD by Sororin's [S/T]ΦFGFExL motif, which would exclude sequestration of $Scc1_N$ (*Figure 42*), and be facilitated by the simultaneous binding of Sororin's [M/W]S[K/Q]KVRRSYSRL motif to the neighboring N-terminal APDAP loop of Pds5 (*Figure 43*).

## Interactions between Scc2/Nipbl and Scc3/SA

One of the key insights revealed here by AF is the canonical rigid body association between SA and Pds5 induced by their simultaneous association with Wapl's CTD (*Figure 16*). According to AF, SA and Pds5 do not otherwise interact directly and only do so in the presence of Wapl. We note, however, that the yeast orthologs do so via their $Scc3_C$:$Pds5_P$ interaction (*Figure 29*). In the course of investigating their association, we used AF to undertake a more systematic investigation of how cohesin's three HAWK subunits interact. Similar to SA/Scc3, AF also predicted no interaction between Nipbl/Scc2 and Pds5. In contrast, it predicted with high confidence and in five out five models a pair of simultaneous interactions between full-length versions of SA and Nipbl (*Figure 44A*, f96). One of these interactions is identical to that observed in a cryo-EM structure of the clamped state (*Figure 44C*, PDB 6WG3; *Shi*

*et al., 2020*) and involves interactions between Nipbl helices next to its N-terminal protrusion and the outside surface of SA where there is a twist in the alignment of its antiparallel HEAT repeats at the point where the protein starts to bend. The second region of interactions has not hitherto been observed and involves an interaction between helices at the point where Nipbl starts to bend (on its outer surface) and the antiparallel helices that make up SA's pronounced 'nose.' An identical pair of interactions were predicted for the orthologous proteins from *A. thaliana* (*Figure 44B*, f97).

In contrast, AF predicted only the nose interaction for yeast Scc2 and Scc3 (*Figure 44D*, f98). In this case, the residues involved are highly conserved on both the Scc3/SA (Scc3: Q323, E327 and SA1: Q165, E169) and Scc2/Nipbl (Scc2: P958, Y959 and Nipbl: P2029, Y2030) sides of the interface and the interaction is therefore in all probability of physiological importance. Apart from microsporidians (*Roig et al., 2014*), where it is absent, the nose is a highly conserved feature of Scc3/SA orthologs. Its deletion is lethal in yeast (*Roig et al., 2014*), but its function has hitherto remained obscure. It is interesting that the interaction between SA's nose and Nipbl was not observed in the PDB 6WG3 cryo-EM structure (*Shi et al., 2020*). This is probably because it would have been incompatible with DNA binding to Nipbl. Now that AF has revealed the details of the interaction, it will be possible to create specific mutations that affect it and thereby address whether the interaction is necessary either for LE or for DNA entrapment within S-K rings, both in vitro and in vivo.

## Discussion

Our use of AF to predict interactions between cohesin subunits has led to numerous hypotheses concerning how cohesin's association with chromatin is regulated. Can one be sufficiently confident to undertake the biochemical and genetic experiments needed to test them? And if so, what will be the most effective techniques to employ? Aside from AF's excellent track record, it so happens that in several instances experimental data, particularly the phenotypes of mutations, was already in the public domain but had hitherto not been interpretable. Thus, AF predictions about the interaction between Wapl's CTD and Scc3/SA explains why mutation of both Wapl and Scc3/SA residues predicted by AF to affect their interaction are defective in RA activity in vivo (*Beckouët et al., 2016*; *Hara et al., 2014*; *Ouyang et al., 2013*; *Rowland et al., 2009*). Likewise, the phenotypes of an extensive set of mutations affecting highly conserved surface residues within the C-terminal half of Wapl's CTD (*Ouyang et al., 2013*) are fully consistent with AF's prediction that its role is to sequester Scc1's N-terminal helices (α-1 and α0). Even more impressive is the remarkable congruence between cohesin subunit interfaces revealed by AF and the locations of mutations that abrogate RA in yeast and were isolated without any prior hypothesis, as spontaneous suppressors of *eco1* mutant lethality (*Rowland et al., 2009*). Yet further support comes from the congruence between AF predictions and sequence conservation (beyond that used by AF), which reflects natural selection and hence is also a robust measure of function. As a consequence, our analyses have been guided not only by the degree of AF's confidence but also by a requirement that predictions be broadly based from a taxonomic viewpoint, namely similar in multiple and as distant as possible eukaryotic lineages, and concern sequences and domains that are evolutionarily conserved. Despite this, AF revealed instances of physiologically important interactions specific to particular eukaryotic lineages, for example, the interactions between the N- and C-terminal faces of Scc3 with Smc3's ATPase and Pds5's protrusion, respectively, which were only detected in yeast and related fungi. Last but not least, a key guiding principle has been whether the models appear plausible and above all explanatory.

Insofar that AF largely functions by correlating the query sequences with known structures and their sequences, and not by calculating chemical or physical properties from first principles, there is reason to be optimistic that the results will be in many cases comparable in robustness to those obtained by experimental structural biology, which often uses or even requires unwarranted conditions (protein concentration, salt concentration, divalent cations, pH, temperature, etc.) and usually evaluates interactions either pairwise or among a very limited number of constituents and not as in vivo where proteins must compete with a large fraction of the proteome, as well as with other macromolecules. We contend that many if not most of the hypotheses described in this article are plausible, novel, in some cases provocative, and above all explanatory, and therefore deserve communication before validation using orthogonal approaches.

What then might these be? At the top of the list must be renewed attempts to visualize the predicted complexes by cryo-EM alongside the use of bifunctional thiol specific reagents to crosslink

both in vitro and in vivo cysteine pairs inserted into the predicted interfaces (*Chapard et al., 2019*; *Srinivasan et al., 2018*). Mutation of key residues will also have an important role, but it is important to stress that mutations alone rarely address whether the resulting physiological defects are due to the loss of a specific interaction. They merely reveal whether or not one or other interaction surface is important. To achieve this goal using mutations alone, it would be necessary to generate not just one or two mutations but a large series and demonstrate a robust structure–function relationship, or better still demonstrate that mutations in both sides of an interface produce identical phenotypes. A good example of the latter is our finding of RA-abrogating mutations identified through classical (i.e. hypothesis free) forward genetics mapping to both sides of interfaces identified by AF, for example, that between $Scc3_C$ and $Pds5_P$ and between $Scc3_N$ and Smc3's KKD loop. Combining cross-linking with a more limited mutational analysis is probably therefore the most economical way of proceeding, but even this is will be a considerable undertaking as all substitutions should be made within endogenous genes.

Though our experience is anecdotal, the modeling that we undertook for this analysis was sufficiently extensive that it is worth sharing the impressions gained into the strengths and weaknesses of AF as well as what may transpire to be best practice. First and foremost is the issue of the optimum scale for embarking on a prediction. On the one hand, AF's ability to detect patterns between specific domains clearly diminishes as the size and number of components increase. The combinatorial space increases with the square of the number of residues provided in the query sequence(s). On the other hand, predicted interactions clearly have greater value if AF manages to predict them within the context of intact and therefore possibly large proteins or protein assemblies, whose size approaches the limits of what is currently practical or even possible given GPU memory limitations. It is our experience that initial forays should be performed while making as few assumptions as possible as to the nature of the interactions and therefore giving AF as large a landscape as is practically possible. Nevertheless, following up interactions revealed in this manner also has value. It is also clear that there is probably no relationship between the confidence of AF predictions and any actual affinity between proteins, as would be expected given the AF algorithm. If biology demands a weak interaction and natural selection has delivered this, AF clearly excels in picking up the resulting patterns and in this regard it provides an approach that is complementary to conventional biochemistry and structural biology with which weak interactions are sometimes difficult to investigate. As a consequence, many of the interactions detected by AF may be challenging to pursue using these techniques and combining experimental structural work with the sort of cross-linking techniques described above may prove useful in this regard. Another important lesson, which may prove to be of general validity, is that complexes such as cohesin should not be regarded as conventional solids, which imply a lack of constituent mobility or as liquids, which would require massively larger stoichiometries. They should instead be considered as promiscuous molecular communities made up by a large set of potential interactions with a great heterogeneity of on and off rates and extensive competition between different constituents, many of which have multiple possible partners and multiple binding interfaces. Some of the interactions are between relatively rigid domains but many are between short motifs embedded in very long intrinsically disordered loops and extensions. Understanding the physical dynamics of such communities as well as the mechanisms underlying the switches between states will be a formidable challenge. In the case of cohesin, both loop extruding and cohesive forms must be capable of switching between forms that associate stably with chromatin fibers to ones that dissociate, and a competition for shared binding sites between factors promoting release or its inhibition presumably create bi-stable switches.

In summary, our AF predictions have yielded several hypotheses for how cohesin's association with chromatin is regulated. These include:

1. The role of cohesin's hinge in effecting a junction between the joints within Smc1 and Smc3's coiled coils. When combined with ATP-driven head engagement in the absence of Scc2/Nipbl, but in the presence of Pds5, this junction is predicted to create a distortion of the coiled coil to which the Scc1's NTD normally binds and thereby triggers its dissociation.

2. The proposal that Wapl's CTD forms a ternary complex with the pair of cohesin's HAWK subunits previously implicated in release, namely SA/Scc3 and Pds5. Upon dissociation from Smc3, Scc1 is predicted to bind to Pds5 in a manner that places Scc1's short N-terminal helices in close proximity to a highly conserved site within Wapl's CTD when part of the ternary complex with SA and Pds5. The subsequent unfolding of Scc1's α-1 and α0 helices from the long α3 helix that

normally associates with Smc3's neck then leads to their sequestration by this part of Wapl. We propose that sequestration in this manner is the chief function of Wapl and that it serves to keep the exit gate open long enough for DNA to pass out of the cohesin ring.

3. The finding that in yeast formation of the canonical ternary complex needed to deliver Scc1's N-terminal helices to Wapl may be a multi-stage process during which an alternative ternary complex involving the $Scc3_C:Pds5_P$ interaction is required for subsequent formation of the canonical one.

4. The first insight into how Smc3's KKD loop facilitates release, at least in *S. cerevisiae* where it binds the N-terminal face of Scc3.

5. An understanding of how the acetyl transferases that block release and compete with Wapl for binding cohesin's HAWK subunits bind to their substrate and how this is facilitated during S phase by simultaneous binding to Pds5 and to the Mcm helicase.

6. The first insight into how Sororin is recruited to cohesin and how it hinders Wapl-mediated release, through the association of a conserved C-terminal α helix with cohesin's exit gate, namely the junction of Scc1's NTD with Smc3's neck.

7. The discovery that Scc2/Nipbl and SA/Scc3 interact with each other via an association involving the latter's highly conserved nose, for which there has hitherto been no explanation. Whether this association has a role during LE or in the entrapment of DNA inside the cohesin ring remains to be addressed.

Of these, the notion that a key function of Wapl's CTD is to bind α-1 and α0 from $Scc1_N$ is probably the most remarkable as it is based not on a single AF prediction but on an elaborate concatenation of numerous models, producing a quarternary complex involving Wapl, Pds5, SA/Scc3, and Scc1, in which Scc1's N-terminal α helices are positioned next to a pair of crucial sites within Wapl's CTD. Key observations underpinning this notion include the conservation of Wapl's putative $Scc1_N$ binding site, the existence of mutations within it that abrogate release in vivo both in yeast and in mammalian cells, the prediction by AF that Pds5, SA/Scc3, and Wapl form a highly specific ternary complex in animal, plant, and fungal lineages, that $Scc1_N$'s association with the spine of Pds5's NTD places the formers' α-1 and α0 helices next to the putative binding site within Wapl's CTD, that formation of the ternary complex in fungi depends on a motif within Wapl's unstructured N-terminal sequences which binds Pds5's APDAP loop and is essential for RA in *S. cerevisiae*, AF predictions for the interaction between $Scc1_N$ and Wapl that appear conserved among a very wide variety of eukaryotes, and lastly on the fact that dissociation of $Scc1_N$ from Smc3 depends on Wapl in vivo in *S. cerevisiae* and that RA is abrogated by their co-translation. Whether like a jigsaw puzzle, our hypothesis proves to be the sole solution to these explicanda will nevertheless require numerous types of wet experiments.

Last but not least, it should be pointed out that due to the nature of AF, none of our models incorporate DNA and therefore shed no direct insight into how sequestration of Scc1N by Wapl actually facilitates cohesin's dissociation from chromosomes in vivo.

## Methods

AF predictions prior to February 2023 were performed via Google CoLab using Alphafold 2, while those performed thereafter used Colabfold v1.5.2 AF2 with MMSeqs2. Supposition of different models were performed using UCSF Chimera while all figures in this article and Video 1 (available at https://doi.org/10.6084/m9.figshare.22567525.v1) were created using PyMOL v2.5 (Schrödinger, LLC). Each subunit was drawn in all figures using the following colors: Smc1 = red; Smc3 = blue; Scc1 = green; Scc2/Nipbl = yellow; Scc3/SA1/SA2 = orange; Pds5 = violet; Wapl = cyan; Eco1/Esco1/Esco2 = ivory; Sororin = gray.

The *WAPL* deletions were introduced into the genome of *S. cerevisiae* using CRISPR-Cas9. Cells of strain KN18714 (a W303-related strain whose endogenous *WAPL* gene was tagged with EGFP) were transformed with a 738 nt (for Δ4–9) or a 138 nt (for Δ208–215) PCR 'healing fragment' along with a 2 micron *LEU2* plasmid, constitutively expressing Cas9 endonuclease and transcribing the structural RNA that contains the Cas9 binding site and the guide RNA that will target the double-strand break formation by Cas9 to the *WAPL* (*RAD61*) gene. The guide RNA was designed 20 nt upstream of the NGG (E228, V229) using the annealed oligos MB526F/MB527R (MB526F: 5'-ATCAAATGAAAT AATGGGAGAAG-3', MB527R: 5'-AACCTTCTCCCATTATTTCATTT-3') and fused afterward at the 5' end of the structural RNA with Sap1 cloning. The PCR 'healing fragment' was created using the oligos MB543F/MB525R for Δ4–9 and MB534F/MB525R for Δ208–215 (MB543F: 5'-TCGCAAAACGAA

ACCATCTTCTTACCCTAAAGCATCCTGTTTCTGAAAAAATGAGAGCAGTTCTCAGAACTCCTTTTAGA
TCTAATAAAGGCTTGCCATC-3', MB525R: 5'- TTACTGTTGTCAGCGTTATTTAGGGACAAGGTGTTA
GCTTTCTGATCTACTTCTTCTCCCATTATTTCATTTTC-3', MB534F: 5'-GCATACGCACATCCAAAGAC
AAAAAAATAAACAAAAATAAAGAAAATG-3') and contained the deletion, a mutagenized NGG Cas9
sequence target (E228, V229: GAGGTA to GAAGTA) and 5'-, 3'- homology arms needed for repair of
the double-strand break created by Cas9. As a control, no colonies were obtained after yeast trans-
formation of the 2 micron *LEU2* plasmid alone without the 'healing fragment.' All mutations were
confirmed by sequencing the genomic locus.

The ability of the EGFP-tagged *WAPL* mutations to suppress *eco1Δ* lethality was tested by crossing
them with strain KN16432, which carries deletions of both *ECO1* and *WAPL*. Dissection of 18 tetrads
from crosses involving *WAPLΔ208–215* yielded 60 viable spores, of which 27 carried *eco1Δ*. Of these,
15 carried the *waplΔ* and 12 the *WAPLΔ208–215* mutation. Thus, *WAPL Δ208–215* was as effective as
*waplΔ* in suppressing *eco1Δ* lethality and this allele is therefore defective in RA. In contrast, dissection
of 18 tetrads from crosses involving *WAPLΔ4–9* yielded 49 viable spores of which 16 carried *eco1Δ*.
Of these, all 16 carried *waplΔ*. Thus, *WAPLΔ4–9* cannot suppress *eco1Δ* lethality and is therefore
proficient for RA. Separate experiments demonstrated that *WAPLΔ208–215* suppressed *eco1Δ* when
untagged or tagged with PK9 while *WAPLΔ4–9* never did so. Mutation of both motifs (*Δ4–9 Δ208–
215*) also suppressed *eco1Δ* lethality as did an allele in which the TYxxxR[T/S]ΦL motif (YGKKRTIL) was
inserted at *Δ4–9* (creating the sequence MRAYGKKRTILVLR) within *Δ4–9 Δ208–215*. In other words,
the TYxxxR[T/S]ΦL motif does not appear to function when placed at Wapl's N-terminus, namely
when further from its CTD.

## Yeast strains

| Strain number | Genotype |
| --- | --- |
| KN18714 | *Mat a, Wapl-EGFP::His* |
| KN24262 | *Mat a, wapl(Δ4–9)-EGFP::His* |
| KN24060 | *Mat a, wapl(Δ208–215)-EGFP::His* |
| KN24224 | *Mat a, wapl(Δ208–215)-EGFP::His, eco1::NatMx* |
| KN16432 | *Mat alpha, eco1::NatMx, wapl::HphMx* |
| KN26163 | *Mat a, wapl(Δ208–215), eco1::NatMx* |
| KN25817 | *Mat a, wapl(Δ208–215)-PK9::NatMX, eco1::HphMx* |
| KN25168 | *Mat alpha, eco1::HphMx, leu2::Eco1p-Eco1::LEU2* |

## Acknowledgements

KAN would like to thank Madhusudhan Srinivasan (Oxford, UK) for numerous discussions that stimu-
lated and informed this work. This work was supported by the Medical Research Council (U105184326
to JL) and Cancer Research UK (26747 to KAN).

## Additional information

### Funding

| Funder | Grant reference number | Author |
| --- | --- | --- |
| Medical Research Council | U105184326 | Jan Löwe |
| Cancer Research UK | 26747 | Kim A Nasmyth |

| Funder | Grant reference number | Author |
|---|---|---|

The funders had no role in study design, data collection and interpretation, or the decision to submit the work for publication.

## Author contributions

Kim A Nasmyth, Conceptualization, Investigation, Writing – original draft, Writing – review and editing; Byung-Gil Lee, Visualization; Maurici Brunet Roig, Investigation; Jan Löwe, Data curation, Supervision, Writing – review and editing

## Author ORCIDs

Kim A Nasmyth ⓘ https://orcid.org/0000-0001-7030-4403
Byung-Gil Lee ⓘ http://orcid.org/0000-0001-9565-6114
Jan Löwe ⓘ https://orcid.org/0000-0002-5218-6615

Reviewer #1 (Public Review): https://doi.org/10.7554/eLife.88656.4.sa1
Reviewer #2 (Public Review): https://doi.org/10.7554/eLife.88656.4.sa2
Author Response https://doi.org/10.7554/eLife.88656.4.sa3

# Additional files

## Supplementary files

• MDAR checklist

## Data availability

All data files are available at https://doi.org/10.6084/m9.figshare.22567318.v2.

The following datasets were generated:

| Author(s) | Year | Dataset title | Dataset URL | Database and Identifier |
|---|---|---|---|---|
| Nasmyth KA | 2023 | What Alphafold tells us about cohesin's retention and release from chromosomes | https://doi.org/10.6084/m9.figshare.22567318.v2 | figshare, 10.6084/m9.figshare.22567318.v2 |
| Nasmyth K, Lee B-G, Löwe J | 2023 | Movie M1 Figure 31 Nasmyth et al., 2023 | https://doi.org/10.6084/m9.figshare.22567525.v1 | figshare, 10.6084/m9.figshare.22567525.v1 |

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
