## [Editor Report · eLife assessment]

This **important** study makes use of AlphaFold2 to predict the models of tens of cohesin subcomplexes from different species. The models, which are in most cases consistent with published cohesin variants with compromised in vitro and in vivo cohesin activity, provide **convincing** evidence that leads to testable hypotheses of cohesin dynamics and regulation. More broadly, this study serves as an example of how to use AlphaFold2 to build models of protein complexes that involve the docking of flexible regions to globular domains.

---

## [Referee Report · Reviewer #1 (Public Review)]

There are a number of outstanding questions concerning how cohesin turnover on DNA is controlled by various accessory factors and how such turnover is controlled by post-translational modification. In this paper, Nasmyth et al. perform a series of AlphaFold structure predictions that aim to address several of these outstanding questions. Their structure predictions suggest that the release factor WAPL forms a ternary complex with PDS5 and SA/SCC3. This ternary complex appears to be able to bind the N-terminal end of SCC1, suggesting how formation of such a complex could stabilize an open state of the cohesin ring. Additional calculations suggest how the Eco/ESCO acetyltransferases and Sororin engage the SMC3 head domain presumably to protect against WAPL-mediated release.

This work thus demonstrates the power of AF prediction methods and how they can lead to a number of interesting and testable hypotheses that can transform our understanding of cohesin regulation. These findings require orthogonal experimental validation, but authors argue convincingly that such validation should not be a pre-requisite to publication.

In their revised version, the authors did not systematically include model confidence scores, and it therefore remains difficult for the reader to evaluate the reliability of the models obtained. The authors correctly point out that such metrics are available on figshare. It is therefore possible to obtain such information. The caveat is that it remains to the user to identify and extract the relevant information. While they claim that they have labeled N- and C-termini in their figures, no such labeling can be seen in the revised version. Addition of such labels, at least for some of the figures, would help the user to navigate the models.

The authors have now updated figure legends to indicate which protein is referred to by the chain labels shown in PAE plots.

It is exciting to see AF-multimer predictions being applied to cohesin. As some of the reported interactions are not universally conserved and some involve relatively small interfaces the possibility arises that these interfaces show poor or borderline confidence scores. As some of these interfaces map to mutants that have previously been obtained by hypothesis-free genetic screens and mutational analyses, they appear nevertheless valid. Thus, an important point to make is that even interfaces that show modest confidence scores may turn out to be valid while others may be not.

---

## [Referee Report · Reviewer #2 (Public Review)]

The ATPase protein machine cohesin shapes the genome by loop extrusion and holds sister chromatids together by topological entrapment. When executing these functions, cohesin is tightly regulated by multiple cofactors, such as Scc2/Nipbl, Pds5, Wapl, and Eco1/Esco1/2, and it undergoes dynamic conformational changes with ATP binding and hydrolysis. The mechanisms by which cohesin extrudes DNA loops and medicates siter-chromatid cohesion are still not understood. A major reason for the lack of understanding of cohesin dynamics and regulation is the failure to capture the structures of intact cohesin in different nucleotide-bound states and in complex with various regulators. So far only the ATP state cohesin bound to NIPBL and DNA have been experimentally determined.

In this manuscript, Nasmyth et al. made use of the powerful protein structure prediction tool, AlphaFold2 (AF), to predict the models of tens of cohesin subcomplexes from different species. The results provide important insight into how the Smc3-Scc1 DNA exiting gate is opened, how Pds5 and Wapl maintain the opened gate, how Pds5 and Scc3/SA recruit different cofactors, how Eco1 and Sororin antagonize Wapl, and how Scc2/Nipbl interacts with Scc3/SA. The models are for the most part consistent with published mutations in these proteins that affect cohesin's functions in vitro and in vivo and raise testable hypotheses of cohesin dynamics and regulation. This study also serves as an example of how to use AF to build models of protein complexes that involve the docking of flexible regions to globular domains.

---

## [Author Response]

The following is the authors’ response to the previous reviews.

We are grateful for the helpful comments of both reviewers and have revised our manuscript with them in mind.

One of the main issues raised was that readers may by default assume that our models are correct. We in fact made it very clear in our discussion that the models are merely hypotheses that will need testing by “wet” experiments and we do not therefore agree that even readers unfamiliar with AF would assume that the models must be correct. It was also suggested that readers could be reassured by including extensive confidence estimates such as PAE plots. As it happens, every single model described in the manuscript had reasonably high PAE scores and more crucially the entire collection of output files, including PAE data, are readily accessible on Figshare at https://doi.org/10.6084/m9.figshare.22567318.v2, a fact that the reviewers appear to have overlooked. The Figshare link is mentioned three times in the manuscript. Embedding these data within the manuscript itself would in our view add even more details and we have therefore not included them in our revised manuscript. Likewise, it is rather simple for any reader to work out which part of a PAE matrix corresponds to an interaction observed in the corresponding pdb prediction. Besides which, it is our view that the biological plausibility and explanatory power of models is just as important as AF metrics in judging whether they may be correct, as is indeed also the case for most experimental work.

Another important point was that the manuscript was too long and not readable. Yes, it is long and it could well be argued that we could have written a different type of manuscript, focusing entirely on what is possibly the simplest and most important finding, namely that our AF models suggest that in animal cells Wapl appears to form a quarternary complex with SA, Pds5, and Scc1 in a manner suggesting that a key function of Wapl’s conserved CTD is to sequester Scc1’s Nterminal domain after it has dissociated from Smc3. For right or for wrong, we decided that this story could not be presented on its own but also required (1) an explanation for how Scc1 is induced to dissociate from Smc3 in the first place and (2) how to explain that the quarternary complex predicted for animal cells was not initially predicted for fungi such as yeast. The yeast situation was an exception that clearly needed explaining if the theory was to have any generality and it turned out that delving into the intricate details of the genetics of releasing activity in yeast was eventually required and yielded valuable new insights. We also believe that our work on the recruitment of Eco/Esco acetyl transferases to cohesin and the finding that sororin binds to the Smc3/Scc1 interface also provided important insight into how releasing activity is regulated. We acknowledge that the paper is indeed long but do not think that it is badly written. It is above all a long and complex story that in our view reveals numerous novel insights into how cohesin’s association with chromosomes is regulated and have endeavoured to eliminate any excessive speculation. We feel it is not our fault that cohesin uses complex mechanisms.

Notwithstanding these considerations, we have in fact simplified a few sections and removed one or two others but acknowledge that we have not made substantial cuts.

It was pointed out that a key feature of our modelling, namely the predicted association of Wapl’s C-terminal domain with SA/Scc3’s CES is inconsistent with published biochemical data. The AF predictions for this interface are universally robust in all eukaryotic lineages and crucially fully consistent with published and unimpeachable genetic data. We note that any model that explains all findings is bound to be wrong for the very simple reason that some of these findings will prove to be incorrect. There is therefore an art in Science of judging which data must be explained and accommodated and which should be ignored. In this particular case, we chose to ignore the biochemistry. Time will tell whether our judgement proves correct.

Last but not least, it was suggested that we might provide some experimental support for our proposed SA/Scc3-Pds5-Scc1-WaplC quaternary complex. We are in fact working on this by introducing cysteine pairs (that can be crosslinked in cells) into the proposed interfaces but decided that such studies should be the topic of a subsequent publication. It would be impossible with the resources available to our labs to follow up all of the potential interactions and we therefore decided to exclude all such experiments.

We are grateful for the detailed comments provided by both reviewers, many of which were very helpful, and in many but not all cases have amended the manuscript accordingly.

With regard to the more specific comments:

**Reviewer #1 (Recommendations For The Authors):**
1. One concern is that observed interfaces/complexes arise because AF-multimer will aim to pack exposed, conserved and hydrophobic surfaces or regions that contain charge complementarity. The risk is that pairwise interaction screens can result in false positive & non-physiological interactions. It is therefore important to report the level of model confidence obtained for such AF calculations:A) The authors should color the key models according to pLDDT scores obtained as reported by AF. This would allow the reader to judge the estimated accuracy of the backbone and side chain rotamers obtained. At least for the key models and interactions it would be important to know if the pLDDT score is >90 (Correct backbone and most rotamers) or >70 (only backbone is correct).B) It would also be important to report the PAE plots to allow estimation of the expected position error for most of the important interactions. pLDDT coloring and PEA plots can be shown side-by-side as shown in other published data (e.g. https://pubmed.ncbi.nlm.nih.gov/35679397/ (Supplementary data)C) The authors should include a Table showing the confidence of template modeling scores for the predicted protein interfaces as ipTM, ipTM+pTM as reported by AlphaFold-multimer. Ideally, they would also include DockQ scores but this may not be essential. Addition of such scores would help classification into Incorrect, Acceptable or of high quality. For example, line 1073 et seq the authors show a model of a SCC1SA and ESCO1 complex (Fig. 37). Are the modeling scores for these interfaces high? It does not help that the authors show cartoons without side chains? Can the authors provide a close-up view of the two interfaces? Are the amino acids are indeed packed in a manner expected for a protein interface? Can we exclude the possibility that the prediction is obtained merely because the sequence segments (e.g. in ESCO1 & ESCO2) are hydrophobic and conserved?

We do not agree that including this level of detail to the text/figures of the manuscript would be suitable. All the relevant data for those who may be sceptical about the models are readily available at https://doi.org/10.6084/m9.figshare.22567318.v2. In our view, the cartoon versions of the models are easier for a reader to navigate. Anyone interested in the molecular details can look at the models directly.

Importantly, no amount of statistical analysis can completely validate these models. What is required are further experiments, which will be the topic of further work from our and I dare from other laboratories.

D) When they predict an interaction between the SA2:SCC1 complex and Sororin's FGF motif, they find that only 1/5 models show an interaction and that the interaction is dissimilar to that seen of CTCF. Again, it would be helpful to know about modeling scores. Can they show a close-up view of the SORORIN FGF binding interface to see if a realistic binding mode is obtained? Can they indicate the relevant region on the PAE plot?

Given that AF greatly favours other interactions of Sororin’s FGF motif over its interaction with SA2-Scc1, we do not agree that dwelling on the latter would serve any purpose.

1. Line 996: AF predicts with high confidence an interaction between Eco1 & SMC3hd. What are the ipTM (& DockQ if available) scores. Would the interface score High, Medium or Acceptable?

As mentioned, see https://doi.org/10.6084/m9.figshare.22567318.v2.

1. Line 1034 et seq: Eco1/ESCO1/ESCO2 interaction with PDS5. Interface scores need to be shown to determine that the models shown are indeed likely to occur. If these interactions have low model confidence, Fig. 36 and discussion around potential relevance to PDS5-Eco1 orientation relative to the SMC3 head remains highly speculative and could be expunged.

See https://doi.org/10.6084/m9.figshare.22567318.v2. It should be clear that the predictions are very similar in fungi and animals. Crucially, we know that Pds5 is essential for acetylation in vivo, so the models appear plausible from a biological point of view.

1. Considering the relatively large interface between ECO1 and SMC3, would the author consider the possibility that in addition to acetylating SMC3's ATPase domain, ECO1 remains bound to cohesin-DNA complex, as proposed for ESCO1 by Rahman et al (10.1073/pnas.1505323112)?

This is certainly possible but we would not want to indulge in such speculation.

1. E.g. Line 875 but also throughout the text: As there is no labeling of the N- and C-termini in the Figures, is frequently unclear what the authors are referring to when they mention that AF models orient chains in a certain manner.

Good point. This has been amended. However, the positions of N- and C- is all available at https://doi.org/10.6084/m9.figshare.22567318.v2.

1. Fig19B: PAE plots: authors should indicate which chains correspond to A, B, C. Which segment corresponds to the TYxxxR[T/S]ΦL motif? Can they highlight this section on the PAE plot?

Good point and amended in the revised manuscript.

Minor comments:1. Line 440: the WAPL YSR motif is not shown in Fig. 14A1. Line 691: Scc3 spelling error.1. Line 931: Sentence ending '... SCC3 (SCC3N).' requires citation.1. Line 1008: Figure reference seems wrong. It should read: Fig. 34A left and right. Fig. 34B does not contain SCC1.

Many thanks for spotting these. Hopefully, all corrected.

1. Fig. 41 can be removed as it shows the absence of the interaction of Sororin with SMC1:SCC1. Sufficient to mention in the text that Sororin does not appear to interact with SMC1:SCC1.

This is possible but we decided to leave this as is.

**Reviewer #2 (Recommendations For The Authors):**
Minor points(1) Are there any predicted models in which one of the two dimer interfaces of the hinge is open when the coiled coils are folded back, as seen in the cryo-EM structure of human cohesin-NIPBL complex in the clamped state?

No AF runs ever predicted half opened hinges. It is possible that the introduction of mutations in one of the two interfaces might reveal a half-opened state and we ought to try this. However, it would not be appropriate for this manuscript, we believe.

(2) Structures of the SA-Scc1 CES bound to [Y/F]xF motifs from Sgo1 and CTCF have been reported, suggesting that a similar motif could interact with SA/Scc3. Surprisingly, AF did not predict an interaction between Scc3/SA and Wapl FGF motifs, which only bind to the Pds5 WEST region. On the other hand, AF predicted interactions of the Sororin FGF motif with both Pds5 WEST and SA CES. Can the authors comment on this Wapl FGF binding specificity? What will happen if a Wapl fragment lacking the CTD is used in the prediction?

This seems to be an academic point as the CTD is always present.